# Computational Rhinology: Unraveling Discrepancies between In Silico and In Vivo Nasal Airflow Assessments for Enhanced Clinical Decision Support

**DOI:** 10.3390/bioengineering11030239

**Published:** 2024-02-28

**Authors:** Sverre Gullikstad Johnsen

**Affiliations:** SINTEF, NO-7465 Trondheim, Norway; sverre.g.johnsen@sintef.no

**Keywords:** computational rhinology, computational fluid dynamics (CFD), large eddy simulation (LES), nasal airflow, nasal resistance, rhinomanometry (RMM), turbulence

## Abstract

Computational rhinology is a specialized branch of biomechanics leveraging engineering techniques for mathematical modelling and simulation to complement the medical field of rhinology. Computational rhinology has already contributed significantly to advancing our understanding of the nasal function, including airflow patterns, mucosal cooling, particle deposition, and drug delivery, and is foreseen as a crucial element in, e.g., the development of virtual surgery as a clinical, patient-specific decision support tool. The current paper delves into the field of computational rhinology from a nasal airflow perspective, highlighting the use of computational fluid dynamics to enhance diagnostics and treatment of breathing disorders. This paper consists of three distinct parts—an introduction to and review of the field of computational rhinology, a review of the published literature on in vitro and in silico studies of nasal airflow, and the presentation and analysis of previously unpublished high-fidelity CFD simulation data of in silico rhinomanometry. While the two first parts of this paper summarize the current status and challenges in the application of computational tools in rhinology, the last part addresses the gross disagreement commonly observed when comparing in silico and in vivo rhinomanometry results. It is concluded that this discrepancy cannot readily be explained by CFD model deficiencies caused by poor choice of turbulence model, insufficient spatial or temporal resolution, or neglecting transient effects. Hence, alternative explanations such as nasal cavity compliance or drag effects due to nasal hair should be investigated.

## 1. Introduction

Computational fluid dynamics (CFD) is an emerging in silico tool in rhinology, leveraging engineering techniques for mathematical modelling of nasal airflow. The interdisciplinary integration of CFD in rhinology is part of computational rhinology, a specialized branch of biomechanics. Computational rhinology has already contributed significantly to advancing our understanding of nasal function, including airflow patterns, mucosal cooling, particle deposition, and drug delivery. Future prospects in computational rhinology encompass the development of virtual surgery as a clinical, patient-specific decision support tool and the refinement of patient selection criteria for treating common nasal airway disorders. These promising advancements may also extend into the broader field of otorhinolaryngology.

The present study derives from a collaborative effort between St. Olavs hospital the University hospital of Trondheim, the Norwegian University of Science and Technology, and the research foundation SINTEF. Our aim is to improve the understanding of obstructive sleep apnea (OSA) by employing engineering tools such as mathematical modelling [1,2,3,4,5,6,7,8,9,10,11]. One notable discovery by the research team underscores the potential of minor anterior nasal surgical intervention (e.g., correcting nasal septum deviation) to alleviate OSA, alone. This was clearly demonstrated in the patient included in **Part III** (Section 4) of this paper, whose OSA markedly improved following surgical septum deviation correction. Other research groups have reported similar positive outcomes [12]. OSA is caused by repetitive collapses of the pharyngeal walls during sleep, and the impact of surgical nasal cavity modification on the onset of OSA is not yet fully understood. Isolated nasal surgery is thus not generally recommended as the first-line treatment for OSA [13], and there are no objective clinical methods available to identify patients who will benefit from such surgery.

CFD has been proposed as an attractive objective tool for predicting how alterations to the upper airways affect patient-specific airflow. However, its effectiveness in such detailed applications is hindered by the lack of in vivo nasal airflow measurements, which are essential for validating CFD models. In vivo rhinomanometry (RMM) stands as the sole method capable of supplying clinical nasal airflow data. While CFD has demonstrated reliability when compared to in vitro airflow measurements in physical nasal cavity replicas, in silico (CFD-based) RMM has been reported to severely underpredict the nasal resistance measured by in vivo RMM, without adequate explanations given. This apparent paradox was the primary driver for the current study. To empower CFD as a practical, patient-specific clinical decision support tool, it is vital to understand the possible reasons for the gross disagreement observed when comparing in silico and in vivo RMM. While this paper does not conclusively resolve this issue, it presents new evidence and discussions to narrow down the list of possible explanations, offering a solid foundation for future development and utilization of CFD-based simulation tools for improved understanding of (patient-specific) nasal function and as clinical decision support in rhinology.

This paper is written to be accessible to readers at all levels, from beginners to experts, in the disciplines of otorhinolaryngology and CFD. Thus, it aims to bridge the gap between these traditionally distinct scientific fields. This paper not only serves as an introductory guide to computational rhinology but also explores and discusses unresolved controversies, dilemmas, and paradoxes within the field and presents new evidence that will help unravel the current disparity between measurements and simulations in rhinology. This paper includes a comprehensive bibliography, meticulously compiled through a combination of ancestry and descendancy literature review approaches, utilizing internet-based publication databases such as Google Scholar, Researchgate, PubMed, and journal web pages. Additionally, it offers novel results from finely resolved large eddy simulation (LES)-based CFD simulations of active anterior RMM. Key terms and concepts regarding rhinology and CFD are presented in Section 2.1, Section 2.2, Section 2.3 and Section 2.4, and an overview of nomenclature and abbreviations is provided in the back matter. Raw data from the simulations are available in the Appendix A [14].

This paper is structured into three main parts. **Parts I** and **II** can be read independently, but **Part III** relies on the two former parts in the sense that these serve as the scientific background of the in silico study performed.

**Part I** (Section 2 Computational Rhinology) offers a general overview of computational rhinology, with a particular focus on nasal airway obstruction and subjective and objective clinical measures. It also traces the evolution of CFD as a clinical decision support tool over the last decades. A comprehensive review is given of important research questions whose answers are crucial for the adoption of CFD as a clinical decision support tool.

**Part II** (Section 3 Overview of Published Literature on in vitro and in silico Nasal Airflow Studies) provides a bibliographic survey, encompassing research involving in vitro measurements in physical nasal cavity replicas and studies focusing on in silico experiments conducted in digital nasal cavity models. These studies are classified by the nature of their inflow boundary conditions (steady state vs. transient) and modeling techniques and the citations are summarized in tabular format.

**Part III** (Section 4 In silico RMM Simulation Results) presents novel patient-specific findings derived from in silico simulations of active anterior RMM, achieved through high-fidelity, transient LES simulations. This part presents one of the most detailed CFD simulations of nasal airflow to date, including three breathing cycles on each side of the nose within fully resolved simulations. The simulations were designed to elucidate core research queries highlighted in **Parts I** and **II**, specifically focusing on whether the discrepancy between in vivo and in silico RMM can be attributed to CFD model deficiencies such as the following:Unresolved transitional or turbulent effects.Unresolved spatial phenomena such as vortices and eddies.Transient effects like hysteresis, developing boundary layers, and meandering.

Assessment of the finely resolved LES-based CFD simulations indicated that they were highly accurate, providing a reliable benchmark for the present study. They revealed that neither pronounced transitional/turbulent nor transient effects are significant contributors to the observed disparities. In conclusion, a pseudo-steady laminar model with a relatively coarse computational mesh yields near-perfect predictions compared to the transient, fully resolved LES model. While the former model can be run within hours on a desktop computer, the latter required approximately 3 million CPU-hours.

## 2. Part I—Computational Rhinology

### 2.1. Rhinology

In humans, the nasal cavity is the primary conduit for lung ventilation, supplying the body with fresh oxygen while expelling carbon dioxide. Additionally, it plays a vital role in the olfactory system by transporting odors to the olfactory sensory system. It is a highly intricate flow channel that is optimized for various functions, including humidification and heating of inhaled air as well as air filtering through particle deposition [15,16].

While nasal breathing is the typical mode of respiration, there are situations when nasal breathing alone may not provide sufficient oxygen saturation in the blood, and oral breathing becomes necessary. Oral breathing may be required in cases of impaired nasal patency, where the nasal passages are partially blocked or restricted, hindering effective airflow. Oral breathing lacks, however, most of the traits of nasal breathing, and it is well known that excessive oral breathing has adverse effects on, e.g., dentofacial development, oral health, and digestive and breathing disorders (e.g., obstructive sleep apnea) [17].

In the medical field, the study of the nasal cavity falls within the discipline of otorhinolaryngology, with a specific focus on the nasal cavity known as rhinology. While rhinology aims to enhance diagnostic and treatment approaches for nasal and sinonasal disorders, its significance extends beyond its primary domain. Otorhinolaryngology, as a whole, benefits from the advancements made in rhinology. Additionally, the knowledge and expertise of rhinologists are essential in the fields of allergy/immunology, pulmonology, sleep medicine, head and neck surgery, and facial plastic and reconstructive surgery. It follows that advancements in understanding nasal physiology and the complexities of nasal airflow can lead to improved diagnostic techniques and innovative treatment strategies across a wide range of disorders within these associated medical fields.

Nasal airway obstruction (NAO), which can be caused by structural abnormalities or deformities such as deviated septum, nasal polyps, turbinate hypertrophy, nasal injuries, or other underlying causes, is an important subtopic of rhinology. It is generally accepted that NAO affects the nasal airflow pattern and correlates with symptoms such as nasal congestion, trouble breathing, sleep-disordered breathing, and others. Mathematical modelling of nasal airflow has been suggested as an important tool to complement objective clinical measurements in the assessment of NAO and evaluation of treatment options [18,19,20].

### 2.2. The Relationship between Obstructive Sleep Apnea and Nasal Airway Obstruction

Obstructive sleep apnea (OSA) is a common breathing disorder caused by recurrent, temporary upper airway collapses during sleep [21]. The collapses are primarily attributed to the Venturi effect, triggered by accelerated airflow through constrictions enveloped by soft tissue within the oropharyngeal tract. A pressure difference between the soft tissue and the airway may cause partial (hypopnea) or complete (apnea) respiratory blockages when neuromuscular response is impaired or relaxed. Interruptions in breathing can lead to frequent awakenings or transitions from deeper to lighter stages of sleep, resulting in overall reduced sleep quality and leading to symptoms of daytime sleepiness, fatigue, and reduced cognitive function.

It is widely acknowledged that OSA can have severely negative effect on patients’ health and wellbeing, as it is correlated with conditions such as cardiovascular diseases, metabolic syndrome and diabetes, and learning disabilities and cognitive development [22,23,24,25,26]. Moreover, OSA has implications for tasks such as operating motor vehicles due to national regulations and driver’s license restrictions [27]. Insurance companies also approach individuals with OSA differently from their healthier counterparts, potentially affecting coverage eligibility [28]. The concern arises that professional truck drivers might avoid OSA screening to circumvent challenges with their driver’s licenses or life insurance coverage.

Various treatment options are available for OSA, including continuous positive airway pressure (CPAP), mandibular advancement devices (MAD), and surgical intervention, complemented by lifestyle adjustments and weight management. The severity of OSA is conventionally quantified using the apnea–hypopnea index (AHI), representing the count of obstructive incidents per hour of sleep.

Young et al. [29] conducted a study that revealed a significant association between NAO and sleep-disordered breathing, including conditions such as snoring and OSA. This finding has been supported by several other studies, demonstrating the negative impact of high nasal flow resistance on snoring and OSA [30,31,32,33]. Singh et al. [17] pointed out that oral breathing is mainly caused by NAO and that several aspects of oral breathing affect OSA adversely. There is, however, weak correlation between AHI and patients’ subjective assessment of nasal symptoms [34]. Hoel et al. [35,36] found that patients with OSA and increased nasal resistance had a higher ratio of hypopneas to apneas.

Scott and Kent [37] advocated the vital role of the nasal cavity in breathing, noting that 90 percent of airflow occurs through the nasal cavity and approximately 60 percent of flow resistance is attributed to the nasal passages. Understanding nasal airflow is therefore crucial for comprehending the development of OSA. They also acknowledged the complexity of surgical interventions, stating that individualized approaches are necessary for optimal outcomes. However, the success rate of septoplasty, a common surgical procedure to alleviate NAO, varies by between 43 and 85 percent based on objective and subjective measures [38].

In the context of CPAP treatment for OSA, Nakata et al. [39] emphasized the significance of addressing nasal resistance, stating that increased nasal resistance can contribute to CPAP failure. They suggested that surgical correction of severe nasal obstruction should be considered to enhance the effectiveness of CPAP therapy.

A recent review found conflicting meta-studies regarding the effect of nasal surgery on AHI and concluded that isolated nasal surgery should not generally be recommended as the first-line treatment for OSA [13]. There are, however, studies that indicate that septoplasty has a pivotal role in combination with inferior turbinate surgery [7,40] or multilevel palate and/or tongue surgery [41], to improve AHI.

To improve the effectiveness of NAO treatment, it is anticipated that patient-specific planning tools, such as mathematical models, can serve as valuable decision support tools [42]. These have the potential to enhance treatment outcomes by providing personalized insights and guidance for surgical interventions and other treatment options.

### 2.3. Clinical Evaluation of Nasal Patency

Nasal patency refers to the degree or extent to which the nasal passages are open and unobstructed [43]. It is influenced by various factors, including the size and shape of the nasal passages, the condition of the nasal mucosa (lining of the nose), the presence of any anatomical abnormalities or obstructions, and the function of the nasal gateway. Because nasal patency is not a physical quantity that can be measured directly, it is the role of the rhinologist to use various clinical measures to assess the nasal patency.

Objective, clinical measurements to evaluate nasal patency include rhinomanometry (RMM), acoustic rhinometry (AR), and peak nasal inspiratory flow (PNIF) in addition to endoscopy and medical imaging techniques such as computed tomography (CT) or magnetic resonance imaging (MRI) [44,45,46]. Despite the objective nature of these measurements, their results must be scrutinized by a medical expert to exploit their diagnostic potential.

Subjective measures include self-reporting questionnaires designed to assess patients’ subjective experience and perception of their nasal patency and quality of life by marking the perceived level of a specific symptom on a prescribed scale. Popular measures include variations of the Visual Analog Scale (VAS) and Nasal Obstruction Symptom Evaluation (NOSE) scale [47] among others.

Computational fluid dynamics (CFD) is an emerging, objective tool in rhinology. It is based on the mathematical modelling of nasal airflow and has the potential to be an important supplement to current clinical measures to advance the understanding of nasal airflow [18,19,20].

Many studies have investigated the correlation between the various objective and subjective measures, but the results are not conclusive.

Several studies [44,48,49,50] indicate that the objective standard measurements performed during clinical exam have limited value in the diagnosis of NAO patients. Subjective sensation of nasal obstruction may be caused by several factors other than objective nasal flow resistance, but unilateral nasal flow resistance correlates better with subjective sensation of nasal patency than bilateral nasal flow resistance [45]. Mozzanica et al. [51] reported significant correlation between subjective sensation of nasal patency, measured with I-NOSE and VAS questionnaires, and RMM data, however, and Hueto et al. [33] highlighted the usefulness of RMM in determining the appropriate pressure settings for CPAP treatment in OSA patients. Preoperative clinical evaluation commonly involves objective measures, and the clinical value of RMM should not be downplayed [52,53].

Zhao and Dalton [18] pointed out that standard rhinometric measurements are poorly correlated with patient-reported subjective symptoms and questioned their clinical value for evaluation of nasal obstruction and subjective evaluation of treatment outcome. Eccles and co-authors [54,55] effectively showed how subjective sensing of nasal patency can be decoupled from objectively measured nasal resistance by application of menthol, which affects the sensory ability of the trigeminal nerve endings. The dissociation between subjective and objective evaluation of nasal patency is evident in empty nose syndrome (ENS), which is a condition that typically occurs after surgical procedures to alleviate NAO and minimize nasal resistance. Paradoxically, some individuals who undergo these surgeries may experience increased sensations of NAO, among other symptoms, even though objective tests show that their nasal passages are open. Impaired trigeminal nerve function has been pointed to as an explanation for ENS [56]. Malik et al. [57] studied how the formation of a middle meatus jet stream is characteristic for ENS patients. Di et al. [58] and Li and co-authors [59,60] studied nasal aerodynamics in ENS patients using CFD.

It can be deduced that the same sensors that cause subjective sensing of nasal patency are responsible for subjective sensing of heat exchange between the nasal airflow and the nasal tissue, and several studies have demonstrated correlations between mucosal cooling and temperature, obtained from CFD simulations, and patients’ subjective evaluation of nasal patency [50,61,62,63,64,65,66,67,68].

Obviously, both local mucosal cooling and overall nasal resistance depend on the nasal airflow pattern. Whereas subjective sensation of nasal patency may depend more on the cooling effect, transport and deposition of nasal spray may be more closely correlated with nasal resistance [69]. The fact that the nasal airflow pattern is highly sensitive to local nasal anatomy/geometry suggests that objective tools such as CFD are required to complement standard clinical flow characterization techniques such as RMM and bridge the gap between subjective and objective clinical measurements.

#### 2.3.1. Nasal Flow Resistance

Flow resistance, R, is an intrinsic flow channel-specific attribute that correlates the pressure drop over the channel, ΔP, and the volumetric flowrate, Q;
(1)ΔP=RQ.

The flow resistance can vary with the flowrate, and its characteristics can be determined by measuring the pressure drop for known flowrates, or vice versa. For pressure-driven flows such as respiratory airflow, Equation (1) offers insight into how the breathing effort (quantified as pressure drop) must increase to maintain a consistent flowrate when encountering heightened flow resistance. For instance, to achieve an adequate inspiratory flowrate in an obstructed airway, greater intrathoracic negative pressure is required than in an unobstructed airway. This offers a simple explanation for why oral breathing may become the favored mode of respiration in the presence of NAO. In general, heightened flow resistance serves as an indicator of diminished patency.

The nature of (internal) fluid flow heavily depends on the geometry of the flow channel. e.g., for fully developed, steady, laminar, single-phase flow in a straight channel, the flow resistance is inversely proportional to the hydraulic diameter, $D_{h}\equiv 4A/O$, raised to the fourth power (Hagen–Poiseuille equation);
(2)$$R=\frac{128L\mu }{\pi D_{h}^{4}}\;,$$
where A, O, and L are the cross-sectional area, perimeter, and length of the flow channel, respectively, and μ is the dynamic viscosity of the fluid.

In complex flow channels such as the nasal cavities, slight changes in the geometrical features may cause unpredictable airflow response. e.g., it can be imagined that in parts of the nasal cavity featuring narrow passages such as in the nasal vestibule or the olfactory slit, a slight modification may reduce the perimeter considerably without affecting the cross-sectional area notably. This may severely impact the local hydraulic diameter. Moreover, flow instabilities or recirculation zones triggered by geometrical features (e.g., abrupt changes in flow direction or cross-sectional area) may result in effective cross-sections smaller than the actual cross-section. It is not obvious that the concept of hydraulic diameter and associated standard flow resistance correlations are applicable for nasal airflow [70,71,72,73].

A peculiar feature of the nasal passages, known as the nasal cycle, is caused by temporal, asymmetric swelling and deswelling of the nasal mucosa. This effect causes an intermittent variation in nasal resistance that can be observed through the sensation of unilateral nasal obstruction. It is believed that this effect is important for the removal of deposited dust particles, etc. Not all humans have it, however, and the periodicity of the phenomenon is neither the same between different individuals nor constant in the same individual [74,75]. The nasal cycle is expected to affect unilateral nasal resistance measurements.

Nasal cavity compliance may permit local expansion or contraction of the nasal cavity cross-section due to periods of over- and under-pressure occurring during the respiratory cycle. This may cause local pressure dependency in the hydraulic diameter, hence the nasal resistance, which may introduce asymmetry with respect to exhalation/inhalation and temporal effects such as hysteresis in the pressure–flow relationship (Equation (1)).

#### 2.3.2. Rhinomanometry

Rhinomanometry (RMM) is the only technique in clinical use that allows for quantitative assessment of the respiratory function of the nose [76], and it is thus of significant importance for the calibration and validation of patient-specific mathematical models of nasal air flow. The theory and background have been thoroughly covered by Vogt et al. [77]. RMM is a method that measures the pressure drop in the nose as a function of volumetric air flowrate. The resulting pressure–flow curves relate the volumetric flowrate to the pressure drop. The measured volumetric flow and pressure drops form the basis for calculation of the nasal resistance and estimation of representative hydraulic diameters [78]. Figure 1 shows an example of a pressure–flow curve.

Whereas bilateral RMM considers the simultaneous measurement of both nasal passages, unilateral RMM considers only one nasal passage at the time by occluding one nostril while assessing the airflow in the open passage. It follows from the definition (Equation (1)) that the reciprocal bilateral resistance is the sum of the reciprocals of the individual unilateral resistances of the two nasal passages:(3)$$\frac{1}{R_{bi}}=\frac{1}{R_{uni,left}}+\frac{1}{R_{uni,right}}.$$

In posterior RMM, the pressure is measured directly in or close to the nasopharynx using a pressure probe typically inserted via the oral cavity. In anterior RMM, one nostril is closed, and the pressure probe is inserted into the nasal vestibule behind the occlusion to provide an indirect measurement of choanae pressure. It has been shown that posterior and anterior RMM are equivalent with respect to (unilateral) pressure measurement [79]. Passive RMM is performed by enforcing external nasal airflow. More commonly employed is active RMM, where the patient’s own physiological airflow is utilized. The RMM procedure combining active breathing with anterior measurement is denoted as active anterior RMM (AAR).

Because the nasal passages are lined with a mucosal membrane subject to unpredictable temporal variations in swelling, RMM is typically performed twice, before and after decongestion. Application of topical nasal decongestant (e.g., xylometazoline) or physical exercise serves to eliminate the vascular component of nasal obstruction caused by swelling of the turbinates, allowing for quantification of the anatomical component of NAO [80]. The anatomical component is determined solely by the rigid tissue of the nose (e.g., bone and cartilage), and it is thus related to the maximal nasal volume, independent of the nasal cycle [45]. In clinical rhinology, such decongestion tests are performed to quantify the different roles of the skeleton and mucosa in NAO. For the mathematical modelling of airflow in the nasal cavities, it is convenient to disregard the complexities associated with soft tissue and temporal variations in unilateral nasal resistance.

#### 2.3.3. Mathematical Illustration of the Principles of Rhinomanometry and Inherent Hysteresis Using Bernoulli’s Equation

The mathematical background of rhinomanometry has been thoroughly covered by others [77]. Here, a brief illustration of the concept of rhinomanometry is provided, based on the mathematical description of simple pipe flow.

In pipe and duct flow engineering, it is common practice to estimate pressure drops by utilizing Bernoulli’s equation [81]. For unsteady, fully developed, horizontal, incompressible flow through a straight, rigid duct of constant cross-sectional area, the pressure drop per unit length results from the contributions of a friction term and an unsteady inertial term,
(4)$$\frac{\Delta P}{L}=\underset{\mathrm{flow}\;\mathrm{resistance}\;\mathrm{per}\;\mathrm{unit}\;\mathrm{length},\;R/L}{\underset{︸}{\frac{8\rho }{\pi^{2}D_{h}^{5}}\left[\underset{\mathrm{friction}\;\mathrm{term}}{\underset{︸}{f_{D}Q}}+\underset{\mathrm{unsteady}\;\mathrm{term}}{\underset{︸}{\frac{\pi D_{h}^{3}}{2}\frac{\partial \ln Q}{\partial t}}}\right]}}Q\;,$$
where ρ is the fluid mass density and $f_{D}$ is the Darcy friction factor,
(5)fD=fD,lam=64/ReforRe≤2000fD,turb(Re,εr)forRe≥4000 ,
where Re is the Reynolds number, which for internal duct flow, can be expressed as
(6)Re=4Q/πνDh ,
where ν=μ/ρ is the fluid kinematic viscosity and $\varepsilon_{r}$ is the relative wall roughness. Equation (4) reduces to Equation (2) for laminar flow (Re≤2000) with ∂Q/∂t=0. Several alternative formulations exist for the turbulent friction factor, e.g., the Haaland equation [82],
(7)1fD,turb=−1.8log10εr3.71.11+6.9Re.

In laminar flow, the flow is characterized by smooth, locally parallel streamlines, and flow variables behave deterministically. In the case of turbulent flow, however, flow variables behave stochastically. In duct flow engineering, the flow is typically considered laminar for Reynolds numbers below 2000 and turbulent for Reynolds numbers above 4000, but these thresholds may vary depending on the specific flow configuration. In the intermediate range, the flow is transitional, which is a generally poorly understood, complex and dynamic flow state. For transitional flow, the friction factor can be approximated by a smooth, weighted average of the laminar and turbulent friction factors, $f_{D,tran}\approx wt\cdot f_{D,lam}+(1-wt)\cdot f_{D,turb}$, where the weight, wt, varies smoothly between 1 and 0 in the laminar and turbulent regimes, respectively.

To improve the general understanding of RMM, a simplified model is used to represent nasal airflow, based on a straight, smooth ($\varepsilon_{r}=0$) duct of a given hydraulic diameter and fully developed sinusoidal (respiratory) flow,
(8)$$Q=Q_{\max}\sin\left(\omega t\right)\;,$$
where t is the time variable, ω=2π/τ is the angular frequency, and τ is the period of the respiratory cycle. Employing this model, synthetic RMM pressure–flow curves can be generated to study the impact of the various parameters. In Figure 2, data are shown for $Q_{\max}=600\;\mathrm{mL}/s$, τ=5 s, and $D_{h}=10\;\mathrm{mm}$, to compare the effects of assuming laminar or turbulent flow and to illustrate the effect of the unsteady term. Figure 2a shows that there is a phase shift between the flowrate and pressure drop due to the unsteady term. In Figure 2b, it can be seen that this causes a hysteresis effect such that the pressure–flow curve does not pass through the origin.

The hysteresis width is found by evaluating Equation (4) at Q=0,
(9)$$W_{\Delta P}=\frac{8\rho L\omega Q_{\max}}{\pi D_{h}^{2}}\;,$$
where Equation (8) was used. It follows that the relative hysteresis width, defined as the width of the pressure–flow curve hysteresis at the level of Q=0 divided by the maximum laminar pressure drop, and can be expressed as
(10)$$W_{\Delta P,rel}=\omega D_{h}^{2}/16\nu .$$

Figure 3 illustrates how the relative hysteresis width increases for increasing hydraulic diameter while the flow resistance per unit length decreases.

In general, the following can be observed through analysis of Equation (4):Steady, laminar pressure–flow curves are straight lines passing through the origin.The slope of the pressure–flow curve decreases with increasing flow resistance. That is, laminar pressure–flow curves are steeper than turbulent ones due to the higher friction factor in turbulent flow. Distinct change in slope in in vivo RMM pressure–flow curves may thus be an indication of transition to turbulence.The unsteady flow resistance term causes a hysteresis effect, such that the pressure–flow curve becomes a closed loop not passing through the origin.The relative hysteresis width is determined by the hydraulic diameter and the period of the respiratory cycle.

The complex shape of the nasal cavity as well as effects of lateral wall movement may complicate this picture considerably for nasal airflow. Blevins [81] provides an overview of applicable methods to approximate friction factors and pressured drop for channels with noncircular cross-sections, bends, changes in flow area, etc.

### 2.4. Computational Fluid Dynamics in Rhinology

Computational fluid dynamics (CFD) combines numerical mathematics, computational sciences, and fluid dynamics to solve partial differential equations that represent the conservation laws of fluid dynamics. By utilizing computers for numerical solutions, CFD enables the analysis of complex fluid dynamics problems. For over five decades, CFD has been extensively employed in various industries, such as automotive and aerospace sectors and process industries. It has become a fundamental component of industrial research and development, providing cost-effective alternatives to performing costly experiments through rapid in silico prototyping. This approach reduces the number of required experiments, mitigating risks and costs. CFD serves multiple purposes, including (1) analyzing and gaining deeper insights into experimental results and observations; (2) supporting experiment design and planning; and (3) facilitating industrial process control. One significant advantage of CFD is its ability to provide detailed understanding of processes and phenomena that are impractical or impossible to directly observe in situ. CFD serves as a valuable tool for both forward (when the cause is known) and backward (when the effect is known) causality mapping. This versatility makes CFD an excellent diagnostics tool for applications in both industrial and medical domains. For example, CFD may be used to analyze the effects of virtual surgery on patient-specific computer models prior to actual surgery, in order to provide objective decision support for medical personnel.

All the steps in the creation of a high-quality CFD model are prone to errors and uncertainties, and an important part of the job as a CFD engineer is the reiteration and improvement of each step until the model performs adequately. While best practice guidelines exist and experience helps, model requirements may vary between different flow situations, and this can be a meticulous and time-consuming process. To utilize CFD as a clinical tool for decision support, there is a need for standardization of best practice guidelines. On one hand, due to relatively large variability between patients, there will be some degree of uncertainty regarding the accuracy of employed standard methods. On the other hand, this variability is an argument for employing the patient-specific diagnostics that only CFD can offer. Despite extensive work over the last decades, there is still controversy regarding best practice for CFD simulation of flow in the upper airways [83,84,85].

#### 2.4.1. Virtual Surgery

The concept of virtual surgery envisions the use of digital, patient-specific models for the purpose of simulating the effect of surgical procedures or alternative treatment options on a computer. This may be carried out as part of clinical preoperative planning or theoretical research. This approach offers the potential to provide objective tools that can prove invaluable in optimizing individualized treatments while simultaneously reducing risks and costs. However, for clinical applications to be successful, it is imperative that these virtual surgery tools possess two key attributes: speed and accuracy. Furthermore, the software’s user interface and automated workflow should be designed to eliminate the need for involvement from a CFD expert. It is in addressing these crucial requirements that significant challenges lie. In contrast, when it comes to scientific research, the demands for speed and user-friendliness may not be as stringent. This is because research activities often have access to cross-disciplinary expertise and the luxury of time.

Borojeni et al. [67] pointed out three main reasons that CFD-based virtual surgery is likely to have an important role in future clinical applications: (1) The subjective sense of nasal patency is primarily affected by local mucosal cooling, for which there are no available clinical measurement techniques. However, it can readily be estimated by CFD simulations. (2) Subjective assessment of nasal resistance correlates stronger with unilateral than bilateral airflow. (3) The inherent ability of CFD simulations to predict how anatomical changes will affect nasal flow distribution and other flow parameters. They proceeded to present normative ranges for selected airflow parameters to form targets for future nasal obstruction surgery planning.

Simulation-based virtual surgery software has already been demonstrated [86,87,88]. Vanhille et al. [89] created a virtual surgery planning software tool using CFD and tested it in a clinical setting by collecting feedback from nine surgeons. Moghaddam et al. [90] published a systematic virtual surgery method to select septoplasty candidates and predict surgical outcome using CFD. They foresee that their method can be used for fully automatic virtual septoplasty.

#### 2.4.2. The Creation and Utilization of a CFD Model

Briefly, the process of creating and using a CFD model requires the following steps:Acquisition and preparation of an adequately accurate digital model of the flow geometry (airway).Spatial discretization of the geometry model to obtain a computational mesh on which the governing equations of the CFD model can be numerically solved.Setting up the flow physics, e.g., which physical phenomena to include, boundary conditions, fluid and solid material properties, etc.Determination of solution strategy, e.g., steady state or transient formulation, which numerical scheme to use, turbulence models, convergence criteria, etc.Running the simulation until convergence.Evaluation of the accuracy of the simulation. In case of unsatisfactory results, return to an earlier point, implement necessary improvements and modifications to the model, and repeat the process.

When the CFD model is finalized, it can be used to extract information about the flow, such as local pressures, temperatures, flow velocities, wall shear stresses and heat fluxes, etc.

It is outside the scope of the current paper to elaborate on the details of each step of the process, and the reader is referred to textbooks on CFD by, e.g., Patankar [91], Anderson [92], Versteeg and Malalasekera [93], Rodriguez [94], and Roychowdhury [95], as well as the user and theory guides of available CFD software. Selected topics are discussed briefly below.

#### 2.4.3. Acquisition and Preparation of the Digital Airway Geometry Model

Realistic, digital airway geometry models can be acquired from medical imaging data (CT, MRI) through the process known as segmentation. The segmentation process typically produces a surface mesh consisting of triangles identified by the three-dimensional Cartesian coordinates of their vertices and normal vector (stereolithographic format). The surface mesh can be converted into volumetric models in CFD pre-processing software. A recent overview of the process was given by Cercos-Pita [96].

Airway geometry surface meshes are created from a set of two-dimensional bitmap images by tracking predefined contrast levels corresponding to the interface between air and tissue. The state of the art is to use semi-automatic segmentation software where only minor manual adjustments are needed after most of the segmentation is performed automatically, based on predefined default or user-provided parameters. The contrast level determining the air–tissue interface is typically given in terms of the Hounsfield unit [97,98]. HU = −1000 corresponds to air, while HU = 0 corresponds to water. Depending on its density, bone is represented by HU in the range 300 to 2500. There is no consensus about the appropriate level to describe the air–tissue interface (e.g., the mucous layer), and the literature reports HU levels used in the range −800 to −300 [99].

In general, an HU threshold closer to the value of air will result in narrower airway geometry, while an HU threshold closer to bone will provide more voluminous geometry. Due to the relatively coarse resolution of medical images compared with the width of the narrow passages in the nasal cavity, gross effects can be observed by inclusion or exclusion of a single layer of pixels around the edge of the airway. Aasgrav [4] used CFD to show that reducing the segmented airway cross-sectional area by removing one pixel around the perimeter in every CT slice used for segmentation corresponded to reducing the HU threshold from −300 to −600 and led to a twofold increase in flow resistance. This was later supported through observations reported by Cherobin et al. [100].

Quadrio et al. [101] indicated that CFD modelling results were robust with respect to the quality of the CT scan. Cherobin et al. [100] highlighted the uncertainties related to interpretation of CT images in the creation of 3D geometries for CFD modelling and evaluated the impact on various flow parameters by changing the Hounsfield unit threshold used in segmentation. They found that “CFD variables (pressure drop, flowrate, airflow resistance) are strongly dependent on the segmentation threshold”.

Depending on the quality of the surface mesh resulting from the segmentation process, additional pre-processing might be required prior to subsequent steps towards a CFD model. e.g., it may be necessary to improve the quality of the surface mesh by eliminating geometrical artefacts, errors, and unnecessary/unphysical details and to convert the surface mesh into a volumetric format.

#### 2.4.4. Computational Meshes

Computational mesh (or grid) refers to the spatial discretization required for the numerical solution of the governing equations of CFD. Rodriguez [94] presented best practice guidelines in establishing computational meshes for CFD simulations, and Lintermann [102] gave an introduction to the creation of computational meshes for the nasal cavity, in particular. There are three main aspects to consider when establishing a computational mesh for CFD simulations, namely, the mesh type, quality, and size.

Various CFD solvers may have different requirements and preferences regarding the mesh quality (e.g., length-to-width aspect ratios, skewness, and orthogonality) and permitted grid cell types (e.g., hexahedral, tetrahedral, polyhedral). The choice of mesh type may have implications for the efficacy and accuracy of the numerical solution. For complex three-dimensional geometries, such as the nasal cavity, tetra- or polyhedral cells are preferred due to the versatility of these grid generation algorithms. Near-wall boundary layers are commonly resolved using prismatic cells. Bass et al. [103] and Thomas and Longest [104] discussed the pros and cons of tetra- and polyhedral meshes in the CFD modelling of respiratory flows.

Flow structures smaller than the grid cells can, in general, not be captured by the numerical solution. Although there is no guarantee, the accuracy of the numerical solution can thus, to some extent, be expected to improve with the number of grid cells (mesh size), since this permits improved resolution of the flow fields. Caution is advised, however, when increasing mesh size, since there are many pitfalls associated with blindly increasing the number of grid cells. Grid convergence studies should be performed to assess the numerical solutions’ dependency on the grid refinement. It is common practice to assume that the solution is accurate if grid refinement has little impact on key flow features.

The required computational power also generally increases with an increasing number of grid cells. Hence, numerical accuracy may be limited by available computational power. The literature review by Inthavong et al. [105,106] indicates that computational mesh sizes increased exponentially between 1993 and 2017. This corresponds well with Moore’s law, which is based on the historical observation that available computational power has grown exponentially over time.

#### 2.4.5. Flow Physics

After the geometry and mesh are established, which physical phenomena to include in the simulation must be determined. In Navier–Stokes-based CFD, the basic equations that need to be solved in a transient (time-dependent) respiratory flow problem are the transient continuity and momentum equations. It can be assumed that respiratory flow is incompressible, single-phase, and inert, so these can be formulated as
(11)∇⋅u=0 ,
and
(12)$$\partial_{t}u+\left(u\cdot \nabla \right)u=-\left(1/\rho \right)\nabla P+\nu \nabla^{2}u\;,$$
respectively. Body forces (e.g., gravity) are neglected, ∇ and $\partial_{t}$ denote the gradient and time derivative operators, u and P are the local instantaneous flow variables (velocity vector and pressure), and ρ and ν are the constant mass density and kinematic viscosity, respectively. If heating/cooling effects are included, an additional equation for the fluid temperature, T, must be considered,
(13)$$\partial_{t}T+\nabla \cdot \left(uT\right)=\left(k/\rho c_{P}\right)\nabla^{2}T\;,$$
where it is assumed that air is thermally perfect and k and $c_{P}$ are the constant thermal conductivity and specific heat capacity, respectively. If solid tissue, mucous, air humidity, airborne particles, non-constant material properties, or other complicating factors are considered, additional equations, variables, and terms are needed. In particular, if movement of the interface between air and soft tissue is considered, additional equations are needed to describe how the deformation of the airway and the soft tissue are interdependent and affected by local stresses on both sides of the air–tissue interface. This is known as fluid–structure interaction (FSI) modelling.

The Lattice–Boltzmann (LB) method is an alternative to the Navier–Stokes-based CFD and is mentioned here for completeness due to its suitability for fluid dynamics simulations in complex geometries. One of the method’s main strengths is its scalability on high-performance computers [102].

In laminar flow, the flow variables are generally considered deterministic, and they can be predicted precisely from the governing flow equations given above. The flow is characterized by smooth, locally parallel streamlines. In the case that inertial terms are dominating over the viscous terms in the momentum equation, however, the flow may be turbulent. The ratio between inertial and viscous terms are typically expressed by the Reynolds number (Equation (6)). Reynolds numbers above a certain threshold is commonly used as a criterium for considering turbulence or not, but care should be taken since the critical Reynolds number for laminar–turbulent transition can be sensitive to the flow configuration and fluid properties. In turbulent flow, the flow variables behave stochastically, and this is a notoriously difficult physical problem to describe. A wide range of modelling strategies and methods have been developed for the CFD modelling of turbulent flow, including popular approaches like Reynolds-averaged Navier–Stokes (RANS) methods, large eddy simulation (LES), and direct numerical simulation (DNS). It is beyond the scope of the current paper to discuss and compare the various turbulence modelling approaches in CFD, in detail. A systematic overview of advantages and limitations associated with the most popular turbulence models were presented by Ashraf et al. [107]. Details regarding the mathematical description of available turbulence models can be found in classical textbooks by, e.g., Tennekes and Lumley [108], Pope [109], or Wilcox [110], and in CFD simulation software user and theory guides. See, e.g., ANSYS Best Practice guidelines [111,112] and Theory guide [113] and the NASA turbulence modeling resource [114].

Boundary Conditions are required to describe the flow variables at all flow domain boundaries, i.e., inlets, outlets, and walls. Boundary conditions take the form of specifying variable values or gradients at the boundaries. Typical examples include specifying the mass flowrate at the inlet, the pressure at the outlet, and zero velocity at walls (no-slip condition), but other variants are also possible.

Initial conditions denote the initial flow variable fields used as a starting point for time evolution of transient solutions or for the search of a steady state solution.

Finally, if steady flow is considered, the time derivative terms are set to zero, $\partial_{t}\equiv 0$, and all variables are constant in time, everywhere. It is noted that steady boundary conditions do not generally guarantee steady flow alone if the flow is inherently unstable.

#### 2.4.6. Correlations between CFD and Clinical Measures of Nasal Patency

The nature of CFD is to predict objective physical quantities, and it is natural to think that CFD must be able to reproduce objective clinical measures. Although correlations between subjective and objective clinical measures are disputed, some authors have reported that subjective measures and RMM are correlated [51]. This spurs optimism towards predicting subjective sensation of nasal patency through CFD simulations, adding clinical value to virtual surgery. However, it has been pointed out by several authors that the subjective sensing of nasal patency might not be a measure of the objective flow resistance, but rather the cooling effect of the nasal mucosa [68]. In this case, CFD must be correlated with subjective measures, directly, because no in vivo measurements exist to measure mucosal temperature or heat flux. Frank-Ito and Garcia [115] presented an in-depth review of the clinical implications of nasal airflow simulations, including their correlations with objective (RMM, AR) and subjective (NOSE, VAS) measures. They proposed that the complex nature of nasal diseases might prevent CFD-based nasal airway diagnostics using single CFD-derived variables alone, and that correlations should be based on combinations of CFD-derived variables.

Kimbell and co-authors presented the first comparisons between patient-reported subjective symptoms and CFD-based flow characteristics. They found moderate correlations between subjective measures (NOSE, VAS) and CFD-based unilateral nasal resistance [116] and heat flux [61] when considering data on the side affected by surgery. They only included a few patients in their investigation, however. Later, several studies demonstrated correlations between CFD simulation results (e.g., airflow patterns, mucosal cooling, and nasal resistance) and patients’ subjective evaluation of nasal resistance [50,62,63,65,66,67]. Cherobin et al. [117] found good agreement between CFD and experimental results in a physical nasal replica. However, for their cohort consisting of 25 patients pre- and postoperatively, they found no correlation between subjective measures and RMM or CFD results.

The combined experience, that subjective and objective measures of nasal patency correlate with each other and also correlate with air flow variables obtained from CFD simulations, is a clear indication that CFD holds significant potential as a clinical decision support tool. However, there are obstacles that must be overcome. e.g., a major shortcoming in published CFD studies of nasal airflow is that CFD has not generally been able to reproduce in vivo RMM results.

Zachow et al. [118] and Hildebrandt [119] published CFD simulation data in excellent agreement with RMM data. However, it was later discovered that there were mistakes in the computations performed by an independent third party on which their conclusions were based [120]. In their later studies, CFD severely underpredicted the nasal resistance compared to RMM measurements [121]. More recently, Dong et al. [122] demonstrated perfect agreement between CFD and RMM.

Several authors have reported unexplained discrepancies between in vivo RMM measurements and in silico RMM based on CFD simulations [4,100,117,121,123,124,125,126]. Hemtiwakorn [123] reported RMM measurements to be one order of magnitude higher than CFD simulation data. Osman et al. [124] pointed out that the “bias between CFD and RMM seems to be a common problem” and “…it appears that the calculation of nasal resistance using CFD often leads to gross underestimation of nasal resistance compared to in-vivo measurements”. Berger et al. [125] reported varying degrees of agreement between in vivo and in silico RMM when comparing pressure–flow curves from five patients. They found perfect matches in Subject 5, while in Subject 4, there was gross mismatch. In Subject 3, it was observed that good agreement was achieved on one side, but not on the other side, and in Subject 1, good agreement was achieved for inhalation but not for exhalation. Cherobin et al. [117] reported that CFD underpredicted nasal resistance compared with RMM.

### 2.5. A Review of Sources of Errors and Uncertainties Affecting Comparison of In Vivo and In Silico Rhinomanometry

Although a few studies report a good match between in vivo and in silico RMM, the general impression is that CFD-based models struggle to reproduce in vivo RMM pressure–flow curves [115]. It appears that in silico studies agree better with in vitro studies in rigid nasal cavity replicas, however [117,127].

It is useful to make a distinction between the terms uncertainty and error. An uncertainty is “a potential deficiency in any phase or activity in the modelling process that is due to lack of knowledge”, whereas an error is “a recognizable deficiency in any phase or activity of modelling and simulation that is not due to lack of knowledge” [128]. It is noted that these definitions differ from typical definitions employed in experimental measurements.

Ideally, CFD simulation results can be validated against well-controlled experiments where uncertainties, e.g., regarding the flow geometry and mass and heat transfer, have been minimized, and measurement errors are under control. Under such conditions, the CFD model is subject to little uncertainty, and errors associated with poor choice of modelling strategies and submodels can readily be assessed, so that the model can be tuned to predict measured data with good accuracy. This might be the reason for the good agreement reported between in vitro and in silico RMM.

Due to difficulties and challenges associated with acquiring and assessing the quality and reproducibility of objective in vivo clinical data, including RMM and CT/MRI imaging data, there is substantial uncertainty related to the quantitative comparison of nasal resistance and pressure–flow curves obtained from in vivo and in silico RMM. e.g., an essential part of in silico RMM, not inherent in standard in vivo RMM, is the requirement of a detailed description of the nasal cavity geometry. In silico RMM typically utilizes medical imaging data to acquire the nasal cavity geometry, but unless specific actions and precautions are taken, it is unknown to what extent the medical images adequately describe the state of the nasal cavity during in vivo RMM.

Computed flow variables strongly depend on the flow geometry. Thus, the lack of knowledge about the instantaneous state of the nasal cavity during in vivo RMM causes major uncertainty regarding the quantitative comparison of nasal resistance and other flow parameters obtained from in vivo and in silico RMM. A discussion of uncertainties and errors in the comparison of in vivo and in silico RMM is therefore incomplete without a separate discussion of the relevance of medical imaging data with respect to describing the nasal cavity at the time of in vivo RMM.

Factors that may influence the comparison of in vivo and in silico RMM results are discussed briefly below, including physiological factors affecting the temporal variability of the nasal cavity geometry as well as uncertainties and errors associated with the procedures of acquisition of the digital nasal cavity geometry model and in vivo and in silico RMM.

#### 2.5.1. Physiological Factors Affecting the Temporal Variability of the Nasal Cavity Geometry

To attain accurate predictions of in vivo RMM results through in silico RMM simulations, it is imperative to employ a digital nasal cavity model that faithfully represents the dynamic state of the nasal cavity during the in vivo RMM examination.

The alignment reported between in silico RMM and in vitro RMM in physical replicas of nasal cavities [127,129,130] starkly contrasts the observed disagreement with in vivo RMM [117,125]. This suggests that CFD models are correctly configured, but somehow fail to adequately represent the nasal cavity’s actual state and function during in vivo RMM examination.

Two physiological mechanisms that can cause digital geometry models to misrepresent the nasal cavity geometry during in vivo RMM are (1) the nasal cycle, known to cause a periodic, temporal variation in nasal cavity volume, and (2) nasal cavity compliance, which may cause spontaneous expansion/contraction of the nasal cavity volume due to over-/under-pressure during respiration. In the following subsections, brief discussions are given about these two physiological phenomena. Other causes for errors and uncertainty associated with the acquisition of a digital nasal cavity geometry are discussed below.

##### Nasal Cycle

The nasal cycle causes spontaneous engorgement, hence cross-sectional variability, in the nasal cavities. Consequently, the nasal cycle affects the reproducibility of objective rhinometric measurements adversely and complicates the objective assessment of nasal patency when comparing pre- and postoperative measurements. Additionally, it poses challenges when comparing in vivo and in silico RMM results, as the nasal cavity’s shape and volume may differ between in vivo RMM examination and the acquisition of medical imaging data used for in silico RMM.

In a study by Hasegawa and Kern [74], which involved 50 subjects, bilateral and unilateral nasal resistance measurements were conducted over a 6–7 h period. They observed that the bilateral nasal resistance remained relatively constant despite cyclic variations in unilateral resistances. The average ratio of highest to lowest unilateral resistances was 4.6 on the right side and 4.4 on the left side, with peak values reaching 16.3 and 13.7, respectively. Hence, accounting for the nasal cycle is crucial when assessing nasal resistance. It is noteworthy that the nasal cycle was found to be non-reproducible in all of the five subjects who underwent re-testing, as the durations and amplitudes of their nasal cycles varied. To quantify the nature of the nasal cycle, in numerical terms, Flanagan and Eccles [75] conducted hourly unilateral airflow measurements over 8 h periods in 52 subjects.

Patel et al. [131] employed CFD to investigate the impact of the nasal cycle on objective measures. They suggested that paradoxical postoperative worsening of NAO observed in simulations could be attributed to the nasal cycle. Gaberino et al. [65] created virtual mid-cycle models to correct for the nasal cycle, resulting in improved correlation between objective and subjective measures of nasal patency. Moghaddam et al. [90] pointed out that mucosal engorgement due to the nasal cycle can significantly affect CT images, thereby influencing the correlation between CFD and subjective and objective nasal patency scores. Susaman et al. [132] emphasized that rhinologists need to take the existence of the nasal cycle, which affects a large percentage of the population, into account when examining and measuring the nose.

Several authors [33,133,134] have pointed out that postural effects on the nasal resistance and the nasal cycle should be expected. The nasal resistance tends to be higher in the supine position. Consequently, differences in posture between medical imaging procedures and RMM examinations may lead to geometrical misrepresentation in in silico RMM simulations.

To mitigate the effects of the nasal cycle, it is common practice to perform RMM both before and after applying a decongestive nasal spray that shrinks the large veins in the nasal epithelium. This approach enables the evaluation of the anatomical nasal patency [45].

The presence of the nasal cycle suggests that if CT/MRI images and RMM measurements are not acquired within a short timeframe, in the same state of decongestion, and in the same posture, they may not reflect the same nasal geometries.

##### Nasal Cavity Compliance

In a study conducted by Fodil et al. [135], a simplified model of the nasal cavity was used to demonstrate that, depending on the pathological condition, the assumption of rigid nasal cavity walls is only valid for low flowrates. The rigid wall assumption generally failed for pressure drops above 20 Pa. In contradiction, Bailie et al. [136] claimed that during resting breathing, one can regard the nasal cavity as a rigid structure. More recent research by Akmenkalne et al. [137] corroborated the earlier work of Fodil et al. [135]. They investigated the mobility of the lateral nasal wall under the influence of breathing and emphasized that even during quiet breathing, we must take into account the deflection of the nasal walls. O’Neill and Tolley [72] used a simplified mathematical model based on Bernoulli’s principle to compute the total pressure loss through the nasal cavity as a sum of minor losses. Their model allowed for the nasal gateway (valve) to dynamically adjust its cross-sectional area based on local pressure and a stiffness coefficient, providing a quantitative rationale for observed discrepancies between AR and RMM. Cherobin et al. [117] observed that while in silico RMM was in good agreement with in vitro RMM, it diverged significantly from in vivo RMM. This disparity was partly attributed to the rigid wall assumption in CFD, which matched the properties of the rigid nasal cavity replica used in in vitro experiments but might have failed to adequately represent physiological nasal cavity compliance. Schmidt et al. [121] reported systematic underprediction of nasal resistance in CFD simulations but found no significant difference between patients with or without nasal valve collapse or between inhalation and exhalation phases.

Considering that nasal cavity expansion/contraction in response to over-/under-pressure during exhalation/inhalation, has an effect on the nasal resistance, it is anticipated that the pressure–flow curves will exhibit asymmetry between these respiratory phases. To assess this, one can compare mirrored exhalation curves with inhalation curves obtained from in vivo RMM. If the two sets of curves align well, it suggests that nasal cavity compliance is minimal and cannot account for the substantial differences between in silico and in vivo RMM results. However, it is important to note that this argument does not consider the Venturi effect. When the Venturi effect dominates over the hydrostatic effect, it can lead to contraction during both inhalation and exhalation, resulting in an overall increase in nasal resistance.

This line of reasoning is consistent with the observations of Akmenkalne et al. [137], who demonstrated contraction during both inhalation and exhalation in quiet breathing. For elevated breathing and forced sniffing, the hydrostatic pressure component appeared to dominate, causing contraction during inhalation and expansion during exhalation. An interesting observation in their Figure 4 was that after a period of forceful sniffing, the deflection of the lateral nasal wall reversed its direction, transitioning from negative to positive but trending downwards. However, this reversal and slow nasal wall relaxation time did not appear to correlate with flow or pressure curves, implying minimal impact on nasal resistance.

The influence of nasal compliance on the hysteresis in RMM pressure–flow curves, illustrated in Section 2.3.3, has been discussed by Vogt and co-authors [77,138]. Wernecke et al. [77], Vogt and Zhang [138], Vogt et al. [139], Bozdemir et al. [76], and Frank-Ito and Garcia [115] presented pressure–flow curves featuring hysteresis where the portions of the curve corresponding to the accelerating and decelerating inspiratory phases were switched when compared to the simplified model showcased in Figure 2b of the present paper. Measurement results by Groß and Peters [140] support the time arrows in Figure 2b, but they attributed the observed pressure–flow curve hysteresis to the measurement technique, rather than nasal airflow dynamics. An adequate explanation for this disagreement is lacking.

It is anticipated that the influence of nasal compliance on RMM pressure–flow curves will manifest as asymmetry between inspiratory and expiratory pressure–flow curves as well as, referring to Figure 2b, a widening of the hysteresis loop. It may, however, be imagined a situation where the Venturi effect dominates over the static pressure such that local under-pressure is effectively independent of flow direction and causes (partial) collapse both during inhalation and exhalation. Owing to the phase disparity between volumetric flowrate and flow resistance, brief periods of counterintuitive over- and under-pressure may occur within the nasal cavity at the culmination of the inspiratory and expiratory phases, respectively (see Figure 2). Consequential local expansion/contraction may introduce complexity to the response of the pressure–flow curves.

#### 2.5.2. Sources of Uncertainties and Errors in the Acquisition of Digital Nasal Cavity Geometry Models

The process of acquisition and preparation of the airway geometry was described in Section 2.4.3. Two of the main steps of the process are (1) the recording of medical imaging data by CT or MRI and (2) the establishment of a surface mesh through segmentation.

It is beyond the scope of the current paper to discuss the technology behind medical imaging, but the following aspects are highlighted:CT and MRI data have relatively low spatial resolution compared to the small-scale features of the nasal cavity. This may cause inaccurate description of the small features of the nasal cavity. Cone beam CT has been proposed as an alternative due to better resolution at lower radiation dosage [141].Low temporal resolution of CT and MRI data requires the patient to hold still while data are acquired to avoid blurred images. Effects of heartbeat, breathing, and swallowing may affect image quality adversely. This suggests that CT is preferred over MRI due to better temporal resolution.Good communication with the radiologist is required to ensure that the entirety of the nasal cavity is included in the data.For comparison of pre- and postoperative airways, the patient’s posture and positioning in the CT/MRI scanner during postoperative examination should be identical to the preoperative situation. For instance, the apparent shape and volume of the pharyngeal tract may be affected by the relative tongue, jaw, head, and neck positions.The nasal cycle can be observed in medical imaging data. Medical imaging should thus be performed in the decongested state similar to decongested RMM, preferably in rapid succession after the clinical RMM procedure.

During segmentation of the imaging data, the following should be observed:The radiodensity threshold used to determine the interface between air and tissue can have a severe effect on the cross-sectional area, hence the nasal resistance. A low/high threshold will result in a narrower/more voluminous airway geometry, respectively, potentially affecting flow variables [100].Automatic segmentation methods may overlook important details or include secondary air spaces such as the paranasal sinuses, Eustachian tubes, or nasolacrimal ducts. It is recommended to confer with medical expertise such as radiology experts or surgeons to assess the resulting geometry model.

#### 2.5.3. Sources of Uncertainties and Errors in In Vivo Rhinomanometry (Clinical)

RMM systems are typically proprietary systems where it is challenging to obtain access to raw measurement data and detailed information about the post processing of the measured data. Some studies have investigated the agreement between RMM measurements performed with devices by different manufacturers [79,121] or between RMM and inhouse benchmarks [142]. There is no evidence of systematic measurement errors in RMM measurement devices, but Hoffrichter et al. [77] pointed out that some rhinometers manipulate or average the measurements, suppressing hysteresis in the measured pressure–flow curves. Silkoff et al. [143] reported a high level of reproducibility in RMM. Carney et al. [144], however, reported unacceptable variation and concluded that single RMM measurements are prone to large errors. Lack of reproducibility has also been reported by Thulesius et al. [145]. Bozdemir et al. [76] pointed out that the reproducibility of RMM measurements relies on ensuring identical conditions in subsequent measurements (e.g., air humidity and temperature, fit of the face mask, contralateral nostril closure, and avoiding oral breathing). This is best achieved by a skilled RMM operator.

Possible errors associated with improper conduction of the RMM procedure include false pressure and/or flowrate measurements due to, e.g.,:Air leakage along the edge of the face mask or contralateral nostril closure.Open mouth and oral breathing.Malfunction of the RMM equipment or post processing software.

In addition, there are uncertainties mainly associated with the geometrical/volumetric state of the nasal cavity during the RMM measurement, due to several factors:The nasal cycle may affect the unilateral nasal resistance.Posture has been shown to influence the nasal cycle [33]. Therefore, positioning of the patient may affect the RMM measurements.Compliance of the nasal walls can cause the nasal cavity to expand due to over-pressure during exhalation and contract due to under-pressure during inhalation or due to Venturi effect. This dynamic behavior may affect the nasal resistance and result in asymmetry and hysteresis in RMM pressure–flow curves. In situations where hysteresis is prominent, the inspiratory and expiratory segments of the pressure–flow curves may yield markedly distinct measurements of nasal resistance.Excessive temporal NAO due to inflammatory reactions or other causes may cause exaggerated nasal resistance that may affect RMM measurements and sometimes even prevent the patient from generating the required volumetric flowrate to conclude the RMM examination.

The uncertainties associated with acquisition of in vivo RMM data are mainly associated with the temporal variations in the geometrical state of the nasal cavity. It is therefore stressed that, for comparison between in vivo and in silico RMM, it should be ensured that medical imaging data correctly represent the nasal cavity during in vivo RMM. This can best be achieved by undertaking medical imaging examination in rapid succession of the RMM examination, in decongested state, and preferably in the same posture.

#### 2.5.4. Sources of Uncertainties and Errors in In Silico Rhinomanometry (CFD)

When presented with experimental data from flow measurements, such as in vivo or in vitro RMM, it rests upon the CFD engineer to set up a CFD model that is able to reproduce the measured data, or to explain observed discrepancies between computed and measured data. A significant preparatory task in CFD modelling is to describe the flow system both qualitatively and quantitatively with respect to geometry (including flow restrictions/walls, inlets, and exits), material properties, and other factors that may affect the flow. Even if the geometry of the flow system is well known, there is a multitude of parameters and settings that must be chosen carefully when setting up the CFD model. Potential sources of error associated with setting up and running CFD simulations have been thoroughly covered by the European Research Community on Flow, Turbulence, and Combustion [146] and many others, e.g., Andersson et al. [147], Rodriguez [94], and Roychowdhury [95]. Inthavong et al. [105] reviewed in silico approaches to simulation of nasal airflow.

Besides fundamental errors and limitations in the program code of the CFD software, such as program bugs or truncation and rounding errors, errors in simulation results can be caused by, e.g.,:Poor computational mesh quality [94].Inadequate spatial or temporal refinement.Poorly selected solver settings and numerical schemes [148].Incorrect definition of flow physics, including, e.g., boundary conditions, material properties, and approximations.Inaccurate or incorrect solution due to poor convergence and/or failure to conserve mass, momentum, or energy.

The main uncertainties are related to the lack of knowledge about the flow problem to be modelled. These can be divided into four main categories related to (1) flow physics; (2) geometry; (3) required spatial and temporal numerical resolution; and (4) boundary conditions. Even if these are implemented correctly without errors, there may be uncertainty associated with their correct description. For simulation parameters associated with high uncertainty, sensitivity analysis may be required to assess the influence of variations in these parameters.

##### Flow Physics

Uncertainties surrounding the flow physics within the nasal cavity encompass aspects that, theoretically, could be elucidated through measurements or experiments. However, practical challenges arise in conducting in vivo measurements on patients, and a lack of in vitro experimental data complicates the matter. Consequently, an ongoing debate persists regarding fundamental aspects of nasal flow physics. This includes the deciding between quasi-steady and transient modeling, determining the optimal turbulence modeling strategy, and addressing other considerations such as the dependence of air’s material and transport properties on pressure, temperature, and humidity.

(A)Modelling of temporal phenomena in respiratory flow

The physiological, respiratory flow in the nasal cavity is normally of pulsative nature. The literature review summarized in **Part II** (Section 3) suggests that steady flow modelling, by far, is the most popular approach in computational rhinology, however. It is appropriate to question the validity of the assumption of quasi-steady flow in respiratory flow modelling. e.g., how does the transient nature of the flow affect temporal effects such as hysteresis, developing flow boundary layers, and meandering of wakes or jets?

The simulation of transient flow adds complexity to CFD simulations compared to modelling steady state flow. Many authors have argued that nasal airflow can be approximated by quasi-steady flow [149,150,151,152,153,154].

In the current context, the concept of quasi-steady state implies that the time response of the overall flow phenomena within a system is much quicker than the variation in transient phenomena occurring in the system. The system’s behavior can thus be assumed to be in instantaneous equilibrium with the transient phenomena, enabling its approximation by steady state simulations. In the case of nasal airflow, quasi-steady state suggests that the nasal flow parameters can be determined from the instantaneous respiratory pressure and velocity boundary conditions, at any given moment. This implies that pressure–flow curves in in silico RMM can be generated through a series of steady state simulations conducted at different volumetric flowrates, instead of relying on a transient simulation of the entire breathing cycle.

The Womersley number, named after J. R. Womersley [155], who studied pulsatile flow in arteries, is defined as the ratio between the transient inertial and viscous forces and is commonly expressed as follows:(14)$$\mathrm{Wo}=D_{h}\sqrt{\pi f/2\nu }\;,$$
where $D_{h}$ is the channel diameter, f is the pulsation frequency, and ν is the kinematic viscosity. The Womersley number can be used to characterize an unsteady flow as quasi-steady or not [156]. The flow may be considered quasi-steady if Wo<1. Inserting for $D_{h}=5\;\mathrm{mm}$, f=0.2 Hz, and $\nu =1.5\times 10^{-5}\;m^{2}/s$, the expected Womersley number in unilateral nasal airflow is approximately Wo≈1. This is in the intermediate range, where the oscillatory nature of the flow is not dominating but may have some influence.

Doorly et al. [157] discussed whether a series of quasi-steady simulations is sufficient to characterize tidal breathing. They referred to Shi et al. [153] and suggested that the quasi-steady assumption is valid for quiet breathing. Bosykh et al. [158] showed that a transient model produced almost identical results to steady state simulations produced by themselves as well as others. Furthermore, they observed that asymmetry in the respiratory cycle had little effect on the flow pattern in the nasal cavity compared to a sinusoidal inhalation/exhalation profile, which follows naturally from quasi-steady behavior. Bradshaw et al. [159] highlighted several phenomena observed in their transient simulations that cannot be seen in steady flow. In particular, their results indicate that transient simulations of the entire breathing cycle are essential in order to correctly capture air conditioning via heating/cooling and humidification.

A noteworthy characteristic of in vivo RMM pressure–flow curves is the presence of a hysteresis pattern [77]. This hysteresis has been attributed, among other factors, to unsteady/inertial pressure drop contribution stemming from varying flowrates during respiration, and it can naturally not be predicted by steady state flow simulations. See Section 2.3.3 for more details.

(B)Modelling of turbulent, transitional, and laminar flow

The complex, dramatically varying flow channel cross-sections in the nasal cavity can have significant impact on the development of turbulent structures within the flow due to, e.g., flow separation, recirculation, varying pressure gradients, secondary flows, developing wall boundary layers, merging of separate flow streams, flow instabilities, etc. The understanding and prediction of turbulence in such scenarios typically requires very detailed CFD models and experiments. It can be expected that the behavior of such flow systems are highly non-linear and three-dimensional. e.g., Tretiakow et al. [160] found that the flow in the ostiomeatal complex (e.g., degree of turbulence) depended on the overall geometric features of the nasal cavity (e.g., nasal septum deviation).

Only DNS is an exact representation of the Navier–Stokes equations. All other turbulence models contain approximations with individual limitations and ranges of validity. e.g., while RANS models are ensemble averaged and unable to model individual turbulent eddies, LES models are able to track eddies larger than a given filter size (typically a function of the computational mesh size) and employ subgrid models to describe the effect of smaller eddies. In principle, LES should approach DNS in the limit of small filter sizes.

While RANS-based models are much cheaper than LES- or DNS-based models, in terms of computational power requirements, they are known to have many limitations. These models were typically created to solve specialized industrial problems with a good balance between accuracy and computational cost. Model parameters were thus tuned to predict standard, industrial flow scenarios. It is not given that these models are suitable for modelling of flow in complex geometries such as the nasal cavity. Moreover, these models are known to have severe limitations with respect to modelling transitional flow. Thus, if the nasal flow is transitioning between laminar and turbulent flow along the length of the nasal cavity and due to the respiratory variation in flow velocity, these models might not be able to predict the flow accurately.

Most authors discussing turbulence in the upper airways seem to consider the Reynolds number only as a criterium for the onset of turbulence. They fail to consider that it takes time to develop turbulence. In pipe flow, at Reynolds numbers above the critical Reynolds number, it generally takes more than 10 pipe diameters’ flow length to fully develop the turbulent velocity profile. During restful tidal breathing, approximately 25 nasal volumes are inhaled/exhaled during one cycle [161] and considering that the length of the nasal passage is between five and fifteen times its hydraulic diameter, the flow field is thus unlikely to be fully developed. Even for steady flow, it seems unlikely that the flow can be fully developed due to the varying cross-sectional area along the nasal cavity, and the many anatomical features that affect the flow pattern. There may, however, be regions within the nasal cavity that experience periods of transitional/turbulent flow during a respiratory cycle. It can be expected that most ensemble-averaged turbulence models are unsuited for such complex spatially and temporally varying laminar/transitional/turbulent flow fields.

For flow channels with cross-sections that slightly deviate from a cylindrical shape, the hydraulic diameter has proven useful in predicting flow resistance using standard friction loss correlations, such as the Haaland correlation (Equation (7)). For cross-sectional shapes deviating from cylindrical shapes, inaccuracies have been reported [73]. This indicates that Reynolds numbers based on hydraulic diameter of complex cross-sections such as those in the nasal cavity may not be appropriate for predicting the transition from laminar to turbulent flow.

Transition between laminar and turbulent unsteady flow is still, despite its importance in many engineering applications, not fully understood [162]. Recently, Guerrero et al. reviewed the literature and performed DNS to investigate the transient behavior of accelerating [163] and decelerating [164] turbulent pipe flows. An early discussion of the transition between laminar and turbulent flow in pulsatile flow in the cardiovascular system was published by Yellin [165], who observed that large instantaneous Reynolds numbers did not result in transition to turbulence everywhere. Gündoğdu and Çarpinlioğlu [166,167] presented the theoretical background for pulsatile laminar, transitional, and turbulent flows, and reviewed theoretical and experimental investigations. Xu et al. [168] investigated the effect of pulsation on transition from laminar to turbulent flow for rigid, straight pipes. They observed that the delay in the transition to turbulence increases with decreasing Womersley number (Wo<12) and increasing pulsation amplitude. The implication is that the turbulent transition threshold for Womersley numbers close to 1 is above Reynolds number 3000, in straight pipes, but turbulent puffs may exist due to the high Reynolds number time intervals of a respiratory cycle.

The combined effect of delayed transition and relatively short flow channel suggests that fully developed turbulent flow within the nasal cavity is improbable during resting respiratory flow. When utilizing RANS turbulence models that assume fully developed turbulent flow in cases where the flow is laminar, transitional, or developing, there is a risk of overestimating turbulent viscosity and, consequently, overall flow resistance. If the validation of in silico models relies solely on nasal resistance measured through in vivo RMM, there is a potential bias towards models that overestimate turbulent viscosity. This bias may help align in silico and in vivo RMM pressure–flow curves but could lead to an incomplete representation of other pertinent phenomena. This example illustrates the perils of overly simplistic analyses in the study of complex problems like nasal airflow and emphasizes the necessity for additional objective, measurable metrics in the assessment of nasal airflow.

Schillaci and Quadrio [148] compared laminar/RANS/LES simulations and concluded that the choice of numerical scheme is more important than the choice of turbulence model, although they emphasized that the chosen turbulence model should be able to handle three-dimensional, vortical, mostly laminar flow conditions. They suggested that LES or DNS is necessary to reliably simulate the full breathing cycle at intermediate intensity. Bradshaw et al. [159] performed hybrid RANS-LES simulations of the entire respiratory cycle and reported bilateral nasal airflow to be dominantly laminar. While LES is widely acknowledged as one of the most accurate turbulence modelling approaches, second only to DNS, it is worth noting that LES also encounters challenges in predicting transitional flow and the initiation of turbulence, as highlighted by Sayadi and Moin [169].

(C)Other aspects

Other aspects of minor importance are just mentioned briefly, for completeness.

Temperature and humidity may affect the material properties of air. Some authors have suggested that these effects should be taken into account [76]. Other authors have dismissed these effects [151]. In the relevant temperature range, the mass density and viscosity of air varies by less than ten percent, and the effect of humidity is of the same order. For most situations, it is thus expected that this is of minor importance.Due to the small effect of pressure on air material properties within the relevant pressure range, and low flow velocities, it is safe to assume atmospheric ambient pressure and constant air material properties.

##### Geometry

The nasal cavity is a highly complex flow channel, with cross-sections that change dramatically in shape and area throughout the nose, and generally deviate significantly from cylindrical shape. Moreover, the nasal cavity is bounded by walls covered in mucosal lining, which may add a transient geometrical variation to the air–tissue interface, as well as constituting a non-rigid structure. The acquisition of a three-dimensional digital nasal cavity geometry model is prone to errors and uncertainties, as discussed in Section 2.5.2. In addition, there are uncertainties regarding the geometrical level of detail required to set up accurate CFD models. For the CFD model to be able to accurately predict the behavior of physical phenomena, it is essential that the geometry model does not misrepresent the actual flow geometry too much. When manufacturing the nasal cavity model, essential questions that should be considered include the following:How much of the surrounding volume outside the nose and in the oropharyngeal tract should be included? This consideration will affect to what extent the boundary conditions will affect the simulated flow fields. This question is closely intertwined with the discussion about boundary conditions, below.How much of the paranasal sinuses should be included? CFD simulation of the flow in the maxillary sinus was performed by Zang et al. [170]. Their conclusion was that the airflow inside the maxillary sinus was much lower (<5%) than the airflow in the nasal cavity. Due to the narrow passage connecting the paranasal sinuses with the nasal cavity, it is expected that negligible gas exchange takes place between the two [171]. This was supported by simulation results presented by Bradshaw et al. [159]. Kaneda et al. [126] reported that the inclusion of the paranasal sinuses did not improve the disagreement between computed and measured nasal resistances.What is the role of the oral cavity? Paz et al. [172] investigated the distribution between nasal and oral breathing under steady and unsteady flow. Chen et al. [173] concluded that the inclusion of the oral cavity in CFD simulations of steady and unsteady nasal cavity flow had very little impact. Open mouth and oral breathing may, however, affect the RMM pressure–flow curve.Should minor geometrical features such as nasal hair or the mucosal lining be considered?
⚬Hahn et al. [151] found that the inclusion of nasal hair increased turbulent intensity in the external nares during inspiratory flow but had little effect on downstream velocity profiles. Stoddard et al. [174] found that a reduction in nasal hair density had a positive impact on both subjective and objective measures of nasal obstruction, however.⚬Lee et al. [175] illustrated how the mucous layer may affect local flow velocities in the nasal cavity.

Uncertainty associated with the geometrical (mis-)representation of the nasal cavity in CFD models may be due to unknown factors affecting the process of manufacturing three-dimensional geometries from medical imaging data, via segmentation, or uncertainty regarding how well the CT/MRI imaging data represent the actual nasal cavity geometry during the RMM procedure.

##### Required Spatial and Temporal Numerical Resolution

Spatial and temporal discretization is required in order to enable the numerical solution of the governing equations of CFD (Equations (9)–(11)). The rule of thumb is that the spatial and temporal resolution must be sufficient to resolve all spatial and temporal flow features of interest. In addition to determining the accuracy of CFD simulations, mesh and time step size may affect numerical stability and robustness of CFD solvers.

To assess the numerical solution’s sensitivity to grid refinement, grid dependency tests should be performed. If the computed flow fields change negligibly by increasing the grid resolution, it is commonly assumed that grid independence is achieved. Frank-Ito et al. [176] reviewed the literature and investigated the requirements for grid independence in their own steady, inspiratory, laminar sinonasal cavity airflow with particle deposition. They emphasized the importance of mesh refinement analysis to obtain trustworthy computational solutions. Similarly, to assess transient solutions’ sensitivity to time step size, comparison between simulations employing relatively short and long time steps should be performed.

Brief discussions of how the spatial and temporal resolution can introduce uncertainties in CFD simulations of nasal airflow are given below.

(A)Spatial Resolution

The spatial resolution of the computational mesh determines the level of detail in the computed flow fields. e.g., in the presence of steep velocity gradients, refined meshes are needed to avoid numerical diffusion. Moreover, turbulent eddies smaller than the mesh size must be modelled by closure laws and subgrid models, which introduce approximations.

Particularly, to describe flow profiles accurately in the vicinity of walls, the near-wall mesh must honor requirements by the turbulence model employed. The theory behind this is described in classical text books on turbulence by, e.g., Tennekes and Lumley [108], Pope [109], or Wilcox [110], and in CFD simulation software user and theory guides.

Distance to the wall is commonly expressed in wall units, where the dimensionless wall distance is expressed as
(15)$$y^{+}=u_{\tau }y/\nu \;,$$
where $u_{\tau }=\sqrt{\tau_{w}/\rho }$ is the shear velocity, $\tau_{w}$ is the wall shear stress, and y is the distance to the wall. The Law of the wall states that the dimensionless velocity parallel to the wall is given by
(16)$$u^{+}=u/u_{\tau }=\left\{\begin{array}{lll} y^{+} & \mathrm{for} & y^{+}≲5\;(\mathrm{viscous}\;\mathrm{sublayer})\\ \frac{1}{0.41}\ln y^{+}+5 & \mathrm{for} & y^{+}≳35\;(\log\;\mathrm{layer}) \end{array}\right.\;,$$
where u denotes the flow velocity parallel to the wall. The intermediate range between the viscous sublayer and the log layer is known as the buffer layer. The original idea by Launder and Spalding [177] was to use Equation (14) as a wall function for the velocity boundary conditions at the wall. This approach, which is used in some classic RANS turbulence models, such as the kε type turbulence models, requires that the centroids of computational grid cells residing at the wall are in the log layer ($y^{+}≳50$). Other RANS turbulence model types, such as the kω and kω SST models and kε with enhanced wall treatment, require (or permit) that the near-wall grid cells are within the viscous sublayer ($y^{+}≲5$). While near wall grid cells in the buffer layer may be handled with blending functions, they are a major source of misrepresentation of wall shear stresses and should be avoided. LES and DNS approaches generally require $y^{+}<1$. Near-wall grid cells outside the appropriate $y^{+}$ range may result in incorrect turbulence production and turbulent viscosity, hence the flow resistance.

Due to the complex geometrical nature of the nasal cavities, it is expected that boundary layer thicknesses will experience significant spatial and temporal variations due to the breathing cycle. The best option might therefore be to ensure that the computational mesh is fine enough to maintain near wall cells in the viscous sublayer, everywhere, for all flowrates, and to utilize a suitable turbulence model. Inthavong et al. [106] discussed mesh resolution requirements for laminar nasal air flow and suggested that $y^{+}<0.27$ in the near-wall grid cells.

Outside the near-wall region, the spatial resolution must be sufficient to resolve all relevant flow structures. It has been suggested that 4–6 million grid cells is generally sufficient to achieve mesh independence [85,106,176], but this is a generalization that should be accepted with caution, since it might not be appropriate for all nasal geometries and volumetric flowrates.

Adaptive meshing, which is a technique that regenerates and/or adapts the mesh based on predefined flow field criteria, is an approach that may be well suited to respiratory breathing, where a wide range of flow characteristics and features can be expected, and a fixed mesh might not be the best choice for the entire range of volumetric flowrates. This technique allows for the refinement of the mesh in regions of steep gradients or small flow features as well as coarsening of the mesh where spatial variations are modest. This further allows for a non-constant number of grid cells, reducing computational cost and giving shorter computation times when the flowrates are lower. It does come at the computational cost of remeshing/adjusting the computational mesh, however. Adaptive meshing is not exclusive to transient simulations but may also be used to improve accuracy in steady state simulations. The present author is not aware of any studies investigating this for nasal airflow.

(B)Temporal Resolution

The temporal resolution determines simulations’ ability to correctly describe transient variations in the flow fields. e.g., long time steps might not be able to capture quickly fluctuating phenomena. The time step size is commonly characterized by the dimensionless Courant–Friedrich–Lewy (CFL) number [178,179], which expresses the ratio of the advected distance during one time step, uΔt, to the characteristic grid size, Δx,
(17)CFL=uΔt/Δx.

Here, u denotes the flow velocity through the grid cell, and Δt is the time step size.

The CFL number plays a critical role in ensuring the numerical stability and accuracy of CFD solvers. Explicit CFD solvers restrict information propagation to the maximum of one grid cell per time step (CFL≤1). Consequently, this limitation forces the use of exceedingly short time steps when dealing with small geometry features resolved by small grid cells, resulting in prolonged and costly simulations. Implicit solvers, on the other hand, permit longer time steps, but incorrect simulation results can be the result for too large CFL numbers.

On a fixed computational mesh, with a fixed time step the CFL number tends to zero at the culmination of the inspiratory and expiratory phases of the respiratory cycle. Depending on the numerical scheme employed, short time steps can introduce numerical diffusion, blurring the details of the flow, but the main downside is an unnecessarily high number of time steps. To mitigate this issue, adaptive time stepping strategies can be employed, where the time step size increases as the volumetric flowrate decreases. This approach helps maintain favorable CFL numbers and reduces computational cost. However, it is important to note that since the volumetric flowrate eventually dwindles to zero, a numerical scheme capable of handling low CFL numbers remains essential.

##### Boundary Conditions

Setting appropriate boundary conditions is a crucial step in configuring CFD models. Accurately describing flow parameters such as velocity, turbulence intensity, pressure, and temperature at the boundaries often presents a challenging task, leaving CFD engineers to rely on best available estimates or educated guesses. Consequently, it is important to position the boundaries at sufficient distance from the region of interest to prevent undue interference with the essential details of the flow. However, expanding the simulation domain to achieve this boundary distance unavoidably incurs higher computational costs. Thus, the determination of boundaries’ locations necessitates careful balance between accuracy and cost efficiency.

In the specific context of nasal airflow analysis, the boundaries include the walls of the nasal cavity and the flow in- and outlets.

The walls are typically treated as smooth non-slip boundaries. Nevertheless, the presence of the mucosal lining introduces the possibility that surface roughness and slip conditions might need consideration. While the nasal wall temperature is commonly assumed to fall within the range of normal body core temperature, this assumption may require more careful consideration if the inhaled air is significantly colder.The nostrils serve as inlets to the nasal cavity during inhalation and outlets during exhalation. However, it is reasonable to suspect that truncating the computational domain at the nostrils may compromise the accurate description of airflow entering or exiting the nasal cavity. An alternative approach is to extend the computational domain to encompass the external airspace around the nose to achieve a more realistic airflow distribution at the nostrils. A study by Taylor et al. [180] suggested that the qualitative description of the inflow conditions at the nares may not be critical when computing general flow patterns and overall measures, but for detailed regional flow patterns, carefully chosen inflow conditions may be necessary.Modeling the entire airway, including the lungs and alveoli, is impractical in nasal airflow studies. Therefore, the airway is typically truncated somewhere in the laryngopharyngeal tract. The location of this truncation has traditionally been based on available computational resources and the specific phenomena of interest. Although the location of truncation may be less critical during inhalation, more attention may be warranted during exhalation. Wu et al. [181] demonstrated, in a physical experiment, that the flow in the pharynx is laminar during normal breathing, but Bradshaw et al. [159] highlighted the importance of including a realistic pharyngeal tract to achieve accurate flow conditions in the nasopharynx during exhalation. The pharyngeal tract is a complex, soft-tissue-enclosed flow channel susceptible to head and neck movements, swallowing, tongue movement, and compliance with over-/under-pressure due to breathing. The exhalatory flow pattern entering the nasopharynx is likely to be affected by this. The level of realism required in the pharyngeal tract to attain acceptable inflow to the nasopharynx is still unresolved.

Once the boundary locations are determined, careful selection of boundary types (e.g., Dirichlet or Neumann) is required to ensure uniqueness of solution before defining boundary values. These boundary values can be constant or vary with time and/or position along the boundaries. There is generally significant uncertainty associated with the determination of the local boundary values, necessitating sensitivity analysis to evaluate the impact of boundary conditions.

### 2.6. Summary of Part I

Rhinology, a specialized branch of otorhinolaryngology, is dedicated to advancing diagnostics and treatment methods for nasal and sinonasal disorders, including conditions like nasal airway obstruction (NAO).The discord between objective and subjective clinical assessments of NAO severity has created an opening for mathematical modeling tools, such as computational fluid dynamics (CFD), to enhance our understanding of nasal function.Computational rhinology, a subfield of biomechanics, employs numerical simulations, like CFD, to gain deeper insights into nasal and sinus function and pathology.Computational rhinology is poised to exert a substantial influence on clinical medicine by offering objective, simulation-based decision support for tailoring patient-specific treatment options within the realm of otorhinolaryngology. Furthermore, it may facilitate research and development of novel or improved treatment methods, as well as comparisons between patient-specific and cohort studies.While substantial progress has been made over the past three decades toward the clinical application of CFD, there is a lack of robust evidence supporting its applicability and value, and it is yet to attain widespread acceptance as a viable clinical decision support tool.Despite significant collaborative efforts from experts in both rhinology and CFD over several decades, CFD is not able to reproduce results from objective clinical measurements, such as rhinomanometry (RMM). In particular, in silico RMM consistently underpredicts nasal resistance compared to in vivo RMM.A comprehensive overview of sources of error and uncertainty affecting the comparison of in vivo and in silico RMM has been presented. The observed discrepancies may be the result of a combination of multiple independent factors, rather than a single, isolated cause. Major sources of uncertainty include the following:
⚬Comparability of nasal cavity geometry during RMM and medical imaging examinations.⚬The impact of nasal compliance.⚬CFD modelling strategies (e.g., turbulence modelling, unsteady/steady flow).Regardless of RMM’s capability to predict a patient’s subjective sensation of nasal patency, it serves as one of few opportunities for validating in silico nasal airflow models. Consequently, RMM plays an indispensable role in the field of computational rhinology.The lacking agreement between in vivo and in silico RMM results is a fundamental problem that must be addressed for CFD to gain recognition as a reliable, objective clinical decision support tool.

## 3. Part II—Overview of Published Literature on In Vitro and In Silico Nasal Airflow Studies

In the field of rhinology, CFD is considered an emerging technology with significant potential. Three decades ago, Keyhani et al. [152] published the first anatomically accurate CFD model of nasal airflow, marking the beginning of advancements in the workflow from medical imaging (such as CT or MRI) to CFD analysis. Since then, several reports have highlighted successful applications of CFD in areas such as surgical intervention planning and improved understanding of nasal airflow. However, despite these advancements, CFD has not yet gained widespread use in clinical practice. One of the main obstacles is the lack of consensus among otolaryngologists regarding objective evaluation criteria for assessing nasal function and guiding surgical decisions [90,182]. This poses a hurdle for utilizing CFD as a clinical decision support tool, because CFD relies on clearly defined questions and produces mainly quantitative results. Despite groundbreaking scientific progress and increasing interest within the otorhinolaryngology community [183], CFD technology is still considered immature. This is evident from the fact that recent reviews of diagnostic tools in rhinology [46] and of recent advances in surgical treatments for obstructive sleep apnea [184] did not even mention CFD. A specific concern raised by Vicory et al. [185] is the labor-intensive nature of manual segmentation in CFD-based virtual surgery, which adds complexity and time requirements to the process. However, there is optimism for future improvements in this area. It is anticipated that segmentation methods will undergo advancements, leveraging novel techniques in machine learning. These advancements have the potential to optimize and automate the workflow from medical images to CFD analysis, streamlining the process for greater efficiency. Wong et al. [20] argue that the maturity of CFD in rhinology is approaching the level where its successful implementation in clinical practice is feasible. They suggested that CFD should be recognized as a valuable diagnostic tool within rhinology. However, despite tremendous efforts in realizing clinical relevance of CFD, there is still no proof that CFD-based clinical decision support will actually improve patient outcomes [115].

Bailie et al.’s overview of numerical modelling of nasal airflow [136], aimed at the otolaryngology community, continues to hold relevance today. However, it is important to keep in mind the significant scientific and engineering advances that have taken place in every stage of the process from generating realistic 3D computer models from medical imaging data to conducting multiphysics simulations and analysis involving, e.g., turbulence and soft tissue movement. These breakthroughs have been accompanied by notable improvements in computational power, facilitating the use of high-fidelity computational meshes with tens of millions of grid cells, while maintaining relatively short computation times. In a more recent publication, Lintermann [186] addresses the general application of CFD in rhinology. While it did not cite the most recent scientific publications in the field, it serves as an accessible introduction to researchers who are new to CFD, providing a valuable starting point for further exploration. Tu et al. [187] published the first textbook presenting a comprehensive overview of the possibilities of utilizing engineering CFD to enhance the medical understanding of respiratory airflow and particle transport. A more recent text book edited by Inthavong et al. [188] “explores computational fluid dynamics in the context of the human nose, …” and “focuses on advanced research topics, such as virtual surgery, AI-assisted clinical applications and therapy, as well as the latest computational modeling techniques, controversies, challenges and future directions in simulation using CFD software”.

In the past fifteen years, the accessibility of generating patient-specific 3D geometry models using CT imaging techniques or similar methods has significantly increased [18,96]. This progress, driven by advancements in automatic segmentation routines, has greatly facilitated the implementation of CFD simulations for studying airflow in the human upper airways by fluid dynamics researchers worldwide. Additionally, simplified CFD simulation setups have made it possible for non-specialists to configure and execute reasonable simulations, even without extensive training in the field. These trends are reflected in the rapidly growing number of publications discussing various topics relevant to otolaryngology, including targeted drug delivery via nasal spray, OSA, NAO, virtual surgery, and more. However, a notable limitation observed in many of these publications is the lack of clinical grounding. For example, numerous publications present simulation results without adequately interpreting their clinical value or applicability. Additionally, some publications overlook the inclusion of all relevant parts of the airway or fail to exclude irrelevant anatomical features. These shortcomings could have been addressed through closer collaboration with clinical personnel to ensure a more comprehensive approach.

In a review paper by Inthavong et al. [105], it was noted that the annual publication count of in silico studies on nasal flows exhibited nearly exponential growth from 1993 to 2018. In 2018 alone, close to eighty articles were published on this subject. The rapidly increasing number of publications has led to a host of review papers being published over the last two decades. Baile et al. [136], Zhao and Dalton [18], and Leong et al. [189] reviewed the earliest literature on the topic and discussed the implications and prospects of mathematical modelling of nasal airflow. Zhao and Dalton pointed out that CFD modelling can have important implications for (1) predicting how inflammation or anatomy can affect airflow patterns and olfaction; (2) optimization of treatment; and (3) prediction of particle transport (hereunder both pollutants and medical drugs). Kim et al. [171] reviewed studies that employed physical models as well as studies focusing on numerical modelling. They gave an overview of available methods and challenges associated with the employment of CFD in patient-specific diagnostics and analysis in rhinology and underlined that experienced otolaryngologists should be involved in quality assurance of 3D airway models. They proceeded to discuss how the nasal cycle may affect the evaluation of nasal patency and reviewed how inaccuracies stemming from various segmentation methods and 3D modelling methods may affect the CFD geometry. A recent review aimed at rhinologists [19] highlighted the potential applicability of CFD in clinical applications as a diagnostic tool. Radulesco et al. [190] gave a literature review focusing on the usefulness of CFD in the assessment of NAO. Other recent literature reviews were given by, e.g., Faizal et al. [191] and Ayodele et al. [99]. Recent publications considering patient-specific mathematical modelling and simulation for clinical decision support include, e.g., [89,90,192,193,194,195]. Interestingly, many research questions raised by Quadrio et al. [196] a decade ago regarding nasal airflow, such as the selection of turbulence models and boundary conditions, as well as the validity of assuming rigid walls, remain unresolved.

The bibliography in the current paper was compiled through a combination of ancestry and descendancy literature review approaches, utilizing internet-based publication databases such as Google Scholar, Researchgate, PubMed, and journal web pages. In particular, recent review papers and textbooks were used as basis for discovering relevant publications. Because this paper focuses on the nasal cavity, publications that did not include the nasal cavity were disregarded. Also, the current paper does not review the extensive scientific literature concerning FSI, acoustic phenomena, or particle transport and deposition, in human airways. Recent literature overviews on these topics were provided by Ashraf et al. [107] and Le et al. [197] for FSI, Xi et al. [198] for acoustics, and Larimi et al. [199] for particle deposition.

Below, a tabulated overview of literature concerned with various topics relevant to CFD modelling in rhinology is given, based on a combination of ancestry and descendancy literature review approaches. In Table 1, cited papers are classified and grouped depending on their approach to physical or numerical modelling of nasal airflow, e.g., steady vs. transient volumetric flowrate and type of modelling approach.

### 3.1. In Vitro Studies in Physical Nasal Replicas

Prior to the advent of CFD capabilities enabling accurate nasal airflow simulations, researchers relied on flow experiments conducted in physical replicas to improve their understanding of nasal airflow distribution during respiration. Pioneering experimental investigations were carried out by Proetz [149], Stuiver [200], Masing [201], Hornung et al. [202], and Hahn et al. [151]. More recently, Kelly et al. [203] utilized particle image velocimetry (PIV) to examine airflow patterns in a physical replica of a nasal cavity and provided an overview of previous experimental work in this domain. Several studies have reported favorable agreement between flow fields obtained from CFD modeling and measurements in physical models [157,204]. Notable recent experimental studies have been carried out by Le et al. [197], Ormiskangas et al. [205], Berger et al. [42], van Strien et al. [206], and Reid et al. [130]. Experimental work continues to play a crucial role in the validation of CFD models.

### 3.2. In Silico Cohort Studies

A few studies have made an effort to compare and evaluate CFD results for a cohort of multiple individuals. Zhao and co-authors [207] made an attempt to classify what constitutes normal nasal airflow in a cohort of 22 healthy subjects. Ramprasad and Frank-Ito [208] used CFD to study the role of three nasal vestibule phenotypes found in their cohort consisting of 16 subjects, on nasal physiology. Gaberino et al. [65] used virtually created nasal cycle mid-point CFD models to study 12 patients. Sanmiguel-Rojas et al. [209] proposed two non-dimensional estimators representing geometrical features and nasal resistance and used CFD simulations to investigate how these estimators varied in 24 healthy and 25 deceased subjects. Radulesco et al. [66] used CFD to evaluate NAO due to septal deviation in 22 patients. Borojeni et al. [67] used CFD to establish normative ranges for nasal air flow variables in a cohort consisting of 47 healthy adults. Li et al. [210] used CFD to study 30 patients with obstructive sleep apnea and NAO pre- and postoperatively to find the effect of nasal surgery on the airflow in the nasal and palate pharyngeal cavities. Ormiskangas et al. [211] compared CFD-based wall shear stress and heat flux results to subjective evaluation of NAO pre- and post-inferior turbinate surgery for 25 patients.

### 3.3. Modelling Approaches in In Silico Studies

It is expected that CFD can provide objective decision support in treatment of breathing-related disorders. However, CFD requires experimental validation to gain full confidence due to the many tuning parameters in the various submodels and solvers involved. Several authors have experienced that CFD results generally deviate from in vivo measurements but display better agreement with in vitro studies (see **Part I**, Section 2.4.6). So far, the reasons for this are unclear. A review of sources of errors and uncertainties affecting comparison between in vivo and in silico RMM was presented in **Part I** (Section 2.5).

There is an ongoing debate about the nature of the flow in the human upper airways in the scientific literature [83,84,85]. One of the main concerns has been whether the flow is laminar, transitional, or turbulent. A wide variety of turbulence models exist, and many of them have been employed in in silico nasal airflow studies, including laminar, RANS, LES, and DNS modelling concepts.

The selection of a turbulence model should generally be based on a good understanding of the flow phenomena that is to be modelled. e.g., the many different RANS models may only differ by small nuances, but they are all tailored to specific calibration cases. It is somewhat naïve to expect a RANS model developed for fully developed pipe flow to perform accurately in a very different flow situation such as unsteady non-equilibrium flow in a complex flow channel such as the nasal cavity. CFD modelling of transitional flow is considered one of the most difficult tasks, and only a few turbulence models are able to handle this with reasonable accuracy.

There is no consensus regarding the most appropriate modelling technique, but the current review indicates that steady state laminar modelling has been the predominant modelling strategy. A rule of thumb adopted by many authors over the last three decades, when faced with the difficult choice of which turbulence model to select in CFD modelling of nasal airflow, is that unilateral nasal airflow is laminar below 15–20 L/min and turbulent above 20 L/min. This generalization has become so common that the original sources are not evident, and some authors do not even reference a source [105,106,212,213]. It appears that the original references of this assumption are the early in vitro unilateral nasal airflow studies by Swift and Proctor [214], Schreck et al. [215], and Hahn et al. [151]. Swift and Proctor [214] observed laminar flow at 125 mL/s and turbulence at 208 mL/s; Schreck et al. [215] observed turbulence at flowrates corresponding to 200 mL/s; and Hahn et al. [151] observed laminar flow at 180 mL/s and turbulence at 560 and 1100 mL/s. Numerical simulations by Li et al. [83] demonstrated that a laminar model achieved good agreement with LES and DNS simulations up to a unilateral flowrate of 180 mL/s. Simmen et al. [216] observed partial turbulence generated in the nasal gateway (valve) region even at low velocities, however, and it appears natural, from an evolutionary perspective, to promote turbulence in the nasal cavities, since this will aid in fulfilling the main functionalities of the nose (heat exchange, humidification, and dust particle deposition).

Aasgrav et al. [3] showed that laminar and RANS CFD models produced similar results. The conclusion has been supported by many researchers including, e.g., Larimi et al. [199], who reported negligible turbulence intensity in the nasal cavity for normal, inspiratory breathing rates, and Schillaci and Quadrio [148] and Bradshaw et al. [159], who reported mostly laminar flow for normal resting breathing conditions.

Turbulence intensity has been reported to be low in the nasal cavity, but the ability to model vortexes has been promoted as an important feature of nasal air flow models, e.g., to predict particle deposition, heat exchange, and humidification correctly [217]. Li et al. [83] reported from a validation study including laminar, RANS, LES, and DNS models, where they concluded that DNS generally reproduced experimental data best, but LES and laminar models were also accurate, depending on the flowrate. They emphasize the need to select the turbulence model carefully, to be able to compute flow quantities of interest accurately. Berger, Pillei et al. [42] compared Lattice–Boltzmann (LB)-, LES-, and Navier–Stokes-based RANS models to experiments in a physical replica of a nasal cavity and reported that the different techniques “agree with some caveats”. Moreover, they demonstrated that LB LES was computationally very cheap and more accurate than Navier–Stokes based RANS. In **Part III** (Section 4) of the present paper, laminar and RANS steady state models utilizing relatively coarse computational meshes are compared to fully resolved transient LES simulations. The results indicate that the flow is near-laminar at all flowrates for the current patient.

Another dispute is related to whether respiratory flow can be approximated by quasi-steady models or not [158,159,218]. The quasi-steady approach assumes that transient, tidal breathing can be approximated by a series of steady state simulations and is typically much cheaper, in terms of computational cost, than transient simulations. Steady state simulations may, however, overlook important transient physical phenomena such as flow instabilities due to unsteady vortices or jets, development of flow structures, and heating and humidification [159]. Many authors have argued that normal, tidal breathing is quasi-steady [149,150,151,152,153,154] (see **Part I**, Section 2.5.4). **Part III** (Section 4) of the present paper concludes that transient effects are negligible with respect to in silico RMM modelling, based on the observation that steady state models performed almost identical to a transient model.

In Table 1, a non-exhaustive overview of in vitro and in silico nasal airflow studies is presented. Each cited paper is classified by its modelling strategy. The left and right columns refer to papers employing steady or transient volumetric flowrates, respectively. Row-wise, a distinction is made between in vitro flow experiments, in physical replicas of the nasal cavity, and in silico CFD studies of nasal flow. In silico studies are further grouped according to their modelling approach, Navier–Stokes- or Lattice–Boltzmann-based, as well as the turbulence modelling approach employed (laminar, RANS, LES, DNS).

**Table 1 bioengineering-11-00239-t001:** A non-exhaustive overview of experimental and numerical studies of human nasal airflow employing steady and transient flow boundary conditions.

			Volumetric Flowrate	
			Steady	Transient
Experiments in physical replicas (in vitro)			[42,83,117,151,173,203,204,205,214,215,216,219,220,221]	[42,127,129,130,149,151,157,197,200,201,202,203,204,206,222,223,224]
Navier–Stokes-based models (in silico)	Laminar		[5,50,57,58,59,61,62,63,65,68,83,86,88,90,100,101,106,116,117,123,126,129,131,148,152,158,172,175,176,180,192,199,205,207,208,209,211,219,225,226,227,228,229,230,231,232,233,234,235,236] *The present study*	[126,129,153,158,172]
	RANS	kε	[3,4,42,83,170,204,210,220,226,230,237,238,239] *The present study*	[224]
		kω	[83,117,121,122,173,204,205,207,226,229,237,240,241,242]	[173,197]
		kω SST	[83,101,124,148,172,175,199,204,205,212,213,220,243,244,245] *The present study*	[118,160,172,194,223]
		Spalart–Allmaras	[204,226,246]	
		Reynolds stress model	[83]	
	LES and RANS-LES hybrid models		[83,148,204,206,217,237,247]	[159,248,249,250] *The present study*
	DNS		[83]	
Lattice–Boltzmann-based methods (in silico)			[42,125,195,251,252,253]	[127]

### 3.4. Summary of Part II

An increasing number of scientific publications are being published in the cross-disciplinary field of computational rhinology. As of 2018, the publication rate was approximately 80 papers per year after several years of near-exponential increase.The rapid increase in the publication rate is mainly attributed to the advancements made in automatic segmentation of medical imaging data, streamlining the process of 3D geometry generation, and automatic, unstructured meshing of complex geometries.There is no consensus regarding the choice of turbulence model in nasal airflow simulations. Laminar flow modelling appears to be the most popular approach, by far, followed by RANS models. Few publications have reported from LES or DNS modelling.Most studies have investigated steady, inspiratory flow. Relatively few publications have reported from expiratory or transient/respiratory flow.

## 4. Part III—In Silico RMM Simulation Results

Computational fluid dynamics (CFD) is a thoroughly proven methodology/technology widely used in engineering and scientific research to study, design, and optimize complex fluid systems. Over the last fifty years, this methodology has warranted technological leaps, which we take for granted in the modern society, within a wide range of application such as weather forecasting and climate research, automotive and aerospace industries, the energy sector, industrial design, and many others.

CFD simulations have contributed to improving the general understanding of nasal function and have been proposed as a promising tool for enhancing diagnostics and treatment planning in otorhinolaryngology. A severe limitation to the employment of CFD as an objective clinical tool is, however, the reported lack of agreement between in vivo measurements such as rhinomanometry (RMM) and CFD (see **Part I**, Section 2.4.6).

In **Part I** (Section 2) of the current paper, the foundations for and challenges within the emerging scientific field of computational rhinology were elaborated. Sources of errors and uncertainties that may affect the comparison of in vivo and in silico RMM were discussed. In **Part II** (Section 3), a literature review of CFD studies involving nasal airflow was provided. This part of the paper showcases previously unpublished high-fidelity patient-specific CFD simulations of nasal airflow in Active Anterior Rhinomanometry (AAR).

The aim of this study was to scrutinize selected possible reasons for the observed discrepancies between in vivo and in silico RMM (refer also to **Part I**, Section 2.5.4). Specifically, this study investigated the threefold hypothesis that these discrepancies can be explained by (1) *poor choice of turbulence model*, (2) *insufficient spatial or temporal resolution*, or (3) *neglecting transient effects*, in CFD models. To do this, finely resolved LES-based transient CFD simulations were performed to obtain benchmark in silico RMM pressure–flow curves on both sides of a patient-specific nasal cavity geometry obtained from CT images. These simulations, which are among the most detailed simulations of nasal airflow reported in the scientific literature, to date, were used to assess more simplistic steady state laminar- and RANS-based CFD simulations utilizing a relatively coarse mesh.

Notably, almost perfect agreement between the various CFD models was observed with respect to overall flow parameters. However, the CFD results deviated significantly from in vivo RMM data. Analysis of the transient LES model indicates that errors attributable to spatial or temporal resolution were insignificant. The main findings from the simulations were thus that the flow was predominantly near-laminar and quasi-steady. Hence, the initial hypotheses were effectively disproved, directing the attention towards other main sources of errors such as the digital geometry model. The most plausible reason for gross disagreement between in silico and in vivo RMM appears to be misrepresentation of the actual nasal cavity by the digital geometry model. It is proposed that nasal cavity compliance and the drag effect of nasal hair are key components in this regard.

### 4.1. Methods

The present study used patient-specific clinical data for creation of and comparison with CFD simulations. Computed tomography (CT) images were utilized to create a digital 3D geometry model of the patient’s nasal cavity, which was used as basis for CFD simulations of active anterior rhinomanometry (AAR). In silico pressure–flow curves were obtained from the CFD simulations and compared with the patient’s AAR data.

#### 4.1.1. Clinical Data

The clinical data were obtained from a 67-year-old male patient with a body mass index (BMI) of 28. Polysomnography resulted in a measured apnea–hypopnea index (AHI) of 23. He was diagnosed with obstructive sleep apnea and scheduled for nasal surgery to correct septum deviation and increase the volume of the left vestibule. The surgical intervention resulted in a significant improvement in AHI. The current paper presents preoperative patient data and patient-specific CFD simulation results. Preoperative clinical examination was performed at St. Olav’s hospital, the university hospital in Trondheim, and included AAR and CT imaging.

AAR was performed with the patient in a seated position using Otopront^®^ RHINO-SYS rhinometry system [254] before and after decongestion to establish in vivo pressure-velocity curves. Decongestion was achieved through the application of xylometazoline via nasal spray, which was allowed to work for 15 min. Figure 1 shows the resulting RMM measurement output, where blue and red curves correspond to left and right sides of the nose, respectively, and light/dark colors correspond to before/after decongestion. Representative pressure–flow curves corresponding to the decongested state of the nose, used for comparison with CFD simulation data, are shown as dashed black curves.

CT images were obtained with a Siemens Sensation 64, with the patient in the supine position (Figure 4). The scan provided 342 slices of 1.0 mm thickness. The 2D CT images consisted of 512 × 512 pixels. The volume of each voxel was 0.167 mm^3^.

The patient was part of a prospective study approved by the Norwegian Ethical Committee and registered in clinicaltrials.gov (NCT01282125). Written informed consent was obtained from the patient.

#### 4.1.2. Geometry Retrieval

A 3D geometry suitable for CFD simulations was obtained through segmentation of the patient’s CT images. Segmentation was performed using the software ITK-SNAP 3.4.0 [255]. Automatic segmentation using a Hounsfield unit (HU) threshold of −300 was supplemented with manual segmentation in cooperation with medical experts to obtain high-quality geometry. For simplicity, the paranasal sinuses were manually excluded. Due to the narrow passage connecting the paranasal sinuses with the nasal cavity, it is expected that negligible gas exchange takes place between the two [171]. Furthermore, the inclusion of the paranasal sinuses will increase the complexity and size of the computational mesh, hence require greater computational effort. The entire procedure of segmentation and geometry preparation was described by Jordal et al. [5]. The 3D geometry was truncated at the nostrils and just below the nasopharynx. The nasopharyngeal boundary was extended by a length corresponding to approximately six hydraulic diameters to distance the flow boundary from the region of interest and to dampen possible effects of the boundary condition. The geometry was considered rigid and is shown in Figure 5.

#### 4.1.3. In Silico Rhinomanometry (CFD Modelling)

In silico RMM was performed by CFD simulation of unilateral nasal airflow in the patient-specific 3D geometry. During the simulations, both nasal cavities were included, but one nostril was closed such that there was only airflow through one nasal cavity at the time. The transnasal pressure drop was thus approximated by the pressure difference between the open and the closed nostril, similar to in vivo AAR. See **Part I** (Section 2.3.2) for additional discussion of AAR.

CFD modelling was performed in ANSYS Fluent 2019 R2 [256]. Steady state simulations were performed to produce quasi-steady RMM curves. These simulations utilized laminar, kε realizable, and kω SST models. Transient simulations were performed with the large eddy simulation (LES) to produce the entire RMM curves including potential effects of unsteady, respiratory flow. Three breathing cycles were simulated on each side.

All simulations were isothermal with constant air properties ($\rho =1.225\;\mathrm{kg}/m^{3}$, $\mu =1.7894\times 10^{-5}\;\mathrm{Pas}$). Additional details regarding the simulations are summarized in Table 2. Most simulation settings were kept at default values as proposed by ANSYS Fluent. Refer to ANSYS Fluent user and theory guides for more details regarding the model settings and theory background [113,257]. In all simulations, convergence was assumed when residuals dropped below 0.001. In the transient simulations, this required between 1 and 7 iterations per time step, depending on the flowrate.

The simulation case-files and simulation results from the LES simulations have been made available to the public under the Creative Commons Attribution-Noncommercial (CC BY-NC) license [14].

**Table 2 bioengineering-11-00239-t002:** Overview of simulations, including simulation type, turbulence model, options activated, and computational mesh utilized.

Turbulence Model	Turbulence Model Options Activated	Computational Mesh	Volumetric Flowrate	Solver Settings	Discretization
Laminar		Coarse	Steady	- SIMPLE pressure-velocity coupling - Rhie–Chow distance-based flux-type	- Pressure: standard - Advected variables: first order upwind - Gradients: cell-based least squares
kε realizable	- Enhanced wall treatment - Curvature correction - Production limiter				
kω SST	- Curvature correction - Production Kato–Launder - Production limited - Intermittency transition model	- Coarse - Fine (peak insp. flow)			
LES	WALE	Fine	Transient	SIMPLE pressure-velocity coupling	- Pressure: second order discretization - Momentum: bounded central differencing - Gradients: node-based Green–Gauss method - Time stepping: bounded second order implicit scheme

##### Turbulence Modelling

For laminar flow, Equations (11) and (12) are solved directly. For turbulent flow employing Reynolds-averaged Navier–Stokes (RANS) turbulence models, such as the kε and kω model families, the velocity is considered a sum of a mean velocity and a fluctuating component whose ensemble average is zero, $u\to \bar{u}+u^{\prime }$ with $\langle u^{\prime }\rangle =0$, such that $\langle u\rangle =\bar{u}$, where the brackets indicate ensemble averaging. Inserting for the fluctuating velocity in Equations (11) and (12) and performing ensemble averaging results in the RANS equations, in which the Reynolds stress term, $\langle \left(u^{\prime }\cdot \nabla \right)u^{\prime }\rangle$, represents the effect of turbulence. In order to close Equation (12), these stress terms are modelled by the Boussinesq hypothesis, where the Reynolds stress terms are assumed to be related to the mean velocity gradients such that the effect of turbulence can be modelled through an increased effective viscosity (eddy viscosity model). Equations (11) and (12) are thus written as the RANS equations,
(18)$$\nabla \cdot \bar{u}=0\;,$$
and
(19)$$\partial_{t}\bar{u}+\left(\bar{u}\cdot \nabla \right)\bar{u}=-\left(1/\rho \right)\nabla \bar{P}+\left(\nu +\nu_{t}\right)\nabla^{2}\bar{u}\;,$$
where the $\nu_{t}$ is the isotropic turbulent kinematic viscosity. In the kε and kω model families, additional equations are solved for the turbulent kinetic energy, k, dissipation rate, ε, and specific dissipation rate, ω, which are used to compute the turbulent kinematic viscosity. kε realizable [258] is an improved variant of the standard kε model that has been shown to have superior performance in modelling complex flows with, e.g., separation and secondary flow features [113]. kω SST is a hybrid model combining the strengths of the kω model, which is considered more accurate in the near-wall region, and the kε model, which is considered better in the outer parts of the turbulent boundary layer. The enhanced wall treatment (EWT) option in ANSYS Fluent also enables near-wall modelling capabilities in the kε models. Thus, both kε with EWT and kω SST are y^+^-insensitive models [112].

For LES turbulence models, the governing equations (Equations (11) and (12)) are filtered through the finite-volume discretization. The LES continuity equation becomes identical to Equation (18), but the Reynolds stress term in the momentum equation is replaced by the subgrid-scale stress tensor, $\tau_{SGS}$,
(20)$$\partial_{t}\bar{u}+\left(\bar{u}\cdot \nabla \right)\bar{u}=-\left(1/\rho \right)\left[\nabla \bar{P}+\nabla \cdot \bar{\tau_{SGS}}\right]+\nu \nabla^{2}\bar{u}.$$

The RANS and LES momentum equations are, however, formally identical. In ANSYS Fluent, the subgrid-scale stress models employ the Boussinesq hypothesis, as the RANS models, and the main difference between the available LES variants is in the calculation of the subgrid-scale turbulent viscosity. Here, the Wall-Adapting Local Eddy-Viscosity (WALE) subgrid-scale stress model [259] was employed. This model has the trait that it predicts zero turbulent viscosity in laminar flow, and it is therefore considered suitable for modelling wall bounded flows with laminar turbulent transition [111,260].

The core idea of the LES modelling strategy is that the majority of the energy carrying turbulent eddies are modelled directly on a sufficiently fine grid, while the remaining (small) fraction of eddies is modelled via the subgrid-scale model. LES is widely accepted as the most accurate turbulence model, second to direct numerical simulation (DNS), which resolves even the smallest turbulent eddies. It is assumed that LES approaches DNS in the limit of high spatial resolution (small grid cells). Additional discussion of turbulence models was provided in **Part I** (Section 2.4.5), and the turbulence models available in ANSYS Fluent are well documented in the ANSYS Fluent User and Theory guides [113,257].

##### Boundary and Initial Conditions

The boundaries of the flow geometry consist of the nasal cavity walls, the left and right nostrils, and the pharyngeal flow boundary (see Figure 5).

CFD simulations were set up to replicate unilateral RMM, where one nostril is occluded while measuring the volumetric flowrate and pressure drop in the contralateral nasal cavity. Hence, all simulations were performed unilaterally, with one nostril closed and the other open, leading to two specific scenarios: left open–right closed (LO-RC) and left closed–right open (LC-RO).

The nasal cavity walls and closed nostril were treated as standard smooth non-slip wall boundaries. The open nostril was specified as a pressure inlet with a specified, constant pressure of 0 Pa(g). In ANSYS Fluent, the pressure formulation at a pressure inlet depends on the flow direction across the boundary. Thus, the total and static pressures at the nostril boundary were alternatingly equal to the specified pressure, during inhalation and exhalation, respectively (see the ANSYS Fluent user guide for details [257]). This choice of boundary condition was aimed at mimicking the difference between inhalation from and exhalation into a stagnant reservoir of air. During inhalation, it is expected that air will experience loss-free acceleration from the far field to the nostril, conserving the total pressure. This implies a negative static gauge pressure of approximately $\rho u^{2}/2$ during inhalation, where u is the mean velocity at the nostril [81]. During exhalation, however, it is expected that the jet of air exiting the nostril will dissipate and lose all its kinetic energy to the surrounding air, and the pressure at the nostril is approximately equal to the far field static pressure.

The pharyngeal flow boundary was specified as an inlet velocity boundary condition, with a uniform velocity corresponding to the specified volumetric flowrate. Negative and positive flowrates indicated inhalation and exhalation, respectively. For steady state simulations, the constant, volumetric flowrate was specified for each simulation, but for the transient simulations, tidal respiration was approximated by
(21)$$Q(t)=-Q_{\max}\cos\left(\frac{2\pi (t-t_{0})}{\tau }\right)\;,$$
where t was the flow time, $t_{0}=1\;s$ was the duration of the pseudo-steady initialization described below, τ=5 s was the duration of one breathing cycle, and the peak volumetric flowrate was $Q_{\max}=600\;\mathrm{mL}/s$. This corresponds to a tidal volume of 955 mL, which is above what would be expected in normal breathing [161,261].

The same turbulence boundary conditions were employed for the open nostril and the pharyngeal flow boundary. Default values were used—5% turbulent intensity, turbulent viscosity ratio of 10, and intermittency of 1 (kω SST, only). For the LES simulations, the synthetic turbulence generator option was selected. See the ANSYS Fluent user and theory guides for details [113,257].

Initial conditions for the transient LES-based simulations were obtained by first performing a steady state fine-mesh kω SST simulation with peak inspiratory flow, $Q=-Q_{\max}$. Next, the resulting flow fields were converted by employing the text user interface command *solve/initialize/init-instantaneous-vel* before pseudo-steady LES simulation with constant peak inspiratory flow, was run for 1 s flow time ($t_{0}=1\;s$). This initialization procedure allowed for the development of the flow fields, and no pronounced start-up effects were observed, in contrast to the report by Bradshaw et al. [159].

##### Spatial and Temporal Resolution

Ansys Meshing [256] was employed to create coarse and fine computational meshes. Default software settings were used, but the capture proximity and capture curvature options were enabled. Inflation layers consisting of five layers of pentahedral prismatic cells and a growth ratio of 1.2 were established along all wall surfaces and the closed nostril, and tetrahedral cells were used for the remaining geometry. The coarse and fine meshes used element sizes of 1 and 0.2 mm, respectively, which resulted in 876 thousand and 44.7 million grid cells. Figure 6 showcases coarse and fine surface meshes. Figure 7 illustrates the computational meshes at selected cross-sections throughout the nasal cavity. For the coarse mesh, the gap between opposing inflation layers was approximately the width of 2–3 grid cells, in the narrowest passages. On the fine mesh, however, even the narrowest passages were finely resolved by many grid cells.

Time steps of 10 microseconds were used for the transient LES simulations.

### 4.2. Results

The main content of the CFD study presented in this part of the paper was the generation of in silico RMM pressure–flow curves from high-fidelity CFD modelling and their comparison to in silico curves obtained from simpler CFD models as well as patient-specific, clinical in vivo measurements.

Most of the simulation results presented here were obtained from the fine-mesh CFD models described earlier. Unless explicitly stated otherwise, the presented data were obtained from the fine-mesh kω SST simulations at peak inspiratory flow. Results from the coarse-mesh steady state kε realizable and kω SST models are limited to the in silico RMM pressure–flow curves. The coarse-mesh laminar simulations were, in addition, used to compare pressure profiles to the LES simulations. Animations showcasing the transient behavior of selected flow field variables, obtained from the LES simulations, are available in the Appendix A.

During the transient simulations, data were systematically saved at regular intervals. For instance, the differential pressure difference between the right and left nostrils, along with pressure and velocity data at selected cross-sections throughout the geometry, was saved at every time step. Additionally, simulation data files containing instantaneous values of all flow variables as well as recorded turbulence statistics, were saved every ten thousandth time step (every tenth second). The entire dataset is accessible in the public domain for non-commercial use [14]. A separate document describing the contents of the dataset is available in the data repository.

Despite capturing turbulence statistics during the transient LES simulations, they are not incorporated or discussed in the current report. This decision was rooted in the continuous variation of mean flowrate in accordance with its sinusoidal boundary condition, making it unclear how turbulence statistics could contribute to the present analysis. Nevertheless, these data are provided in the published dataset, inviting other researchers to scrutinize and engage in discussion regarding its potential applications.

#### 4.2.1. In Silico RMM-Results

In the transient LES-based simulations, three breathing cycles were simulated on each side of the nose. Throughout these simulations, the differential pressure between the right and left nostrils were monitored throughout the simulations. Figure 8 shows the transient behavior of the differential pressure. There were no evident differences between the three breathing cycles on either side. For steady state simulations, repeated simulations with different pharyngeal flowrate conditions were performed.

In generating in silico RMM pressure–flow curves, the volumetric flowrate was plotted against the differential pressure data. During inhalation, the pressure at the closed nostril is lower than at the open nostril, while during exhalation, the opposite situation occurs. Consequently, the differential pressure between the right and left nostril will be positive during inhalation and negative during exhalation on the right side, and vice versa on the left side. It is noted that in the clinical in vivo RMM pressure–flow curves depicted in Figure 1, both differential pressure and volumetric flowrate are non-negative. In the in silico RMM pressure–flow curves presenting the current simulation data, the volumetric flowrate remains non-negative throughout the respiratory cycle. However, the sign of the differential pressure was adjusted to align with the convention where inhalation corresponds to a positive pressure difference between the open nostril and the nasopharynx, and exhalation corresponds to a negative pressure difference. The resulting in silico RMM pressure–flow curves thus emulate clinical RMM curves, ensuring that pressure–flow curves for inhalatory and exhalatory breathing phases on the left and right sides are appropriately plotted in the corresponding quadrants of the pressure–flow chart. Specifically, pressure–flow curves for the inhalation phase on the right and left sides are represented in the upper and lower right quadrants, respectively. Conversely, pressure–flow curves for the exhalation phase on the right and left sides are plotted in the lower and upper left quadrants, respectively.

Figure 9 illustrates how the various steady state coarse-mesh simulations (laminar, kε realizable, and kω SST models) produced almost identical RMM pressure–flow curves, on both sides of the nose. These curves closely agree with the fine-mesh kω SST model, at peak inspiratory flow. During exhalation, the agreement among the models is somewhat diminished, with the laminar model predicting the highest nasal resistance and the kε realizable model predicting the lowest. Figure 10 reveals that the transient fine-mesh LES yields RMM pressure–flow curves nearly identical to those generated by the steady state coarse-mesh laminar simulations. Figure 11 provides a comparison between in silico RMM pressure–flow curves and representative, decongested in vivo RMM pressure–flow curves for the specific patient in this study. It is evident that the CFD simulations severely underpredicted the pressure drop, hence nasal resistance, displaying a discrepancy exceeding one order of magnitude.

Assuming a defined channel length and curve-fitting Equation (4) to RMM data, representative hydraulic diameters for the nasal cavities can be determined. In Figure 12, best-fit curves and best-fit hydraulic diameters are presented for current in vivo and in silico RMM data utilizing a channel length of L=0.1 m. The nasal resistance on the left side was higher than on the right, resulting in a smaller hydraulic diameter on the left side. The hydraulic diameters that best capture the in silico data are notably higher than those representing the in vivo data. Additionally, a discernible difference between inhalation and exhalation, particularly on the right side, is observed in the in silico data. Examination of the in vivo data reveals that on the left side, the curves flatten out for high pressure drops. The inability of the volumetric flowrate to increase with rising pressure drop indicates a proportional relationship between hydraulic resistance and pressure drop, leading to a constant volumetric flowrate (see Equation (1)). This phenomenon may be attributed to a shrinking cross-sectional area caused by the collapse of the nasal cavity. It is interesting to note that despite the observed effect on the left side, the in vivo RMM curves exhibit symmetry with respect to inhalation and exhalation. This symmetry suggests that nasal compliance occurs due to strong Venturi effect, which remains insensitive to the flow direction.

Figure 13 illustrates hysteresis loop resulting from the by the unsteady flow in the transient LES simulation, as discussed in **Part I** (Section 2.3.3). The hysteresis width introduces a fitting restraint on the channel length and hydraulic diameter, and the channel length L is eliminated. The combination of Equations (4) and (9) results in
(22)$$\Delta P=\frac{W_{\Delta P}}{\omega Q_{\max}}\left[\frac{f_{D}}{\pi D_{h}^{3}}Q^{2}+\frac{1}{2}\frac{\partial Q}{\partial t}\right]\;,$$
where $W_{\Delta P}$ is the hysteresis width. By applying Equation (21) to describe the unsteady volumetric flowrate and employing the hysteresis widths observed in the LES-based RMM curves, the hydraulic diameter was adjusted to achieve the best-fit curves shown in Figure 13. The curve fitting process was conducted separately for the inspiratory and expiratory phases on both sides. To optimize the fit, the Haaland equation (Equation (7)) for the (turbulent) friction factor was employed across all flowrates. While the best-fit curve accurately represented the hysteresis loop on the right side, the fit on the left side was less precise. The resulting channel lengths derived by inserting the best-fit hydraulic diameter, measured hysteresis width, and air properties into Equation (9) were L=3.3 cm and 2.6 cm for inhalation and exhalation on the left side, respectively. On the right side, the corresponding lengths were L=4.2 cm and 2.9 cm. These numbers are significantly lower than the physical length of the nasal cavity, underscoring a limitation in the comprehensiveness of Equation (22) in capturing essential physical aspects of the intricacies of nasal air flow. Hydraulic diameters derived from curve-fits of the laminar simulation data demonstrated good agreement with those representing the LES hysteresis when similar channel lengths were employed. This consistency suggests that the laminar simulation effectively captures pertinent physics.

#### 4.2.2. Spatial and Temporal Resolution

Based on peak inspiratory and expiratory flowrates, the coarse-mesh simulations yielded wall $y^{+}$ values below 5, almost everywhere. The fine-mesh simulations yielded wall $y^{+}$ values below 1. Hence, the near wall grid cells were situated in the viscous sublayer almost everywhere, for all flowrates (see Equation (16)), in all the simulations. Wall $y^{+}$ contour plots obtained from the fine-mesh kω SST model are shown in Figure 14 and Figure 15 for the LO-RC and LC-RO peak inspiratory flow simulations, respectively.

Due to the heightened velocities resulting from the narrower nasal cavity, $y^{+}$ values on the left side were slightly higher than on the right side. The narrowest sections exhibited the highest wall shear stress, hence $y^{+}$ values, due to the relatively elevated flow velocities in these regions.

The subgrid-scale turbulent viscosity in the WALE LES model, as implemented in ANSYS Fluent and used in the current simulations, is proportional to the square of the mixing length for subgrid scales [113],
(23)$$l_{LES}=\min(\kappa y,0.325\Delta )\;,$$
where κ=0.4187 is the von Kármán constant, y represents the distance to the nearest wall, κy is thus the length scale of the largest eddies, and Δ is the cube root of the grid cell volume. On the fine mesh, it was only in the two first layers of prismatic grid cells along the walls that Δ>κy/0.325. Thus, the subgrid-scale turbulent viscosity was predominantly determined by the computational grid cell size. The two first layers of near-wall grid cells were inside the viscous sublayer for all flowrates, which aligns with the capabilities of the WALE LES model to correctly represent the behavior of wall-bounded flows [113].

The Kolmogorov length and time scales, representing the smallest scales of turbulence, can be expressed as
(24)$$\eta ={\left(\nu^{3}/\varepsilon \right)}^{1/4},$$
and
(25)$$\tau_{\eta }={\left(\nu /\varepsilon \right)}^{1/2}\;,$$
respectively, where ν is the kinematic viscosity, ε=−dk/dt is the turbulent dissipation rate, and k is the turbulent kinetic energy [109]. Utilizing results from the fine-mesh kω SST simulations at peak inspiratory flow, the Kolmogorov microscales were computed from Equations (24) and (25). Contour plots at selected cross-sections are depicted in Figure 16 for LO-RC and LC-RO simulations.

To assess the spatial and temporal resolution of the fine-mesh simulations, local subgrid scale mixing lengths determined from Equation (23) and time step size were compared to the Kolmogorov length and time scales determined by Equations (24) and (25), respectively. The turbulent dissipation rates needed to compute the Kolmogorov microscales were obtained from the fine-mesh kω SST simulations of peak inspiratory flow. As the turbulent dissipation rate varies with location and flow velocity, the Kolmogorov microscales also exhibit corresponding variations. The contour plots in Figure 17 depict the ratio of (a) the LES and Kolmogorov length scales, $l_{LES}/\eta$; and (b) the time step size and the Kolmogorov time scale, $\Delta t/\tau_{\eta }$, at selected cross sections for LO-RC and LC-RO simulations. Figure 18 presents the distributions of length and time scale ratios through histogram plots with a bin size of 0.1. It is evident that the vast majority of grid cells exhibited subgrid scale mixing lengths smaller than the local Kolmogorov length scale ($l_{LES}/\eta <1$) and had time steps shorter than the Kolmogorov time scale ($\Delta t/\tau_{\eta }<1$). However, the LO-RC simulation generally demonstrated higher ratios compared to the LC-RO simulation, attributed to the overall higher flow velocities in the former. In the LO-RC and LC-RO simulations, 1.2% and 0.2% of the grid cells, respectively, featured subgrid scale mixing lengths exceeding the Kolmogorov length scale. Only the LO-RC simulation exhibited a minor fraction of grid cells (0.01%) where the Kolmogorov time scale was smaller than the time step size. Grid cells with elevated length scale ratios were generally situated within the wall boundary layers, albeit not directly at the wall, in regions characterized by relatively high wall shear stress. The grid cells with the highest time scale ratios were found at the anterior of the middle meatus.

The subgrid-scale turbulent viscosity ratio was close to zero almost everywhere in the nasal cavity, as shown in Figure 19. In the nasopharynx, however, the turbulent viscosity ratio is substantially higher. The grid cells displaying the poorest spatial and temporal resolution were in concordance with those featuring the highest subgrid scale turbulent viscosity.

CFL numbers (see Equation (17)), as calculated by ANSYS Fluent, were obtained from the fine-mesh kω SST simulations of peak inspiratory flow. Contour plots presented in Figure 20 reveal values below 2 on the right side and below 4 on the left side. Hence, the chosen time steps exceeded the recommended range for LES (CFL∼1). It has, however, been shown that CFL∼5 can work in some cases [111]. Given the time-varying flowrates during the respiratory cycle, the CFL number varied periodically between 0 and these maximum values. The implicit formulation of the CFD simulation is robust with respect to high CFL numbers, and no issues regarding numerical stability were experienced.

#### 4.2.3. Flow Velocity Fields

In Figure 21, flow velocity contours on selected cross sections for the LO-RC and LC-RO fine-mesh kω SST simulations of peak inspiratory flow are presented. Notable variability is evident within each cross-section, reflecting a complex flow pattern. The lower part of the nasal cavity, adjacent to the inferior turbinate, emerges as the preferred flow path. Qualitatively, it was noted that the favored flow path on the left side was slightly higher than on the right side, closer to the middle turbinate.

Figure 22 displays velocity contours and vectors on the sagittal symmetry plane, illustrating distinct flow patterns in the nasopharynx at peak inspiratory and expiratory flow. During inspiration, significant mixing occurs, dominated by relatively large unsteady vortical structures. During expiration, however, the nasopharyngeal flow is dominated by the nasopharyngeal jet, as highlighted by Bradshaw et al. [159]. Airflow velocities were generally low enough that the assumption of incompressible flow was valid.

Using unilateral hydraulic diameters from the open side of the cross-sections in Figure 23, Reynolds numbers based on the peak flowrate (600 mL/s) were computed (see Equation (6)). Figure 24 depicts these results for unilateral flow on the left and right sides, respectively. In the anterior part of the nose (nasal gateway), the Reynolds number is notably higher on the left side, whereas in the posterior nasal cavity, it is slightly higher on the right side. Throughout the nasal cavity, the estimated peak Reynolds numbers are in the range that would be expected to be turbulent for fully developed flow in straight channels (>4000).

In Figure 25 and Figure 26, Q-criterion isosurfaces colored by velocity provide a visual representation of vortical structure generation in the transient LES simulations. Notably, during exhalation, there is minimal production of vortexes in the straight pharyngeal inlet section. Conversely, the nasal cavity exhibits substantial vortex production and mixing. During inhalation, the vortexes produced in the nasal cavity are transported downstream through the pharyngeal section. Intriguingly, some vortexes penetrate deeply into the closed nasal cavity. Although their velocities are low, vortexes facilitate the movement of air toward the olfactory region at the top of the nasal cavity, where the sense of smell is located.

#### 4.2.4. Stress Fields

Figure 27 displays static pressure contours on selected cross sections, for the LO-RC and LC-RO fine-mesh kω SST simulations of peak inspiratory flow. In Figure 28 and Figure 29, the distributions of normal stress (static pressure) and wall shear stress on the nasal cavity wall are presented. The most notable pressure drops occur at the nasal gateways, coinciding with the smallest cross-sectional areas. These regions also exhibit the highest wall shear stresses, consistent with the identified $y^{+}$ hot spots in Figure 14 and Figure 15, due to the elevated flow velocity. The significantly higher pressure drop observed in the left nasal gateway, compared to the right, is attributed to the narrower passage on the left due to the deviating septum (see Figure 7).

Figure 30 and Figure 31 show the area-averaged static and total pressures at the cross-sections indicated in Figure 23 for peak inspiratory and expiratory flow in the fine-mesh LES and coarse-mesh laminar simulations, respectively. The averaging was exclusively based on the open nasal cavity. The figures reveal a generally good agreement between the laminar and LES-based models. Additionally, two notable observations emerge from these curves:Comparison of the total pressure development along the length of the nasal cavity indicates varying pressure losses and local flow resistance in different parts of the nose, depending on the flow direction. This suggests that the total unilateral nasal resistance, derived from the integral of the local resistances along the nasal cavity passage, may exhibit dependence on the flow direction.During inhalation, the stagnation pressure at the occluded nostril corresponds well to the static pressure in the nasopharynx for both sides of the nose. Conversely, during exhalation, a closer correspondence is observed between the stagnation pressure at the occluded nostril and the total pressure in the nasopharynx.

**Figure 30 bioengineering-11-00239-f030:**
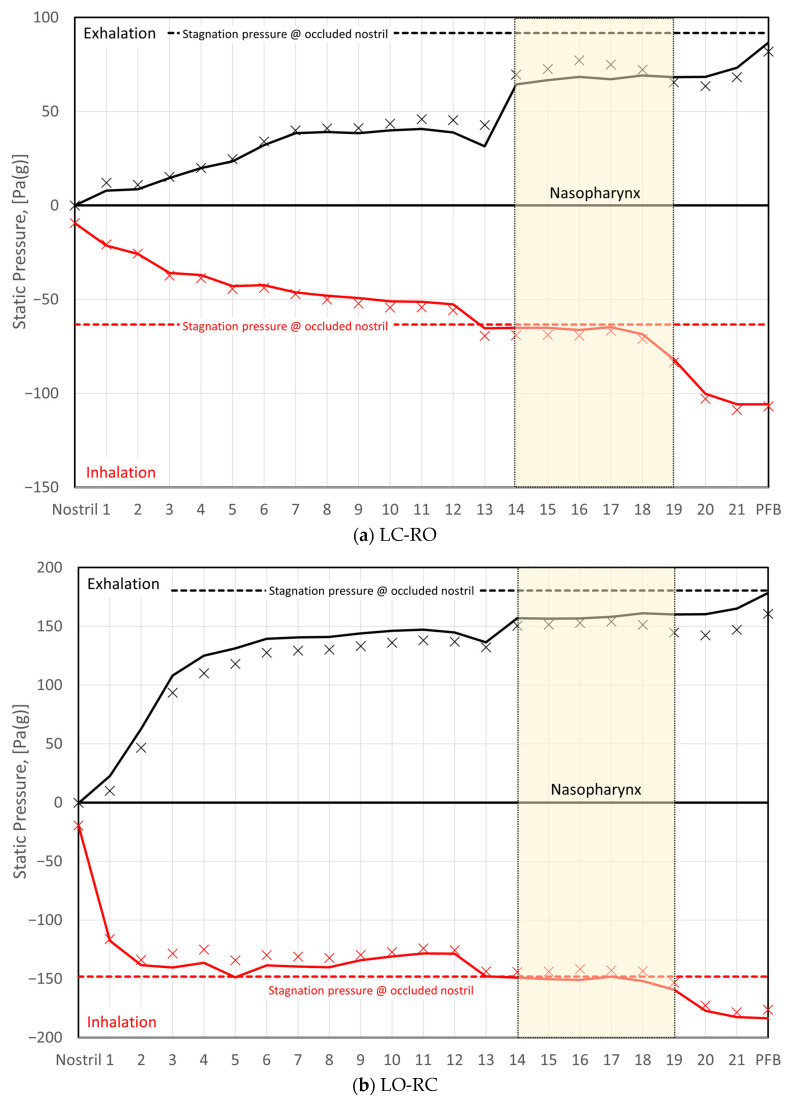
Area-averaged static pressures computed at cross-sections indicated in Figure 23. The data were obtained from fine-mesh LES (solid line) and coarse-mesh laminar (X) simulations, at peak expiratory (black) and inspiratory (red) unilateral flow. The dashed, horizontal lines indicate the area-averaged static pressure at the occluded nostril, obtained from the LES simulation. The hatched area highlights the nasopharyngeal cross-sections.

**Figure 31 bioengineering-11-00239-f031:**
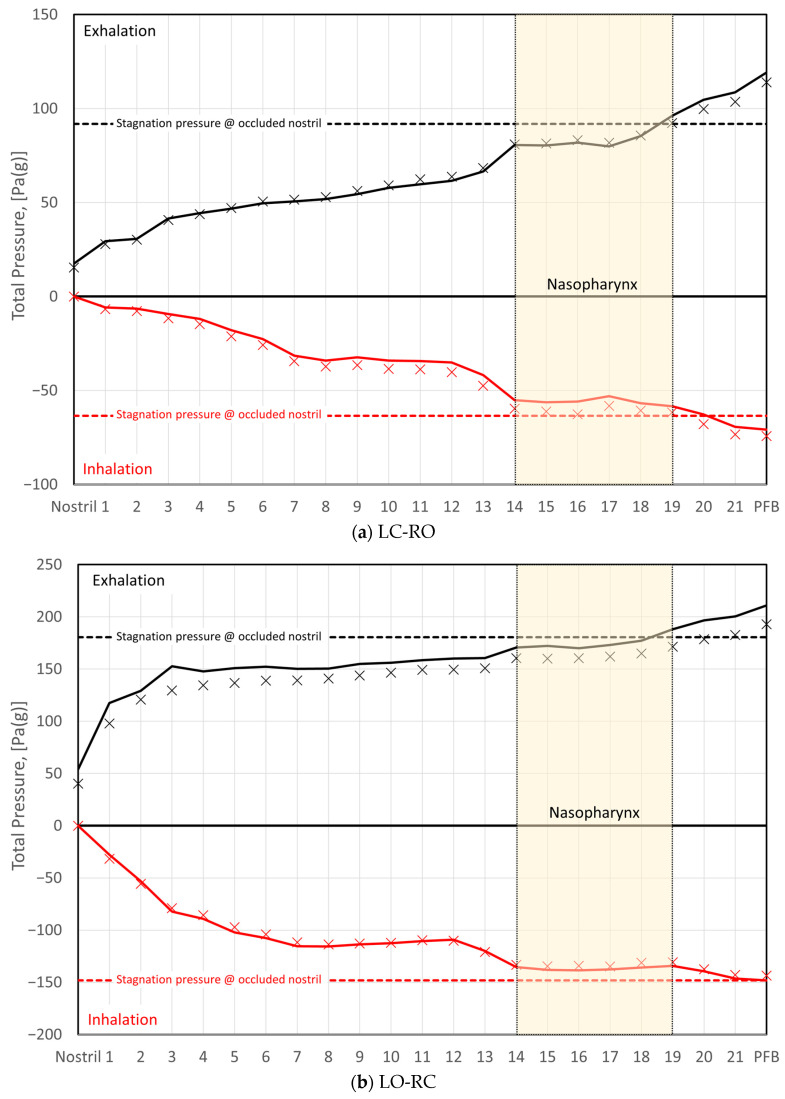
Area-averaged total pressures computed at cross-sections indicated in Figure 23. The data were obtained from fine-mesh LES (solid line) and coarse-mesh laminar (X) simulations, at peak expiratory (black) and inspiratory (red) unilateral flow. The dashed horizontal lines indicate the area-averaged static pressure at the occluded nostril, obtained from the LES simulation. The hatched area highlights the nasopharyngeal cross-sections.

### 4.3. Discussion

The simulation of nasal airflow using CFD poses a challenging task owing to the intricate geometry of the nasal cavity, the dynamic nature of respiratory flow, and limited availability of validation measurements. Despite the growing popularity of CFD in computational rhinology, there remains a shortage of high-fidelity studies assessing the validity of prevalent modeling approaches, such as steady state laminar flow or RANS turbulence modeling, in nasal airflow.

This paper presents highly detailed nasal airflow simulations, representing some of the most comprehensive work published in this field to date. The aim is to contribute significantly to ongoing debates on optimal simulation set-up. Specifically, focus has been on establishing a benchmark for evaluating laminar and RANS-based models, addressing observed gross discrepancies between in vivo and in silico RMM. A key finding is the minimal variation between in silico RMM pressure–flow curves resulting from different turbulence models, including coarse-mesh steady state laminar, kε realizable, and kω SST and fine-mesh steady state kω SST and transient LES. This challenges this study’s initial hypotheses that the disagreement between in vivo and in silico RMM may be explained by poor choice of turbulence model, inadequate spatial/temporal resolution, or unsteady effects.

The following discussion delves into detailed considerations for each element of the initial hypothesis, followed by exploration of factors related to the nasal cavity geometry model. The section is concluded with some final remarks regarding the patient-specific RMM pressure–flow curves and the curve-fitting of the Bernoulli equation (Equation (4)). The discussion should be understood in the context of the broader discussion provided in **Part I** (Section 2.5).

#### 4.3.1. Turbulence Modelling

While the assumption of laminar flow has traditionally dominated CFD modelling of nasal airflow (refer to **Part II**, Section 3), a consensus on whether its characteristic behavior is laminar, transitional, or turbulent remains elusive [83,84,85]. Published comparisons between laminar and turbulent modelling approaches have typically employed steady state RANS-based turbulence models relying on the Boussinesq hypothesis. A big disadvantage with this approach is its assumption that turbulent viscosity is an isotropic scalar quantity, which is not generally true. More sophisticated modelling approaches that account for transient phenomena or employ turbulence models such as Reynolds stress models (RSMs), large eddy simulation (LES), or direct numerical simulation (DNS) tend to be computationally expensive. Few publications have investigated the effects of imposing unsteady flow boundary conditions, such as respiratory tidal flow, or applied these turbulence models to simulate nasal airflow. In addition to the present study, recent investigations utilizing finely resolved LES models with time-varying flow have been conducted by Lu et al. [248], Calmet et al. [249,250], Bradshaw et al. [159], and Hebbink et al. [127]. Other publications have also utilized high-fidelity CFD models under steady flow conditions; refer to Table 1 for an overview.

The present paper’s findings, asserting the similarity between laminar CFD models and more complex turbulence modeling approaches in predicting gross flow features, find support in several other studies. Li et al. [83] demonstrated that their laminar model exhibited good agreement with LES and DNS results at a volumetric flowrate of 180 mL/s. Calmet et al. [250] similarly concluded that a laminar flow model accurately predicted gross flow features, albeit with some underestimation of local flow fluctuations and turbulence intensity. Calmet et al. [250] and Bradshaw et al. [159] observed transitional and dominantly laminar nasal airflow, respectively. In the current study, the unilateral pressure drop exhibited low sensitivity to the choice of computational mesh or turbulence modelling approach on both sides of the nose across a wide range of volumetric flowrates.

Considering the calculated Reynolds numbers for the current patient-specific model alone, turbulent flow would be anticipated in both nasal cavities at peak flowrates (refer to Figure 24). However, when factoring in the respiratory frequency of f=0.2 Hz, the kinematic viscosity of air, and a hydraulic diameter of less than 1 cm into Equation (14), it becomes evident that the Womersley number for the current simulations was of the order of 1. This falls within the low range, as discussed by Xu et al. [168], suggesting a delayed transition to turbulence. These findings, coupled with the simulation results indicating near-laminar flow, illustrate that the critical Reynolds number for turbulent pipe flow is not a reliable indicator for selecting an appropriate turbulence model for respiratory nasal airflow.

In wall-bounded flows, there is a risk of underestimating turbulence intensity in RANS- and LES-based CFD models if the wall boundary layer is not treated appropriately. The nasal cavity is characterized by intricate flow cross-sections featuring high aspect ratios between the axial, spanwise, and wall-normal length scales. Hence, geometrical complexity alone makes it challenging to achieve a computational mesh that satisfies the wall requirements of conventional wall-function-based turbulence models. For varying volumetric flowrates, the challenge amplifies. It is, therefore, recommended to employ a computational grid that resolves the viscous sublayer and utilize $y^{+}$-insensitive turbulence models capable of accurate representation of the wall shear stresses. In the present study, these criteria were met for all turbulence models employed, indicating the attainment of accurate and reliable simulation results. A discussion of spatial and temporal resolution is provided below.

Lastly, it is acknowledged that turbulence requires initiation. This can occur through boundary effects such as wall roughness, turbulent fluctuations, e.g., at inlets, geometric irregularities such as bends, edges, or steps, or flow obstacles such as nasal hair. If crucial turbulence triggers are missing from the model, predicted laminar-like flow might not accurately reflect the physiological situation. As shown in Figure 25, during exhalation, little vortical flow existed in the pharyngeal inlet section. The vortices generated at the inlet dissipated before reaching the nasopharynx. Nevertheless, the significant vortex production in the nasopharynx and nasal cavities implies that there were sufficient triggers present to generate turbulence in the nasal cavity.

#### 4.3.2. Spatial and Temporal Resolution

The wall $y^{+}$-insensitive RANS models provided in ANSYS Fluent are expected to perform well in the wall $y^{+}$ ranges seen in the current simulations, both on the coarse and fine meshes. Using these RANS models avoids the problems associated with employing wall functions [112]. Wall $y^{+}<1$, which was seen almost everywhere at peak flow in the fine-mesh simulations, generally satisfies the requirement of fully resolved LES models [111], where the wall shear stress is computed directly from the laminar stress–strain relationship [113].

The Kolmogorov length and time scales (Equations (24) and (25)) offer a valuable metric for evaluating the spatial and temporal resolution in numerical models of turbulent flow [108,109,110]. At the Kolmogorov scale, viscous forces dominate, leading to dissipation of turbulent energy into thermal energy. These scales characterize the smallest turbulent eddies, and detailed models such as DNS must therefore resolve the Kolmogorov microscales [262]. Kolmogorov’s hypotheses [263] posit that turbulence is isotropic at the smallest scales, implying that the Boussinesq hypothesis can be reasonably accurate. Consequently, $y^{+}$-insensitive RANS- and LES-based models exhibit high accuracy, approaching DNS, when the spatial and temporal resolution is of the order of the Kolmogorov microscales.

Figure 18 indicates that the majority of fine-mesh grid cells featured LES mixing lengths and timesteps smaller than the Kolmogorov length and time scales, respectively. The spatial and temporal resolution, comparable to the Kolmogorov microscales, along with a finely resolved wall boundary layer, suggests that the fine-mesh LES simulations were finely resolved, albeit not fully, during peak flow. Because the Kolmogorov microscales increase with decreasing velocities, the LES was most likely fully resolved at lower volumetric flowrates, however. As the computational mesh exhibited sufficient resolution to resolve the smallest turbulent eddies, and the temporal resolution effectively resolved the turbulent time scale, the LES can be considered near-DNS, within the nasal cavity. This view is further supported by the relatively small turbulent viscosity ratios depicted in Figure 19. Consequently, errors attributable to spatial or temporal resolution are deemed negligible in the fine-mesh LES.

It is noted, however, that the turbulent dissipation rate is not readily available from the present LES simulations, since it relies on statistical analysis of fully developed turbulent flow. In the current simulations, the volumetric flowrate was continuously varying due to the oscillatory nature of respiratory flow, and pseudo-steady statistics were thus not available. Hence, the formulae for calculating the Kolmogorov microscales (Equations (24) and (25)) are not directly applicable for assessing the LES simulations. To address this limitation, the Kolmogorov microscales used in Figure 16, Figure 17 and Figure 18 were derived from the steady state fine-mesh kω SST simulations at peak inspiratory flow. These simulations assumed constant volumetric flowrate, hence pseudo-steady turbulence statistics. This approach, while a necessary adaptation, provides valuable insights despite the dynamic nature of the respiratory flow in the current simulations.

Rigorous statistical analysis of the LES data was not performed in the current study, but the simulation data have been made freely available to the public [14]. Other investigators are encouraged to examine and analyze the dataset. It is worth noting that in the continuously developing flow fields due to the oscillatory nature of respiratory flow, the steady state statistics of classical turbulence theory might not hold. The temporal (and spatial) development of the out-of-equilibrium turbulent energy spectrum is a topic for further study [264].

#### 4.3.3. Transient Effects

The physiological respiratory cycle deviates from a perfectly sinusoidal function of time. Notably, there exists an inherent asymmetry between inhalation and exhalation, leading to a marginally higher peak inspiratory flowrate compared to the peak expiratory rate. Consequently, during exhalation, a plateau emerges, characterized by an almost constant flowrate. Realistic breathing cycles have been presented by, e.g., Van Hove et al. [223], Calmet et al. [250], and Pawade [129]. In transient simulations, obtaining turbulence statistics benefits from prolonged intervals of near-constant flowrate, enabling a broader time-averaging window. With a continuously varying mean flow velocity, as in the current LES, turbulence statistics sampling becomes challenging with respect to computational cost, since it must rely on a high number of breathing cycles.

Turbulence evolves distinctively in continuously accelerating/decelerating flow compared to steady flow [167]. A period of near-steady flow, as seen in physiological expiration curves, could potentially promote turbulence development if the Reynolds number is sufficiently high. Nevertheless, in the current simulations, the RANS-based models, representing fully developed steady flow, contradict this notion concerning peak flow in the specific patient-specific model.

Inthavong et al. [105] proposed that eliminating start-up effects in transient simulations, starting from a quiescent state, required two to three breathing cycles, whereas Bradshaw et al. [159] dismissed the first cycle. In the current simulations, a fine-mesh steady state kω SST model was employed to establish initial peak inspiratory flow conditions for the transient LES. Notably, no start-up effects were observed, and there were no discernible differences between successive breathing cycles. Consequently, it can be inferred that the start-up effects were either enduring or entirely absent.

The nasal airflow simulation results presented in this paper show that a series of steady state simulations closely replicated the transient simulation of the entire (sinusoidal) breathing cycle. Thus, for the specific patient considered, the respiratory nasal airflow could be characterized as quasi-steady. This observation aligns with expectations, given the low Womersley number (Equation (14)). Quasi-steadiness implies that the shape of the respiratory cycle is non-substantial. However, it is important to note that the width and shape of the unsteady hysteresis affecting RMM pressure–flow curves are influenced by the time-derivative of the volumetric flowrate, hence the slope of the breathing profile (refer to **Part I**, Section 2.3.3 and Figure 13). It is underlined that the hysteresis loops affecting the pressure–flow curves resulting from the transient LES were solely due to the non-zero time derivative of the flowrate, not start-up effects or nasal compliance.

To sum up, while the present simulation data revealed some transient effects, their influence on the in silico RMM results was found to be minimal. Therefore, transient effects are deemed highly improbable as an explanation for the significant disparity observed between in silico and in vivo RMM results.

#### 4.3.4. Geometry

One potential source of error in the comparison of in silico and in vivo RMM data, not investigated in this study, pertains to the consistency between the digital geometry model used in CFD simulations and the actual nasal cavity geometry during RMM. For instance, Aasgrav [4] highlighted the substantial impact of segmentation settings on nasal resistance by contrasting CFD models based on different HU threshold settings. Beyond this, there are additional points that warrant discussion in relation to patient-specific geometry models. This section aims to underscore key aspects to be mindful of when evaluating the congruence between the digital representation and the real anatomical features. Refer also to the broader discussions in **Part I** (see Section 2.4.3, Section 2.5.2 and Section 2.5.4).

A notable deviation from realistic airway geometry in the current digital patient-specific model is the truncation of the geometry at the nostrils and the subsequent truncation and extrusion of the pharyngeal tract. The former restricts the faithful representation of airflow distribution across the open nostril during inhalation and exhalation, as well as the turbulent intensity during inhalation. However, Taylor et al. [180] observed that overall flow patterns and measures were insensitive to the detailed prescription of inflow conditions at the nares. The latter predominantly affects the turbulent intensity during exhalation, as illustrated in Figure 25, where the flow featured minimal vorticity when it entered the nasopharynx, during exhalation. This indicates that production of vorticity and turbulent structures was delayed until the nasopharynx and nasal cavity. While the precise influence on simulation results remains unclear, this observation aligns with Bradshaw et al.’s findings [159], underscoring the importance of accurately describing and including the pharyngeal tract to model vortex generation and transport during both inhalation and exhalation. Neglecting the paranasal sinuses is consistent with the perspective of several other researchers [126,159,170,171].

When exploring additional factors that could influence the comparability of the patient-specific geometry employed in in silico RMM and the actual airway geometry during in vivo RMM, two primary considerations emerge: (1) the condition of the nasal cavity during CT image acquisition and (2) the impact of nasal compliance.

During in vivo RMM, nasal cavity decongestion was achieved through the application of xylometazoline. This is expected to maximize nasal cavity volume, through a reduction in turbinate swelling, and minimize nasal resistance. However, the CT image acquisition, the basis for the digital patient-specific geometry model, took place in the natural, non-decongested state. As a result, it was subject to the nasal cycle and various factors affecting spontaneous turbinate and nasal mucosa swelling. For instance, in the natural non-decongested state, postural effect on the nasal resistance and nasal cycle may be anticipated [33,133,134]. The nasal resistance is expected to be higher in the supine position, in which CT images were obtained, than in the sitting position, in which the in vivo RMM data were obtained.The lack of decongestion and the postural effect are both expected to increase the nasal resistance. Hence, the correction of these sources of error would presumably reduce the nasal resistance predicted by the in silico RMM, further increasing the disagreement with the in vivo RMM data.It has been proposed that nasal compliance may affect RMM curves (see **Part I**, Section 2.5.1), and that rigid CFD geometries will fail to reproduce in vivo RMM curves due to this. However, for the current patient, it is observed that the patient-specific in vivo RMM curves are almost symmetrical with respect to in/exhalation (see Figure 12a). At the same time, particularly on the left side, the in vivo RMM pressure–flow curve plateaus, suggesting a significant increase in nasal resistance beyond a critical flowrate. To explain these observations by nasal compliance, a collapsible constriction is required, where the Venturi effect dominates over the static pressure in such a way that the collapse is independent of the flow direction. Without such a constriction, asymmetrical collapse would be expected due to the under-/over-pressures in the nasal cavity during the inspiratory and expiratory phases, respectively. For instance, the phenomenon of nasal gateway collapse exemplifies this, where the collapse occurs exclusively during inhalation [265,266].

The first point suggests that the discordance between in silico and in vivo RMM may be more prominent than indicated by the current findings. On the other hand, the second point provides a plausible rationale for the significant misalignment between simulations and measurements. This contradicts the conventional notion that a decongested nasal cavity retains rigidity, calling for robust evidence to demonstrate the (partial) symmetrical collapse of certain nasal cavity parts during both inhalation and exhalation.

For future studies comparing in silico and in vivo RMM, it is recommended to exercise caution in ensuring that the patient-specific geometry model accurately represents the airway geometry during in vivo RMM. This can be achieved by minimizing the time gap between clinical RMM and CT data acquisition and ensuring that both are obtained in a decongested state, preferably in the same position (either supine or sitting). This approach will contribute to reducing a major source of error in the analysis of in silico nasal airflow data.

As a final point, the role of nasal hair should be considered. Hahn et al. [151] and Stoddard et al. [174] presented differing conclusions regarding the significance of nasal hair. Hahn et al. found little impact on overall airflow, while Stoddard et al. observed a positive effect on both subjective and objective measures of nasal obstruction following the removal of nasal hair, suggesting its noteworthy contribution to nasal resistance. A recent CFD study by Haghnegahdar et al. [267] indicated that nasal hair significantly influenced nasal airflow patterns, although they did not report nasal resistance. The potential impact of nasal hair on nasal resistance appears to warrant increased attention.

#### 4.3.5. Final Remarks and Observations Regarding the Analysis of Rhinomanometry

##### Pressure Measurements

In the course of anterior rhinomanometry (RMM), the assessment of nasopharyngeal pressure involves measuring the stagnation pressure at the occluded nostril. This pressure is then used as an approximation of the nasopharyngeal pressure. The differential pressure between the open nostril and the occluded nostril thus serves as an indicator of the pressure drop across the nasal cavity, facilitating the calculation of nasal resistance (refer to **Part I**, Section 2.3 for more details). However, the simulation findings discussed in the preceding section (see Figure 30 and Figure 31) indicate that interpreting these measurements may not be straightforward. Firstly, the results indicate that nasal resistance could be influenced by the direction of airflow (inhalation versus exhalation). Additionally, it is noted that the measured stagnation pressure corresponds primarily to the static nasopharyngeal pressure during inhalation, while during exhalation, it aligns more closely with the total pressure.

Anticipation of the former aligns with classical potential theory, where symmetry in flow direction reversal only occurs in ideal situations. As illustrated in Figure 25 and Figure 26, the vorticity of nasal airflow is substantial, and non-recoverable pressure losses, such as those caused by vortex shedding, are expected to vary based on the flow direction, particularly when there are abrupt changes in cross-sectional areas or flow direction [81].

The latter phenomenon can be elucidated using principles akin to those governing airspeed measurements with Pitot tubes in, for example, aviation (see Figure 32). In this context, when the flow of air is directed towards the cavity with its opening facing the flow, it is deflected due to a force exerted by the fluid inside the cavity. This force is counteracted by the stagnation pressure at the cavity’s bottom. Conversely, for the cavity with its opening facing the opposite direction, the wake behind the cavity may induce a suction force, reducing the pressure at the cavity’s bottom. The observed disparity in simulation results between the static pressure at the occluded nostril and the total pressure in the nasopharynx during exhalation can be attributed to flow curvature. As the airflow enters the nasopharynx, it bends towards the open nasal cavity, and vortical structures penetrate into the occluded nasal cavity (see Figure 25 and Figure 26). These factors necessitate the application of a force that influences the stagnation pressure. In the context of RMM, this implies a potential overestimation of nasopharyngeal pressure during exhalation, leading to an underestimation of pressure drop and nasal resistance.

In standard RMM, there is unfortunately insufficient information to rectify the inherent overprediction of nasopharyngeal pressure during exhalation. However, if the cross-sectional area of the open nostril were measured, it would allow for the estimation of total pressure at the nostril, enabling the calculation of pressure drop as a change in total pressure. In channels with varying cross-sectional areas, recoverable changes in static pressure may result from the acceleration or deceleration of the flow. Conversely, changes in total pressure are solely due to non-recoverable losses, suggesting that total pressure measurements may provide a more reliable basis for estimating nasal resistance compared to static pressure measurements.

From this discussion, it becomes evident that interpretation and application of standard clinical anterior RMM examinations should consider two key factors: (1) the physical nasal resistance may be influenced by the direction of airflow, and (2) exhalatory pressure measurements are likely to underpredict the pressure drop and unilateral nasal resistance.

##### The Use of Bernoulli’s Equation

The patient-specific clinical in vivo RMM pressure–flow data presented in Figure 1 shows that there was significant effect on the nasal resistance by decongesting the nose, on both sides. However, the data exhibits significant variability, with evident large hysteresis loops affecting each breathing cycle. Particularly on the left side, where the nasal resistance was highest, the scatter in differential pressure values span a wide range, for given volumetric flowrates. The characteristic loops observed in the in vivo RMM pressure–flow curves have been previously discussed by Vogt et al. [77,139], who proposed that nasal compliance contributes to the hysteresis, and Groß and Peters [140], who suggested that measurement techniques in RMM may be a contributing factor. In **Part I** (Section 2.3.3) it was illustrated how hysteresis might result from unsteady flow, alone.

The representative in vivo RMM curves presented in Figure 1 were created by manual placement of pressure–flow points, to represent the minimal nasal resistance observed in the data. It is evident that this representative nasal resistance significantly underestimates the actual nasal resistance during certain recorded breathing cycles.

The discrepancy between in silico and in vivo RMM is evident not only in terms of nasal resistance but also in other aspects. First, there is much less scatter in the in silico pressure data. Although pressure fluctuations are present in the LES-based pressure–flow curves (see Figure 8 and Figure 10), they manifest as a noisy signal typical of turbulent fluctuations. Additionally, the tight hysteresis envelopes inherent in the LES data appear quite different from the large hysteresis loops observed in the in vivo data (compare Figure 1 and Figure 13).

Excellent curve-fits of Bernoulli’s equation (Equation (4)) were obtained for both in silico and in vivo RMM pressure–flow curves (refer to Figure 12 and Figure 13). Deviation from the straight lines associated with laminar flow in a straight tube, have been attributed to the transition to turbulent flow [78]. Such behavior was observed both in in vivo and in silico RMM curves presented here. The present CFD simulations have, however, provided compelling evidence that the flow was laminar. Conversely, the optimal curve-fit was obtained using the (turbulent) friction factor obtained from the Haaland equation (Equation (7)) regardless of Reynolds number. These findings suggest that considerations of turbulence alone may not be sufficient for assessing RMM pressure–flow curves. Without delving into an exhaustive analysis, it is suggested that the apparent turbulent behavior, when assessing the Bernoulli equation curve-fit in isolation, may be influenced by other factors related to the deviation between the nasal cavity and a straight pipe of constant diameter. For instance, the non-circular shape of the nasal cavity, numerous abrupt changes in cross-sectional area, bends, and curves, along with the presence of nasal hair and mucosal lining, may introduce minor friction losses that should be accounted for [81]. Additionally, the impact of unsteady flow, such as significant vortex shedding, as illustrated in Figure 25 and Figure 26, and the possibility of nasal compliance should be considered.

### 4.4. Summary and Conclusions

Some of the most detailed CFD simulations of nasal airflow, to date, were performed to investigate commonly reported gross discrepancies between in vivo unilateral anterior rhinomanometry and in silico CFD simulation results. Specifically, the aim of this study was to investigate the threefold hypothesis that these discrepancies can be explained by (1) poor choice of turbulence model, (2) insufficient spatial or temporal resolution, or (3) neglecting transient effects, in CFD models.

A patient-specific digital nasal airway geometry model was obtained by segmentation of preoperative CT images from an OSA patient. Finely resolved transient LES was employed to assess steady state laminar and $y^{+}$ insensitive RANS simulations on a relatively coarse mesh. A remarkable agreement was observed between pressure–flow curves obtained with the various CFD models. Nasal pressure drop, hence nasal resistance, was severely underpredicted compared to in vivo RMM, however.

Although local Reynolds numbers ranged from 4 to 14 thousand throughout the nasal cavities at peak flow, the simulation results imply that the modelled flow was predominantly near-laminar/transitional. The delayed development of turbulence may be explained by the oscillatory nature of respiratory flow as well as the complex shape of the nasal airway, preventing the development of turbulent boundary layers.

Spatial and temporal resolution of the LES model was of the order of the Kolmogorov microscales at peak flow, suggesting that errors attributable to spatial or temporal resolution were insignificant.

Although transient phenomena such as pressure–flow curve hysteresis and unsteady vortex shedding were observed in the transient LES results, comparison of pressure–flow curves from steady state and transient simulations revealed that the flow could be considered quasi-steady.

In summary, the present simulation results effectively disproved the paper’s initial hypothesis, and the cause for disagreement between in silico and in vivo RMM must be explained by other means. The most plausible factor appears to be that the digital geometry model was a poor representation of the actual nasal cavity. Key aspects in this regard may be nasal cavity compliance or drag caused by nasal hair.

## Figures and Tables

**Figure 1 bioengineering-11-00239-f001:**
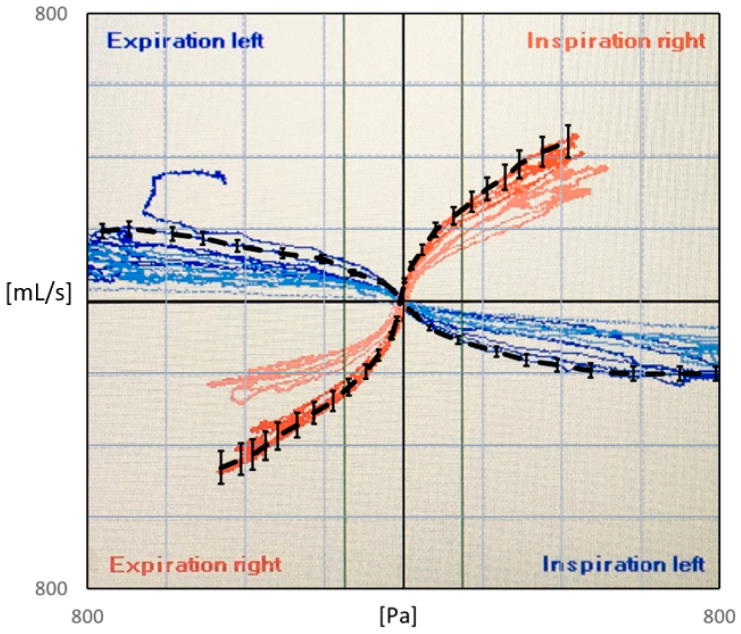
Rhinomanometry output for the current patient. Red corresponds to right side and blue corresponds to left side RMM, and light/dark colors indicate before/after administration of decongestive nasal spray. Black curves with 10% error bars show the selected “measured data” used in the current paper.

**Figure 2 bioengineering-11-00239-f002:**
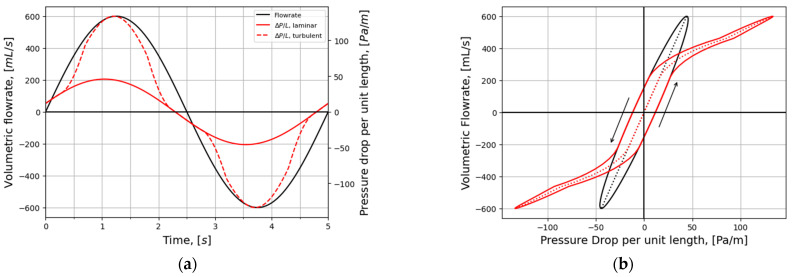
Synthetic rhinomanometry data obtained using a sinusoidal volumetric flowrate (Equation (8)), with $Q_{\max}=600\;\mathrm{mL}/s$ and τ=5 s, and pressure drop per unit length calculated from Bernoulli’s equation, Equation (4), for a horizontal, straight, smooth pipe of hydraulic diameter $D_{h}=10\;\mathrm{mm}$. (**a**) Volumetric flowrate (black) and pressure drop per unit length (red) as functions of time, for laminar (solid curve) and turbulent (dashed curve) flow. (**b**) Pressure–flow curves for laminar (black) and turbulent (red) flow. Dotted curves neglect the unsteady term in Equation (4). Arrows indicate the evolution in time.

**Figure 3 bioengineering-11-00239-f003:**
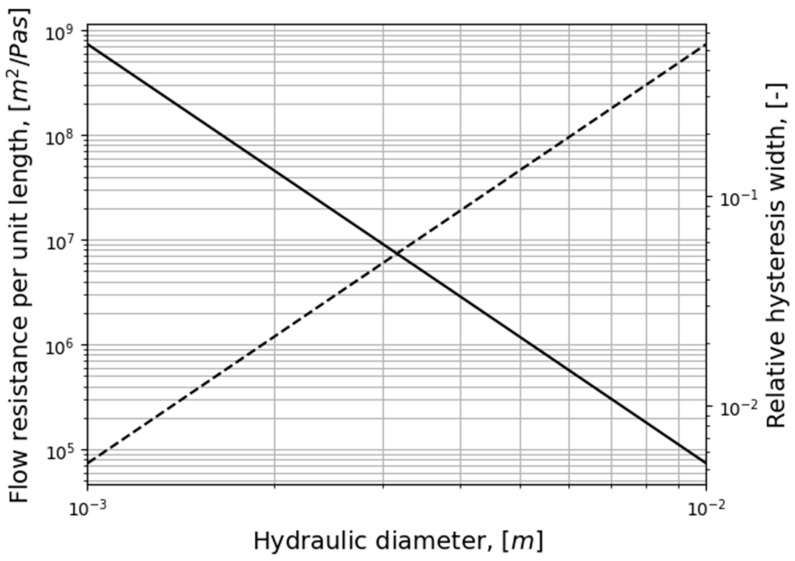
Flow resistance per unit length (Equation (2)) (solid curve) and relative hysteresis width (Equation (10)) (dashed curve) for laminar air flow.

**Figure 4 bioengineering-11-00239-f004:**
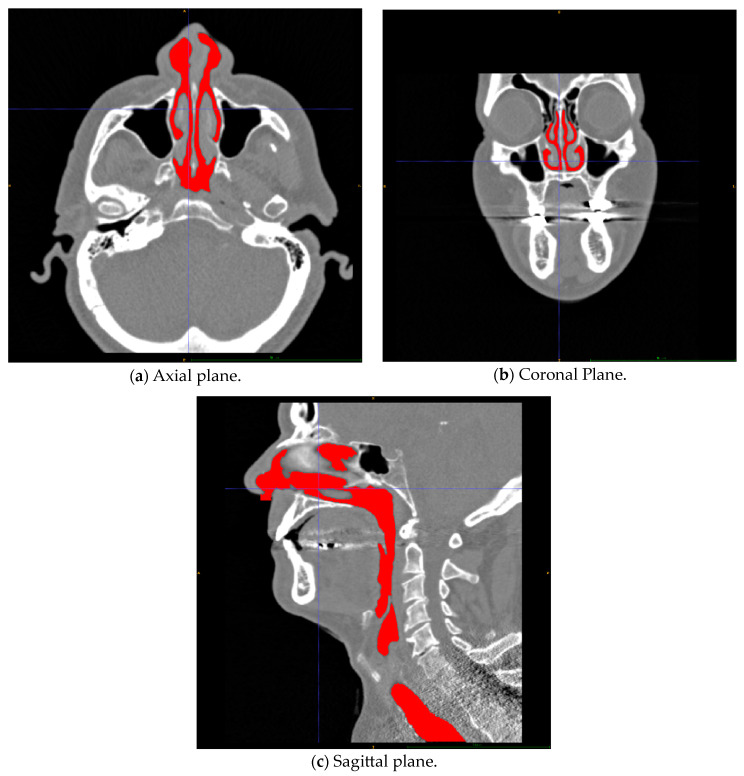
Computed tomography images of the current patient. The modelled airway is colored red, and the blue lines in each cross-section indicate the position of the other cross-sections.

**Figure 5 bioengineering-11-00239-f005:**
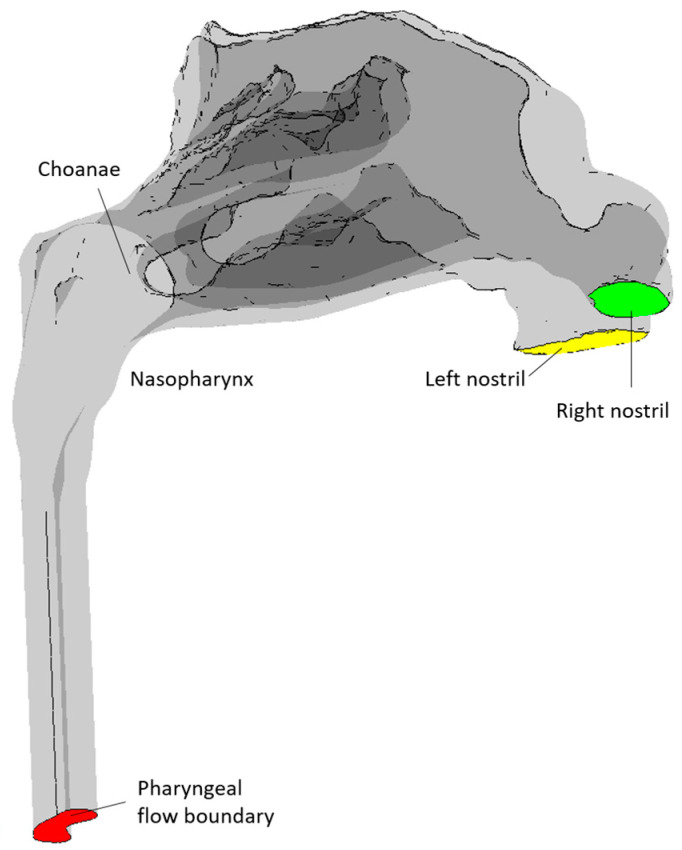
Flow geometry obtained from segmentation of CT images. Paranasal sinuses and the oral cavity are omitted from the model. Computational boundaries are indicated by color: Air–tissue wall boundary (gray), pharyngeal flow boundary (red); right nostril (green); left nostril (yellow).

**Figure 6 bioengineering-11-00239-f006:**
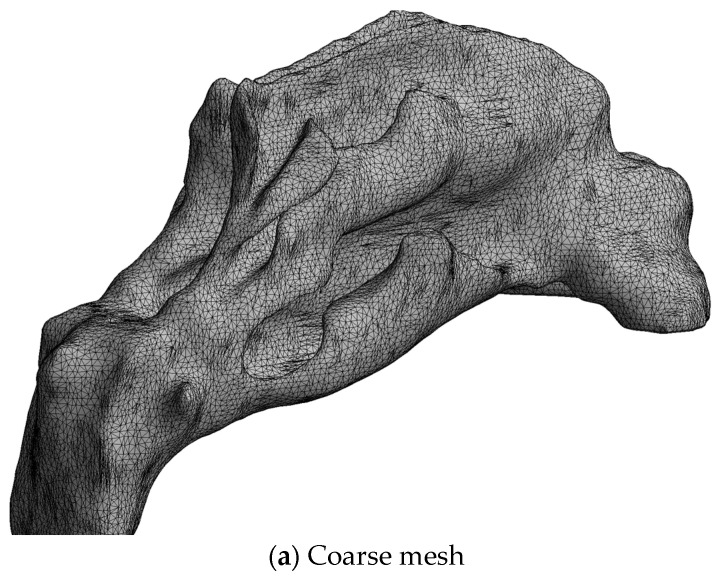
Surface mesh.

**Figure 7 bioengineering-11-00239-f007:**
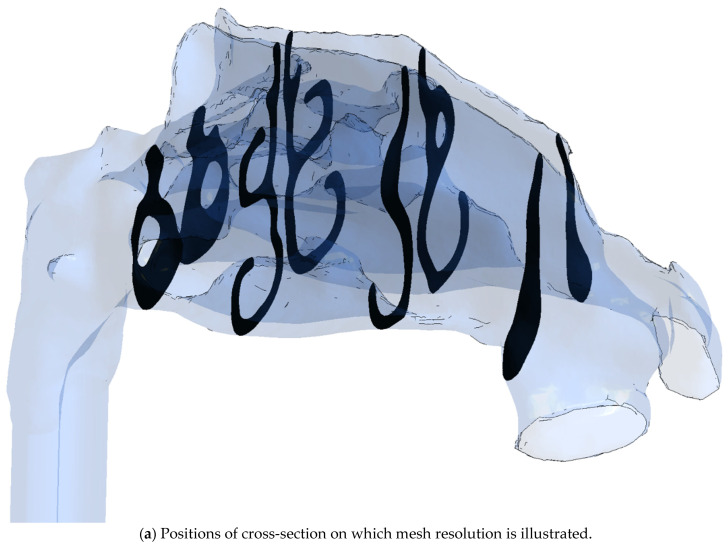
Visualization of the coarse and fine mesh at selected cross-sections through the nasal cavity.

**Figure 8 bioengineering-11-00239-f008:**
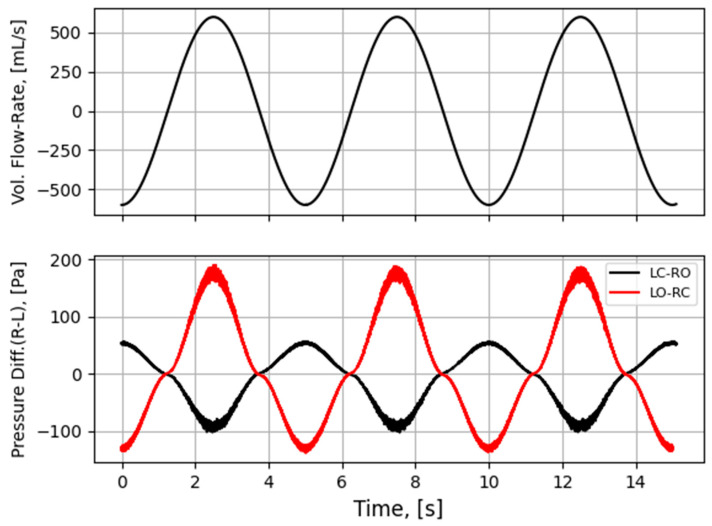
Volumetric flowrate (**top**) and nasal pressure drop (**bottom**) vs. time, in LES-based in silico unilateral AAR. Negative and positive flowrates indicate inhalation and exhalation, respectively. LC/O = left closed/open. RC/O = right closed/open.

**Figure 9 bioengineering-11-00239-f009:**
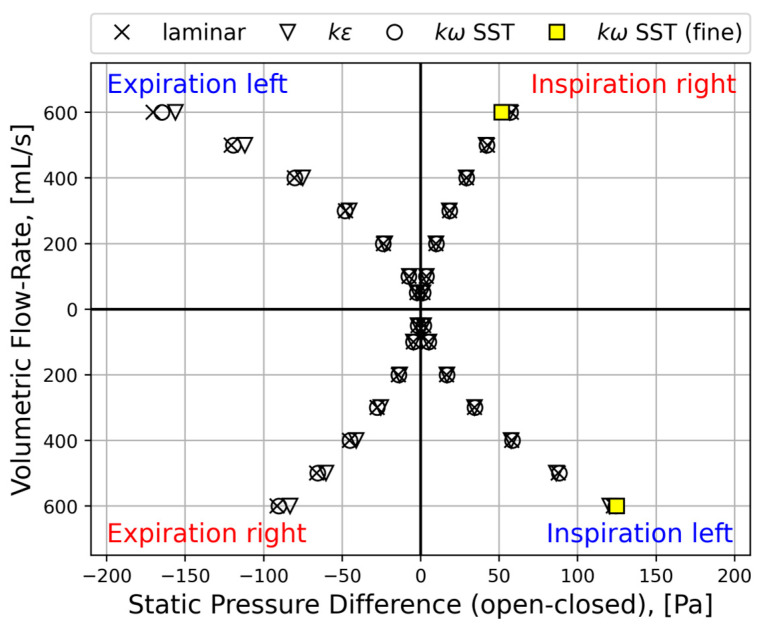
Quasi-steady in silico RMM pressure–flow curves obtained from CFD simulations utilizing steady state laminar (×), kε realizable with enhanced wall treatment (∇), and kω SST (◯) on the coarse mesh, and kω SST (□) on the fine mesh.

**Figure 10 bioengineering-11-00239-f010:**
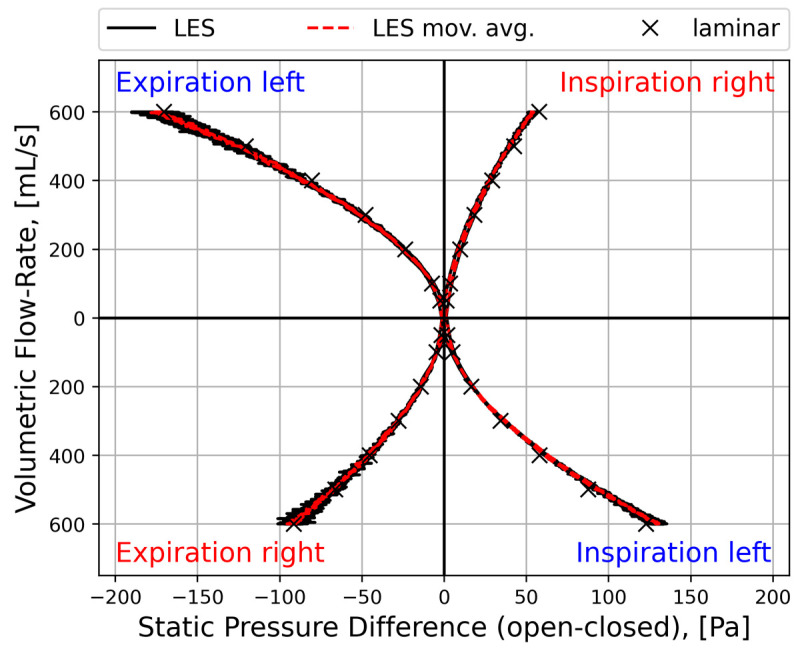
In silico RMM pressure–flow curves obtained from transient LES-based (solid line) and quasi-steady laminar (×) CFD simulations.

**Figure 11 bioengineering-11-00239-f011:**
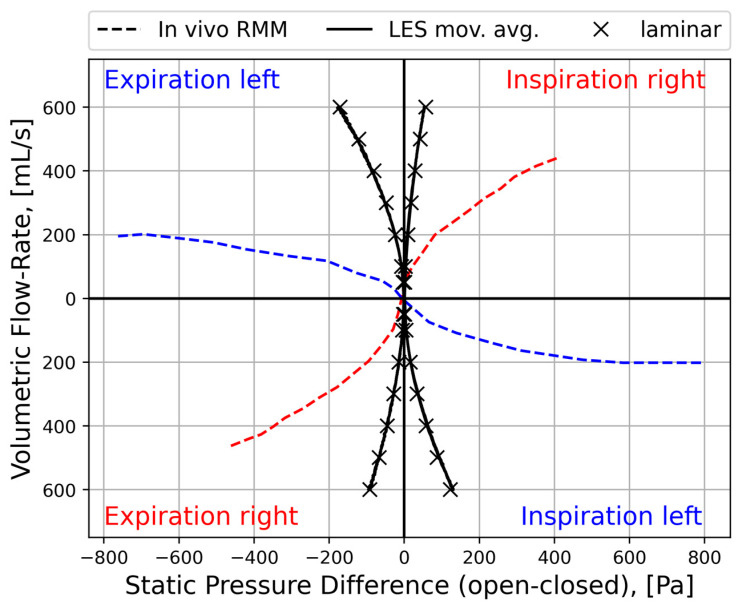
Comparison of in vivo and in silico RMM pressure–flow curves obtained from clinical measurements (dashed line) and moving average of the transient LES-based (solid line) and quasi-steady laminar (×) CFD simulations.

**Figure 12 bioengineering-11-00239-f012:**
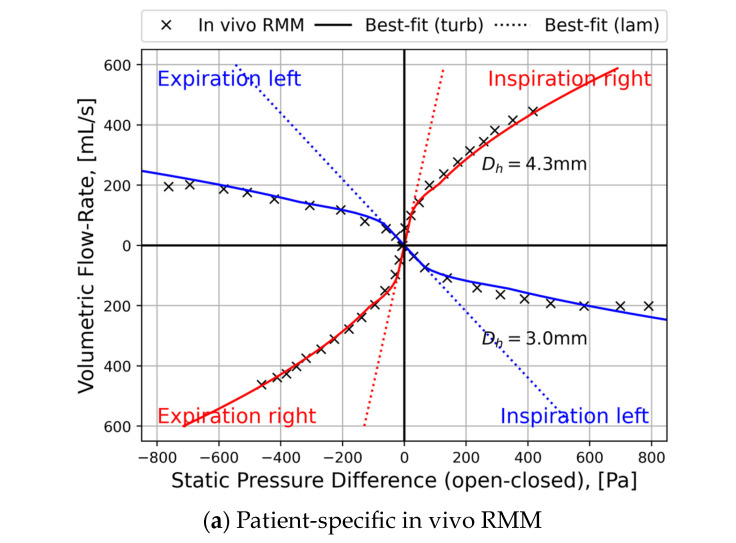
RMM curves (×) with best-fit (solid) curves obtained by assuming a channel length of L=0.1 m, neglecting the unsteady term, and adjusting the hydraulic diameter, $D_{h}$, in Equation (4). Best-fit hydraulic diameters are printed in the figures. For the in vivo data, a single hydraulic diameter was used for in- and exhalation, but for the in silico data, in- and exhalation data were curve-fitted separately. The in vivo RMM curves are representative of patient-specific RMM pressure–flow curves obtained in the decongested state (see Figure 1).

**Figure 13 bioengineering-11-00239-f013:**
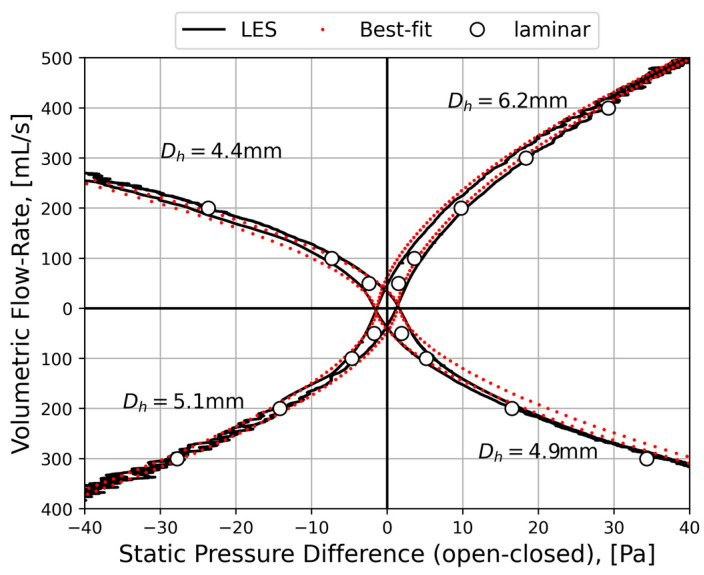
Comparison between LES (solid black), LES best-fit curve (dotted red), and laminar (O) in silico RMM simulation results.

**Figure 14 bioengineering-11-00239-f014:**
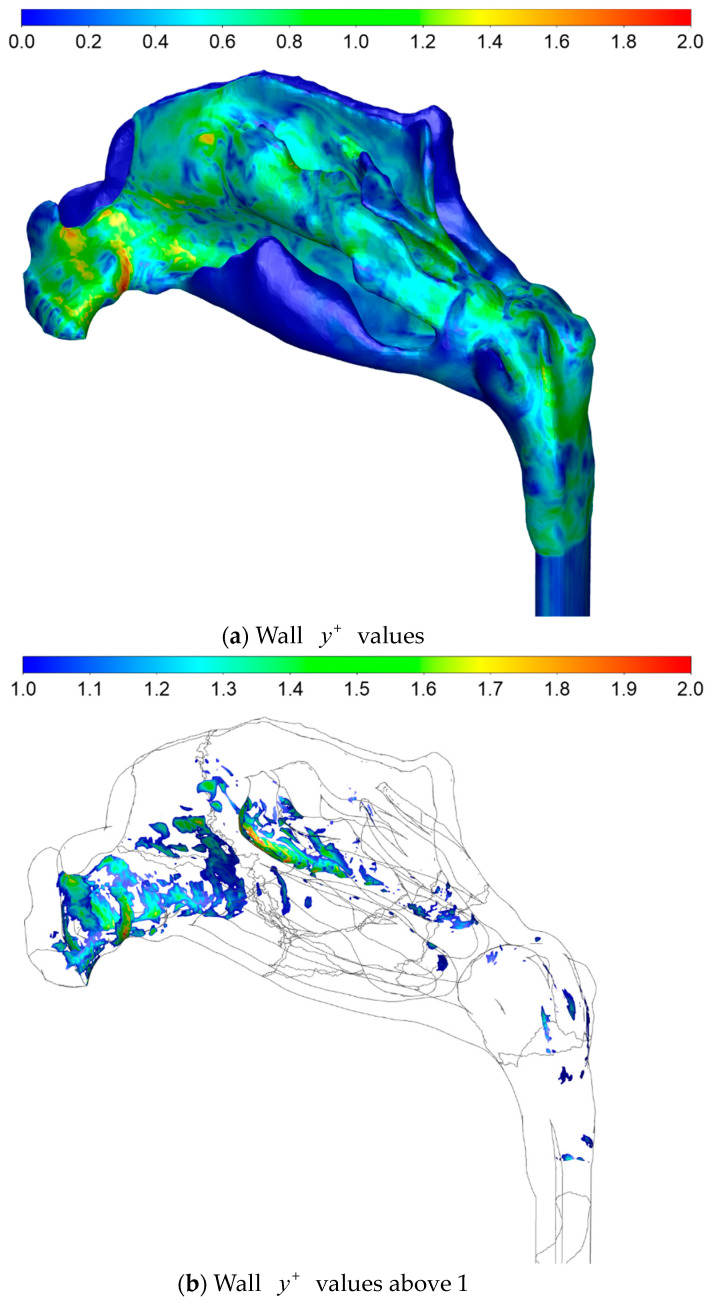
Estimated wall $y^{+}$ values based on the LO-RC steady state fine-mesh *kω* SST simulation of peak inspiratory flow (see Equation (15)).

**Figure 15 bioengineering-11-00239-f015:**
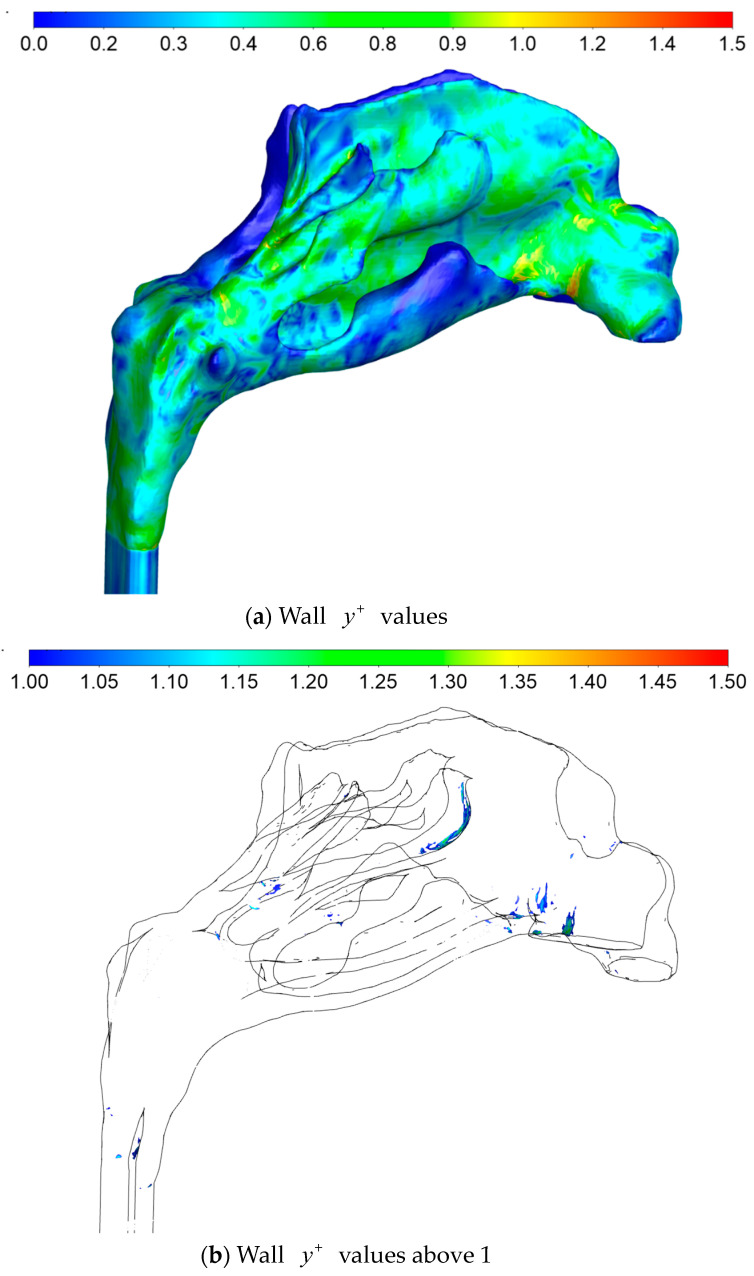
Estimated wall $y^{+}$ values based on the LC-RO steady state fine-mesh *kω* SST simulation of peak inspiratory flow (see Equation (15)).

**Figure 16 bioengineering-11-00239-f016:**
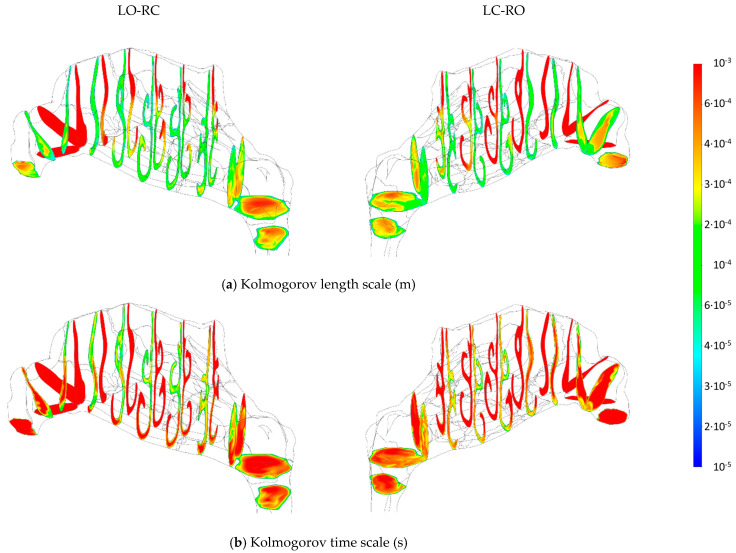
Kolmogorov micro scales: (**a**) length scale; and (**b**) time scale computed from the LO-RC and LC-RO fine-mesh *kω* SST peak inspiratory flow simulations.

**Figure 17 bioengineering-11-00239-f017:**
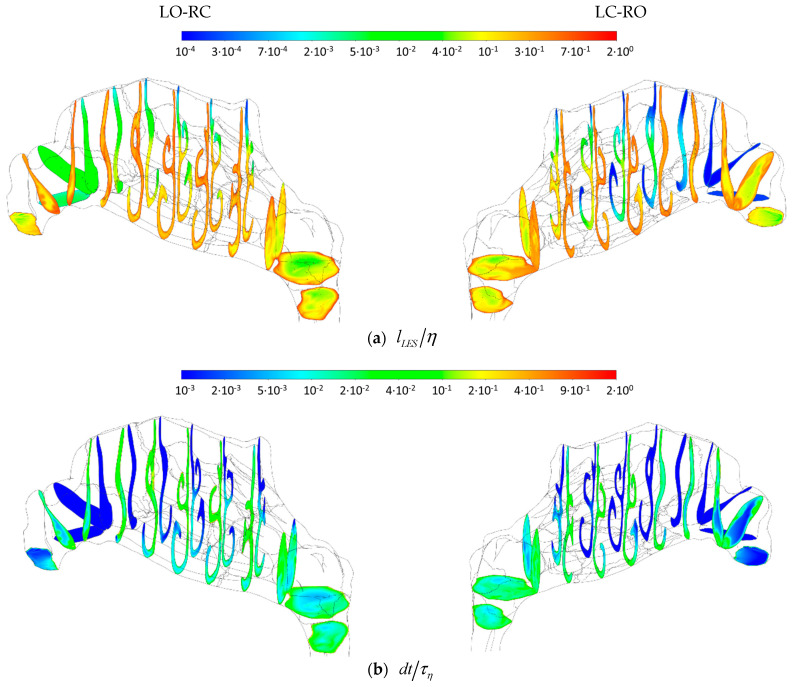
Contour plots showing the ratios of (**a**) LES simulations mixing lengths (Equation (23)) to the Kolmogorov length scales (Equation (24)); (**b**) LES simulations time step size (10 μs) to Kolmogorov time scales (Equation (25)), based on the LO-RC and LC-RO fine-mesh *kω* SST peak inspiratory flow simulations.

**Figure 18 bioengineering-11-00239-f018:**
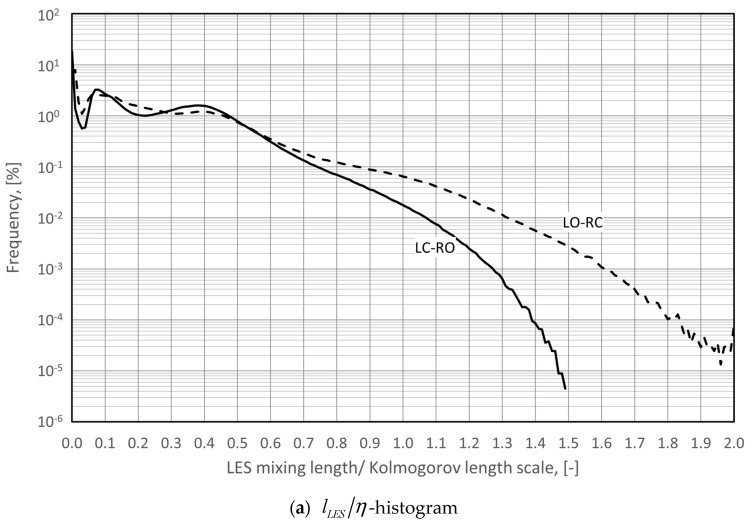
Histograms illustrating the distribution of ratios of (**a**) LES simulations mixing lengths (Equation (23)) to the Kolmogorov length scales (Equation (24)); (**b**) LES simulations time step size (10 μs) to Kolmogorov time scales (Equation (25)), based on the LO-RC and LC-RO fine-mesh *kω* SST peak inspiratory flow simulations. Bin sizes are 0.01.

**Figure 19 bioengineering-11-00239-f019:**
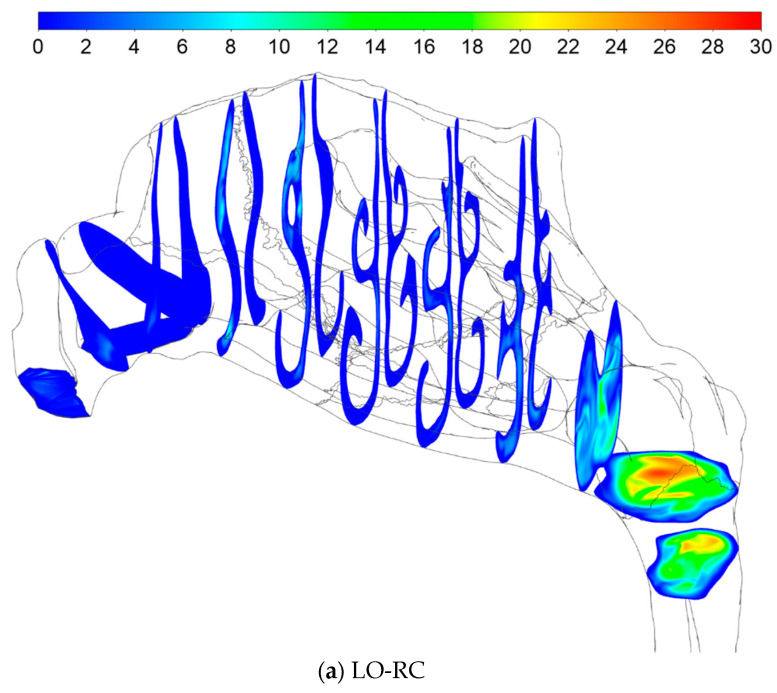
Estimated turbulent viscosity ratio values based on the fine-mesh *kω* SST peak inspiratory flow simulations.

**Figure 20 bioengineering-11-00239-f020:**
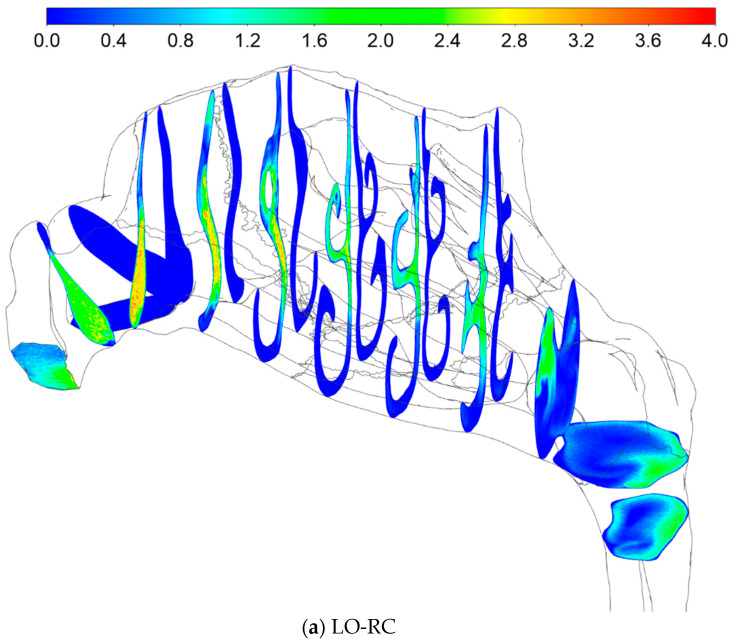
*CFL* number contour plots at selected cross-sections based on the fine-mesh *kω* SST peak inspiratory flow simulations and time step size of Δt=10 μs (see Equation (17)).

**Figure 21 bioengineering-11-00239-f021:**
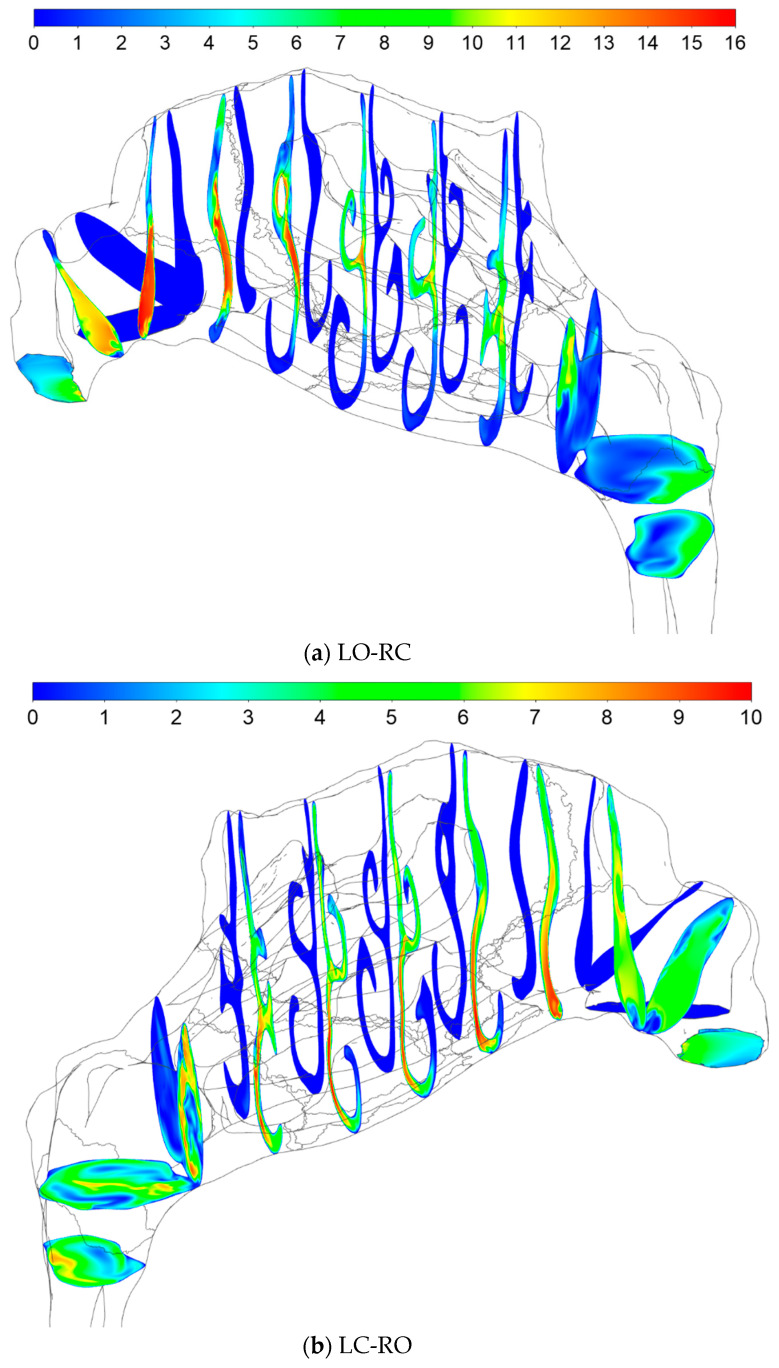
Flow velocity magnitude (m/s) contour plots at selected cross-sections for (**a**) LO-RC and (**b**) LC-RO, fine-mesh *kω* SST peak inspiratory flow simulations.

**Figure 22 bioengineering-11-00239-f022:**
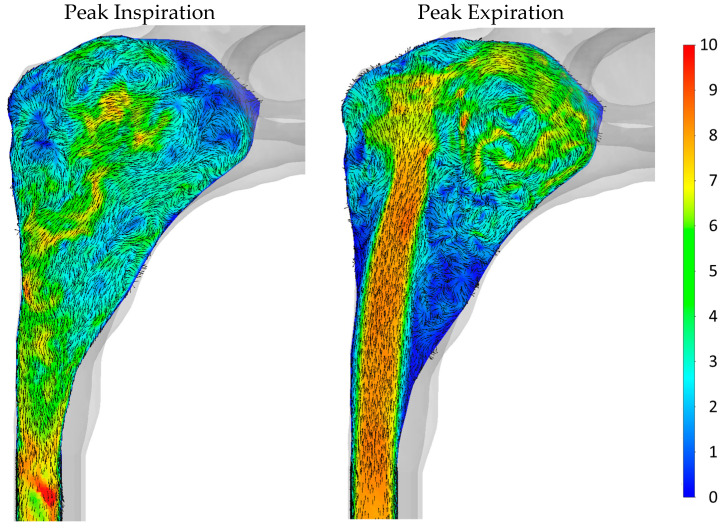
Nasopharyngeal velocity contours (m/s) and vectors drawn at the sagittal symmetry plane for peak inspiratory and expiratory flow for the transient LES-based LC-RO simulation on the fine mesh.

**Figure 23 bioengineering-11-00239-f023:**
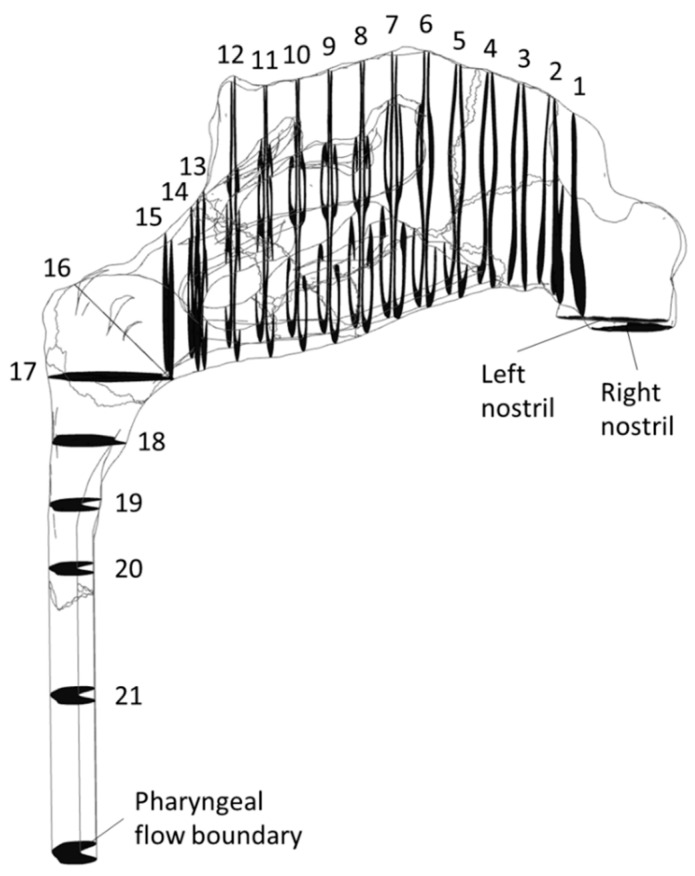
Illustration of cross section positions used in subsequent figures. The cross-sections 1–13 were positioned at regular intervals of 0.5 mm. The two nasal cavities merge into the nasopharynx between cross-sections 13 and 14. Nasopharynx is thus represented by cross-sections 14–19. The cross-sections 20 and 21 are identical in shape and size to the pharyngeal outflow boundary.

**Figure 24 bioengineering-11-00239-f024:**
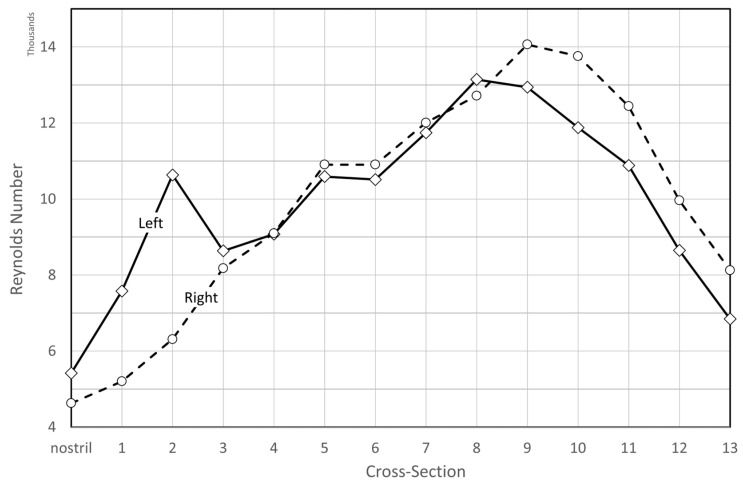
Estimated Reynolds number based on unilateral peak flow (600 mL/s) and the unilateral hydraulic diameter of the open side (Equation (6)). Cross-section numbers refer to positions given in Figure 23.

**Figure 25 bioengineering-11-00239-f025:**
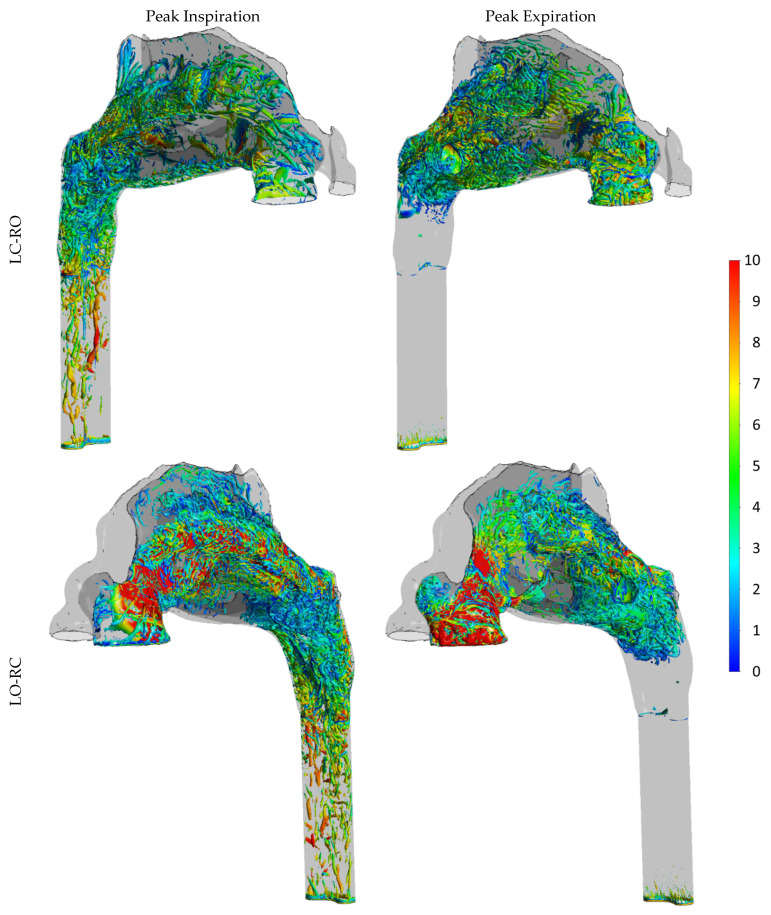
Q-criterion iso surfaces (value of $3\times 10^{6}\;L/s^{2}$) viewed from the side (sagittal-coronal view) and colored by velocity magnitude (m/s), at peak inspiration (**left**) and expiration (**right**) during the third respiratory cycle in in silico unilateral rhinomanometry, based on transient LES simulations on the fine mesh.

**Figure 26 bioengineering-11-00239-f026:**
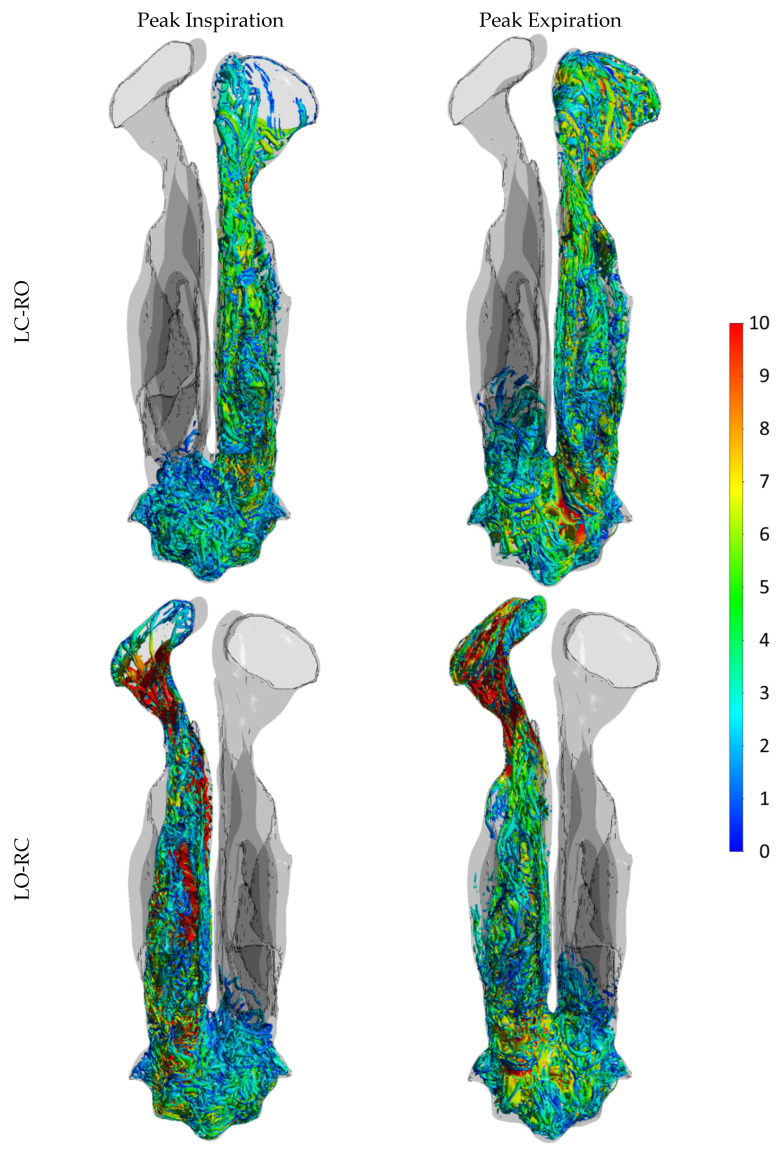
Q-criterion iso surfaces (value of $3\cdot 10^{6}\;L/s^{2}$) viewed from above (axial view) and colored by velocity magnitude, at peak inspiration (**left**) and expiration (**right**) during the third respiratory cycle in in silico unilateral rhinomanometry, based on transient LES simulations on the fine mesh.

**Figure 27 bioengineering-11-00239-f027:**
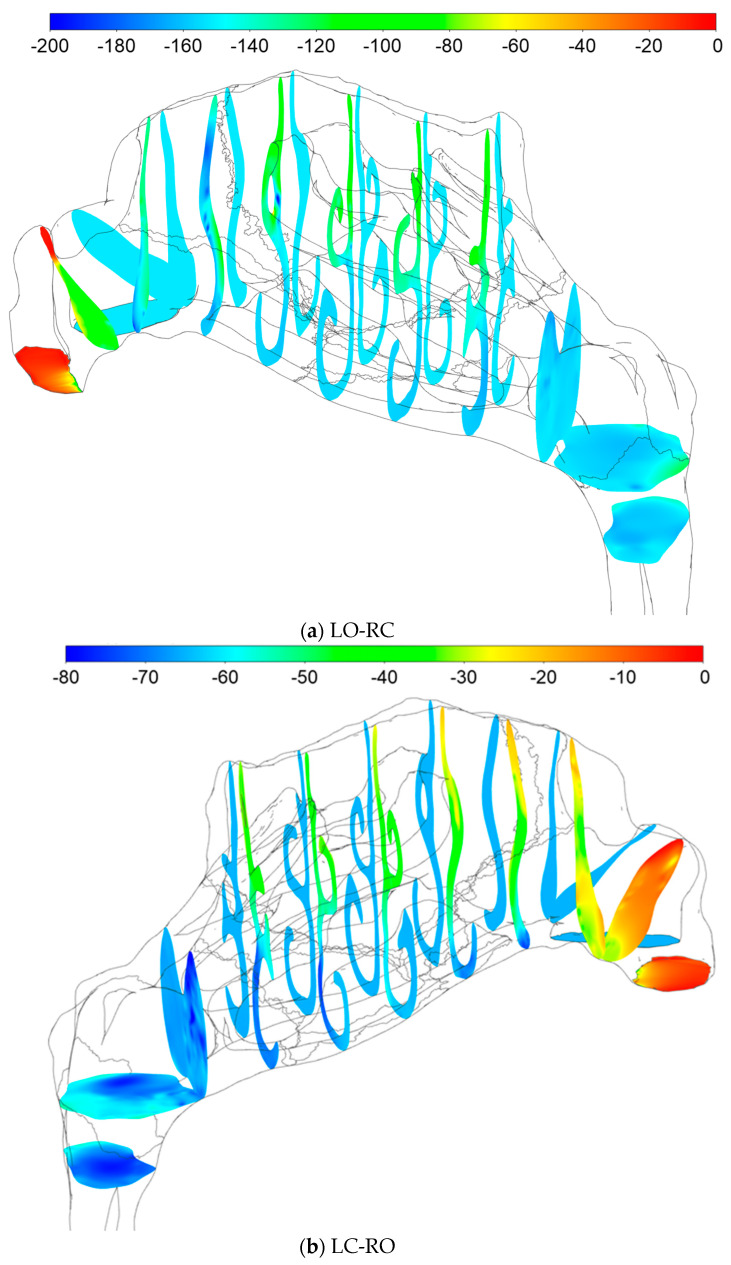
Static pressure contour plots (Pa) at selected cross-sections for (**a**) LO-RC and (**b**) LC-RO, steady state fine-mesh *kω* SST simulation of peak inspiratory flow.

**Figure 28 bioengineering-11-00239-f028:**
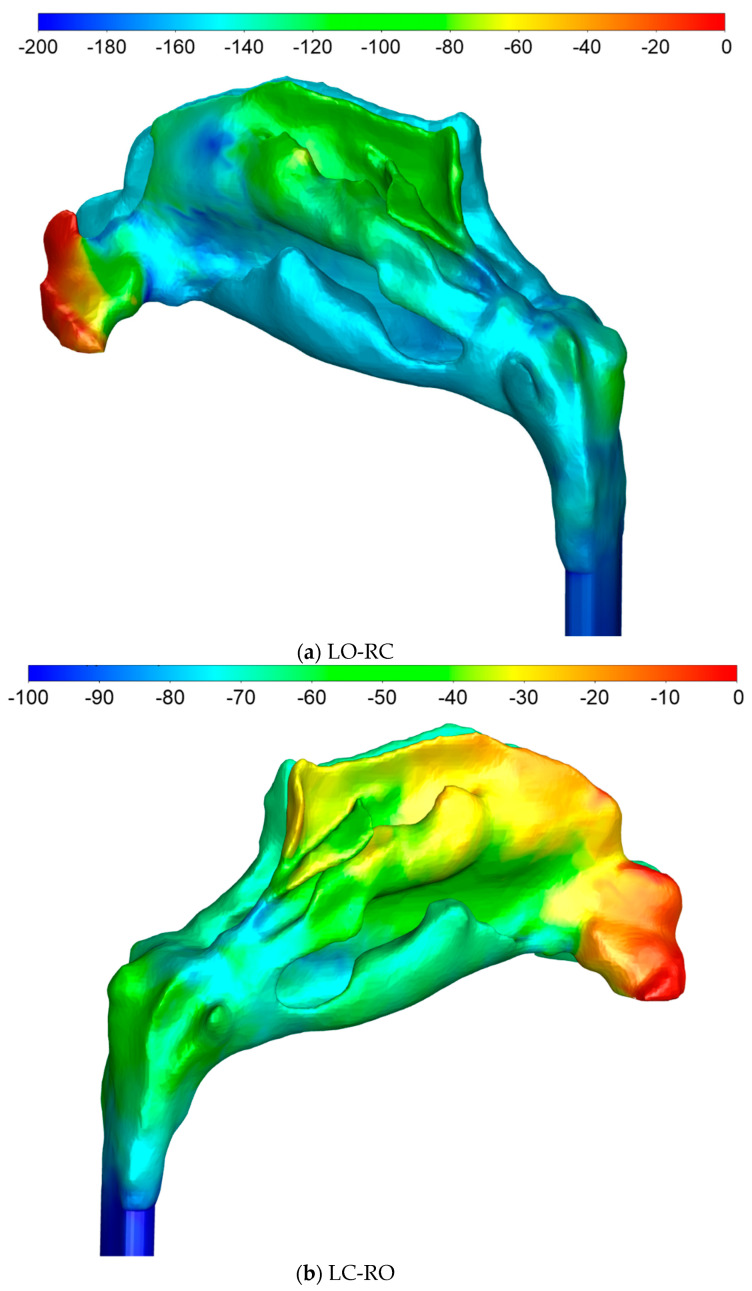
Static pressure contour plots (Pa) at the nasal cavity wall for (**a**) LO-RC and (**b**) LC-RO, steady state fine-mesh *kω* SST simulation of peak inspiratory flow.

**Figure 29 bioengineering-11-00239-f029:**
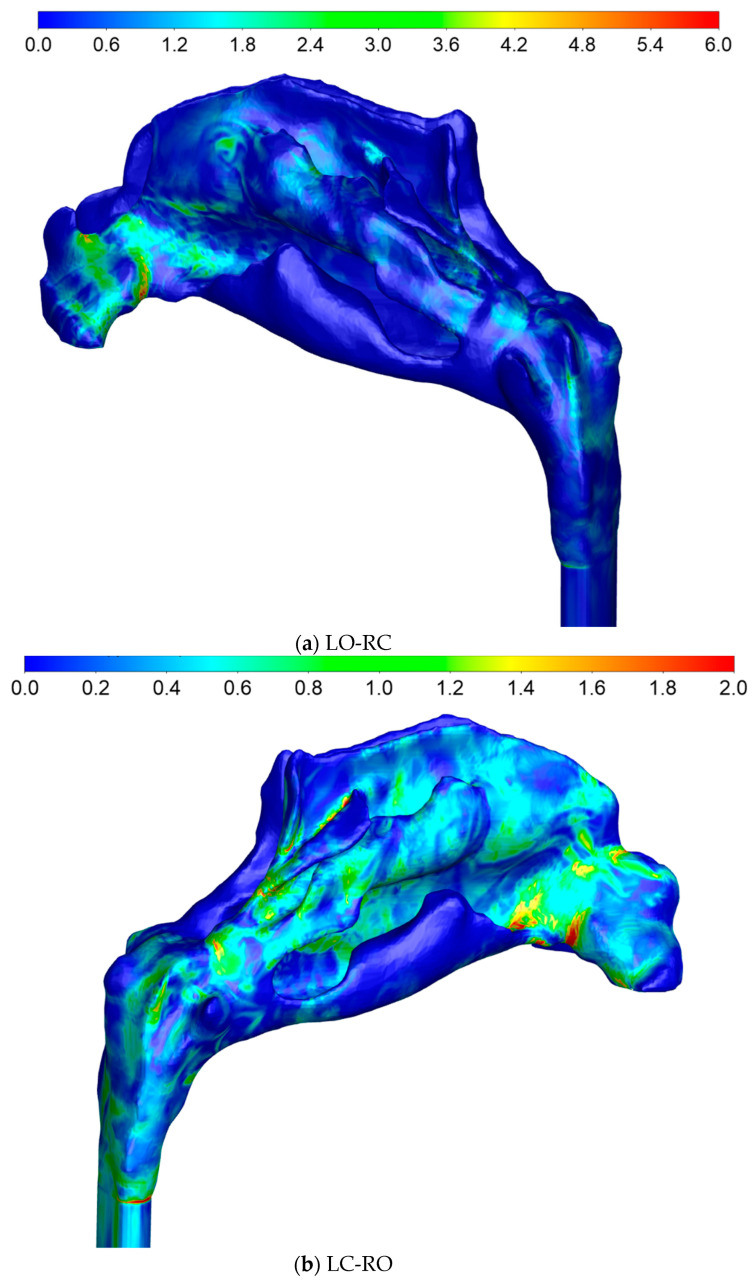
Wall shear stress contour plots (Pa) at the nasal cavity wall for (**a**) LO-RC and (**b**) LC-RO, steady state fine-mesh *kω* SST simulation of peak inspiratory flow.

**Figure 32 bioengineering-11-00239-f032:**
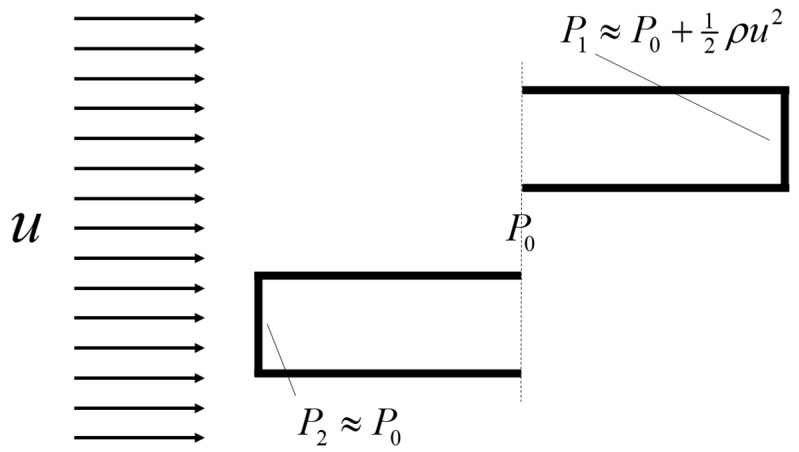
Two identical containers are placed in a flowing fluid of mass density, ρ, and uniform velocity, u, such that one (1) has its opening facing upwind and the other (2) faces downwind. The containers are aligned such that the static pressures at the openings of the containers are identical. The static pressures at the bottom of the two containers will differ by the dynamic pressure of the flowing fluid, $\frac{1}{2}\rho u^{2}$, since this is the pressure necessary to be exerted on the fluid to bring it to stagnation at the opening of the container. Thus, pressures at the bottom of the upwind and downwind facing container are approximately equal to the total and static pressures of the fluid, respectively, at the container openings. The described scenario is ideal, and “approximate” indicates that entrance effects, such as vortices in the openings of the containers, may affect the actual pressures.

## Data Availability

Transient LES case files and data presented in this study are publicly available in the NIRD Research Data Archive [14] (accessed on 1 January 2024). Additional simulation files and data are available upon request.

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
