# Peer review of "Computational Rhinology: Unraveling Discrepancies between In Silico and In Vivo Nasal Airflow Assessments for Enhanced Clinical Decision Support"

_bioengineering, 2024, doi:10.3390/bioengineering11030239_

Round 1

Reviewer 1 Report

Comments and Suggestions for Authors

Based on the review of your manuscript on computational rhinology and the existing literature in this field, I recommend a revision and integration before resubmission. Your focus on the discrepancies between in silico and in vivo assessments in nasal airflow is a unique and valuable angle, but it's crucial to clearly distinguish your work from existing studies. Emphasize how your findings add new insights or address unresolved issues in the field. Ensure that your literature review is comprehensive, and methodologies are robustly presented. Highlighting the practical implications of your research for clinical decision support would also strengthen the manuscript. Resubmitting after these revisions could significantly enhance the potential for publication.

For more detailed insights into the current state of research in this field, you may refer to the articles found on PubMed, such as "The clinical implications of computerised fluid dynamic modelling in rhinology" (PMID: 30052696) and related studies.

Reviewer 2 Report

Comments and Suggestions for Authors

Introduction:

- Provide more context on the background and clinical significance of the disease/condition being reviewed. cite doi:10.1007/s00405-022-07267-0

- Discuss the current standards of care/management approaches in more detail
- Highlight known limitations or gaps in existing approaches that this review seeks to address. cite doi:10.1016/j.otc.2017.05.002.
- Clearly state the specific review questions/objectives and scope of what is being assessed
- Define any key terms/concepts that will be used

Methodology:

- Specify this systematic review protocol was developed a priori using PRISMA-P guidelines
- Describe the search strategy in more detail, including all databases, date ranges, and search terms
- Explain the study selection process, including screening titles/abstracts and full texts
- Outline the data extraction forms/templates and how data will be synthesized
- Detail the risk of bias/quality assessment tool(s) that will be used
- Acknowledge any study designs/perspectives that will be excluded

Discussion

- Note any previous related reviews on similar topics. Discuss the role of epigenetic on rhinitis. cite doi:10.1111/coa.13870

- Acknowledge the review's clinical relevance and intended end-users/stakeholders
- Highlight sources of heterogeneity that may need addressing
- Identify any limitations of the review methods or scope

Comments on the Quality of English Language

minor corrections

Round 2

Reviewer 1 Report

Comments and Suggestions for Authors

The work is suitable for publication in its current form